# From Mean to Extreme: Formal Differential Privacy Bounds on the Success of Real-World Data Reconstruction Attacks

## Abstract

Data reconstruction attacks against machine learning (ML) models pose a strong risk of leaking sensitive data. When training ML models with differential privacy (DP), the success of such reconstruction attacks can be upper bounded. Our work explores DP's role in protecting against a category of data reconstruction attacks from literature which do not rely on prior knowledge of the data, namely analytic gradient inversion attacks. Analytic gradient inversion attacks are particularly effective and difficult to detect in real-world scenarios, operating under a threat model where an adversary can manipulate the ML model before or during training *without* needing any knowledge of the input data beyond its dimensionality.

Our theoretical contributions include (1) formulating an optimal attack strategy under the mean squared error for the specified threat model, (2) measuring the attack's success by comparing the reconstruction to the input data using three different metrics, and (3) computing theoretical bounds for these metrics. Notably, we analyse the probabilistic behaviour of the reconstruction success from expectation *(mean)* to tail behaviour *(extreme)*. Additionally, we experimentally demonstrate and visualise the validity of our optimal reconstruction strategy and highlight the relevance of our theoretical bounds by comparing them to the experimental success of the attack. Theoretically and empirically, our work underscores the protection DP provides against analytic gradient inversion attacks across varying privacy guarantees and model choices. By providing bounds for the success of data reconstruction attacks in real-world scenarios, we provide practitioners with a richer foundation for understanding specific reconstruction risks.

## 1 Introduction

Machine learning (ML) techniques and innovations have revolutionised a multitude of research and application areas, such as computer vision and natural language processing. However, the performance of ML methods is contingent on the availability of large real-world data sets. In certain domains, such as finance or medicine, data sets contain highly sensitive information about individuals and, therefore, are subject to (stronger) privacy legislation that can restrict their potentially already limited accessibility. This particularly concerns ML models since, for instance, they and/or their gradient updates store information about the training data, allowing for an (almost) perfect reconstruction of data samples (Fowl et al., 2022; Boenisch et al., 2023; Feng & Tramèr, 2024). For the acceptance of stakeholders and, by that, the availability of their data for ML training, it is thus imperative to implement protective measures. This has sparked the research area of privacy-preserving ML.

Differential privacy (DP) (Dwork et al., 2014) is the gold standard technique for providing individuals whose data is being used, e.g. to train ML models, with quantifiable privacy protection. Informally, employing DP through a privacy mechanism $\mathcal{M}$ ensures that the output of a (ML model's) computation remains almost unchanged when generated from *adjacent* input data sets $\mathcal{D}$ and $\mathcal{D}'$ that differ in only one entry $x$. Hence, DP is usually understood in the context of *membership inference attacks*, where an adversary attempts to determine whether a privatised output is derived from one of two nearly identical data sets, thereby assessing an individual's participation in the input data set. DP guarantees are typically constructed under a very

powerful adversary, who has access to the model computations and architecture and can manipulate these settings in their favour. Additionally, the adversary knows the adjacent data sets $\mathcal{D}$ and $\mathcal{D}'$ as well as the parameters of the privacy mechanism $\mathcal{M}$. The above-mentioned specifications are colloquially referred to as the *DP threat model*. In particular, the DP parameters of the privacy mechanism $\mathcal{M}$ directly modulate the protection against membership inference attacks performed by the specified adversary.

However, translating DP guarantees into explicit, quantitative protection against other attacks, such as *data reconstruction attacks* (DRAs), is neither straightforward nor trivial. As the name suggests, an adversary performing a DRA aims to recover specific training data records. DRAs are interesting to study because they pose a more significant risk compared to membership inference attacks, as they can expose sensitive information in its totality. Moreover, DRAs that aim to reconstruct $N > 1$ bits of information are inherently more challenging than membership inference attacks since the latter can be viewed as reconstructing only one bit of information (whether a subject was part of the training set or not). Therefore, an "appropriate" analysis concerning theoretical guarantees against DRAs requires *relaxing* the original DP threat model to at least remove *exactly knowing* the specific *target* data. Incidentally, when we consider an adversary less powerful than the one depicted in the DP threat model, we construct privacy guarantees under a *relaxed threat model*.

Assessing privacy guarantees in terms of the risk associated with specific attacks under a distinct, relaxed threat model allows for a more nuanced understanding of the provided protection for all parties involved. In the following, we refer to such guarantees concerning interpretable risks as *semantic guarantees* (Guo et al., 2022). In order to construct such semantic guarantees for DRAs, the attack's success needs to be evaluated, for instance, by measuring the similarity between the reconstructed and target data. Utilising DP inherently enforces theoretical upper bounds on the success of DRAs (Guo et al., 2022; Balle et al., 2022; Hayes et al., 2023; Kaissis et al., 2023a). Providing such bounds by applying DP allows individuals who share their data for training to directly control the risks of their data being disclosed.

In this work, we address DP's role in protecting against a specific form of DRA that necessitates an *analytic gradient inversion* strategy. We refer to a DRA as an *analytic gradient inversion attack* when an adversary manipulates the model and leverages intermediate gradient updates to reconstruct the input data analytically. The works of Fowl et al. (2022); Boenisch et al. (2023); Feng & Tramèr (2024) have presented state-of-the-art (SOTA) real-world DRAs that utilise an analytic gradient inversion strategy. Analytic gradient inversion attacks are feasible in federated learning (FL) setups with a malicious central server, where gradient updates are shared during training (Fowl et al., 2022; Boenisch et al., 2023). As FL is often naively employed as a privacy-preserving technique, analysing these reconstruction risks is crucial for practical purposes, particularly when handling sensitive data. However, such a threat extends beyond FL scenarios. An adversary with access to the model before and after training can insert *data traps* that encode gradient updates into model weights, allowing the reconstruction of sensitive data even after training (Feng & Tramèr, 2024). This attack is likewise highly relevant in practice, as many workflows rely on pretrained models, which get fine-tuned on private data and released afterwards. Therefore, protecting against analytic gradient inversion attacks is not only of theoretical interest but also significantly important for real-world scenarios.

Interestingly, the success of the above-mentioned attacks (Fowl et al., 2022; Boenisch et al., 2023; Feng & Tramèr, 2024) is not even dependent on adversarial knowledge beyond the input data's dimensionality. Unsurprisingly, these attacks have recently received substantial attention, as they (a) are conceivable and hard to detect in real-world scenarios, (b) allow for an (almost) perfect reconstruction success, and (c) are based on computationally low-cost analytical procedures, in contrast to optimisation based attacks (Geiping et al., 2020). However, to our knowledge, no previous work has provided theoretical semantic guarantees against analytic gradient inversion attacks tailored to a relaxed threat model *without* adversarial knowledge about the input data beyond the dimensionality.

Relaxed threat models help in understanding threats posed by adversaries that differ from those described in the DP threat model. Specifically, by determining the adversary's capabilities and the form of the attack, one can *quantify* the associated reconstruction risks and calibrate the DP parameters of a mechanism $\mathcal{M}$ to mitigate these specific risks. Previous works have investigated these semantic guarantees employing DP (e.g. Nasr et al. (2021)). However, they have almost exclusively analysed scenarios under worst-case assumptions,

in which the most powerful conceivable adversary performs the attack. Such a worst-case scenario includes knowledge about the reconstruction target. Defining this prior knowledge of an adversary is not trivial. However, at worst, the reconstruction attack in such a scenario is equivalent to matching an element of a set of candidates of the training data to a specific model or gradient. While this follows a Bayesian notion of updating a prior belief with a posterior distribution, in many practical applications, an adversary is rather interested in recovering the data sample that is not already in their possession.

Only considering the worst-case scenario given by the DP threat model comes at a cost: if the adversary is weaker than presumed, the constructed bounds become loose, overrating the adversary's success. Overestimating privacy risks is problematic because it drives practitioners to implement stricter privacy conditions, which, in turn, reduces the accuracy of the model. The resulting compromise between privacy and performance is called the privacy-utility trade-off (De et al., 2022). It imposes a dilemma on practitioners, particularly in fields where highly accurate results are critical, such as the medical domain. For example, in oncological classification or segmentation tasks, they must choose between correctly detecting tumours or accepting the risk of leaking patients' private information. Hence, in practice, it is crucial to determine and analyse the case-dependent most powerful but realistic adversary to enforce the "least amount of privacy" that sufficiently protects against such threat without overly compromising the model's performance.

In this work, we consider a relaxed threat model without adversarial knowledge about the input data beyond the dimensionality and we formulate probabilistic bounds on the reconstruction success for DRAs that perform under an analytic gradient inversion strategy, such as Fowl et al. (2022); Boenisch et al. (2023); Feng & Tramèr (2024). Our work helps practitioners make an informed decision about privacy parameters, which effectively protect against data reconstruction while at the same time not being overly pessimistic, which would result in imprecise outputs and, in the context of ML, performance losses of the resulting models. We envision that such analyses can lead to full system cards (analogously to model cards (Mitchell et al., 2019)), where the privacy risks in different contexts (i.e., threat models) are described, thereby equipping practitioners with deeper insights into specific reconstruction risks.

### 1.1 Contributions

In this work, we investigate the semantic privacy guarantees of Gaussian mechanisms against analytic gradient inversion attacks under a relaxed threat model, where the adversary can manipulate the machine learning model *before* or *during* training *without* requiring any knowledge of the input data beyond its dimensionality. Our contributions can be summarised as follows:

- In Section 4.1, we formulate an optimal attack strategy for analytic gradient inversion attacks that delivers the optimal estimator under the MSE for any target input.

- In Section 4.2.1, we measure the success of such reconstruction attacks using the MSE and PSNR, and formulate probabilistic bounds for these metrics in terms of reconstruction robustness (Balle et al., 2022).

- In Section 4.2.2, we measure the success of reconstruction attacks using NCC, and derive a theoretical upper bound for this quantity.

- In Section 5, we visualize the implications of our theoretical results and their correspondence to empirical measurements.

## 2 Related Work

Our work is parellely concerned with three different aspects: understanding a specific kind of DRAs called analytic gradient inversion attacks, translating differential privacy (DP) guarantees into quantifiable protection against such attacks and challenging the traditional DP threat model by considering scenarios where the adversary's capabilities are relaxed. Therefore, in this background section, we first review the implementation and practical significance of analytic gradient inversion attacks (see Section 2.1). We then briefly recall various definitions and concepts concerning DP to highlight the complexity of understanding privacy guarantees (see

Section 2.2). Finally, we examine previous research that has addressed semantic guarantees against data reconstruction attacks (see Section 2.3).

## 2.1 Analytic Gradient Inversion Attacks

As mentioned in the introduction, gradient inversion attacks can occur in FL setups (Fowl et al., 2022; Boenisch et al., 2023) and when practitioners utilise a contaminated pretrained model (Feng & Tramèr, 2024). By comprehending these attacks across these scenarios, we can formulate a relaxed threat model that accounts for the specific nature of the attack and the adversary's capabilities required to execute it. The concrete threat model utilised in this work can be found in Section 3.1.

Fowl et al. (2022) and Boenisch et al. (2023) concurrently proposed attacks on a decentralised, federated learning setup. They showed that potentially unnoticed modifications to a deep learning model by a malicious central server can lead to the input data being stored in the gradients of the model. In particular, they exploit that (a) linear layers are a common type of neural network component and (b) they have the property to encode their input in the gradients calculated by the chain rule (backward pass). The former aspect (a) makes them unsuspicious for any check, which is unaware of the latter (b). To conduct the attack such a linear layer is *prepended* to the original network architecture *without* replacing it. Here, the input is encoded once for each projection, i.e., a layer projecting an input $x \in \mathbb{R}^N$ to $b \in \mathbb{R}^M$, recovers $M$ versions of $x$ (bins). Specifically, the exact input can be recovered by performing an element-wise division between the gradient of the bias and the gradient of the weight of the linear layer. For batch sizes larger than one or multiple consecutive update steps, each bin encodes an arbitrary amount of inputs to the network. This is problematic, as any reconstruction of more than one (meaningful) input, is an average of its inputs, which implies that the signals of these inputs overlap and may make them meaningless. While the authors propose several sophisticated strategies to avoid this effect, it cannot be impeded entirely. Hence, from a privacy perspective, the worst case is a single sample update step (i.e. batch size is one) where no data can overlap.

Based upon the same attack principle, Feng & Tramèr (2024) proposed an extension that eliminates the need for access to intermediate gradients during training. The presented attack, however, requires that the adversary has access to the model before and after training. This distinction concerning the capabilities of the adversary makes such attacks significantly more critical for real-world scenarios, as practitioners only need to use a manipulated pretrained model for the finetuning input data to be compromised. In particular, a malicious model provider represents a realistic adversary satisfying the mentioned capabilities. Feng & Tramèr (2024) carry out this attack by designing and inserting a so-called *data trap* into the model, i.e., a part of the network where the intermediate gradients related to the data are encoded and, with high probability, not contaminated by any other gradients. From there, the adversary can perform an analytic gradient inversion after training and recover the data used for fine-tuning. To effectively retrieve multiple training inputs, it is necessary to embed several data traps into the model. However, this scenario poses a risk of multiple inputs interfering with the same trap. Thus, employing a single data trap to extract one training input represents the worst-case scenario from a privacy standpoint. All in all, Feng & Tramèr (2024) demonstrate that these attacks can be executed without causing significant performance losses and with only minimal changes to the model architecture, making detection challenging.

Fowl et al. (2022) empirically show the success of their attack against a model trained with different DP guarantees but provide no theoretical bounds for the success of their attack. Boenisch et al. (2023) mention DP as a quantifiable privacy-preserving strategy to potentially mitigate the success of their attack. Yet, they do not analyse the effect of DP on their attack strategy. Feng & Tramèr (2024) investigate the effectiveness of data traps for membership inference attacks on models trained with DP. However, they do not explore the efficacy of DRAs utilising data traps in the context of models trained with DP. The following questions remain open: If participating nodes in a FL setting train locally using DP and consequently share privatised model updates with the malicious central server, how would this impact the success of the attack? Furthermore, if practitioners fine-tune a pretrained model using DP and subsequently release it, would the data trap still enable a successful reconstruction attack?

Unequivocally, there is a breach in understanding the interaction between DP and analytic gradient inversion attacks. For instance, since none of the above-mentioned works theoretically analyse their attack strategies

under DP, none consider the "optimal" attack choices – within the analytic gradient inversion attack's category – when DP is utilised. An optimal attack strategy is necessary to quantify and generalise the protection provided by DP against *any* analytic gradient inversion attack. The lack of theoretical privacy guarantees adapted to such pressing real-world threats motivates our work.

## 2.2 Differential Privacy

Before concretely addressing previous works concerned with constructing privacy guarantees against DRAs using DP, we first recall some key concepts of DP. The definition of $(\varepsilon, \delta)$-DP is as follows:

**Definition 1** (Definition 2.4 of Dwork et al. (2014)). A randomised algorithm $\mathcal{M}$ with domain $\mathcal{X}$ is $(\varepsilon, \delta)$-differentially private if for all $\mathcal{S} \subseteq \text{Range}(\mathcal{M})$ and for all $\mathcal{D}, \mathcal{D}' \in \mathcal{X}$, with $\mathcal{D} \simeq \mathcal{D}'$:

$$\mathbb{P}\left(\mathcal{M}(\mathcal{D}) \in \mathcal{S}\right) \le e^{\varepsilon} \mathbb{P}\left(\mathcal{M}(\mathcal{D}') \in \mathcal{S}\right) + \delta, \tag{1}$$

where the relation $\simeq$ denotes that $\mathcal{D}$ and $\mathcal{D}'$ are adjacent, i.e., data sets which differ in exactly one entry $x$.

It is evident that the parameter pair $(\varepsilon, \delta)$ determines the strength of the privacy guarantee. In particular, if a randomised algorithm $\mathcal{M}$ satisfies $(\varepsilon, \delta)$-DP for sufficiently small (non-negative) values of both $\varepsilon$ and $\delta$, then the probability of $\mathcal{M}$ producing similar outputs on any pair of adjacent data sets is very high, namely $1 - \delta$. In such a case, this probabilistic guarantee ensures that it is *effectively* indistinguishable whether a specific entry was present in the data set used. However, determining "sufficiently small" values for $\varepsilon$ and $\delta$ is not straightforward, especially when practitioners must balance privacy protection and utility, and, for example, membership inference is not a (primary) privacy concern.

Fulfilling the requirements of DP can be achieved by randomising the output of a deterministic function $q : \mathcal{X} \to \mathbb{R}^N$, where $\mathcal{X}$ denotes a universe of sensitive data bases. This randomisation can be conducted by employing a DP *additive noise mechanism* $\mathcal{M}$, that adds noise drawn from an appropriate distribution – calibrated to the (global) sensitivity $\Delta_p$ of $q$ – to the deterministic output $q(\mathcal{D})$ of the function $q$ over a data set $\mathcal{D}$. The sensitivity $\Delta_p$ of $q$ is calculated using the $\ell_p$-norm: $\Delta_p(q) = \sup_{\mathcal{D}, \mathcal{D}' \in \mathcal{X}, \mathcal{D} \simeq \mathcal{D}'} \|q(\mathcal{D}) - q(\mathcal{D}')\|_p$, for $p \in [1, \infty)$.

In this work, we consider the Gaussian additive noise mechanism or *Gaussian mechanism*, the predominant approach in applications with high-dimensional data. Let $\mathcal{M}$ be the Gaussian mechanism over a query function $q : \mathcal{X} \to \mathbb{R}^N$ with sensitivity $\Delta_2 := \Delta_2(q)$, then the output of the DP mechanism $\mathcal{M}$ is given by

$$\mathcal{M}(q(D)) = q(D) + \xi, \quad \xi \sim \mathcal{N}\left(0, s^2 I_N\right), \tag{2}$$

for $I_N$ being the $N$-dimensional identity matrix and $s$ the noise multiplier. If one prefers, the variance of the mechanism's noise can be directly calibrated to the sensitivity of $q$ by expressing the noise multiplier $s$ as the product of a noise scale $\sigma$ and the sensitivity $\Delta_2$, such that $s = \sigma \Delta_2$. The choice of $s$ – or, alternatively, the choice of $\sigma$ – modulates the privacy guarantee provided by (a *single* execution of) the Gaussian mechanism $\mathcal{M}$. Specifically, the Gaussian mechanism $\mathcal{M}$, as described above (see Expression 2), satisfies $(\varepsilon, \delta_{\mathcal{M}}(\varepsilon))$-DP for

$$\delta_{\mathcal{M}}(\varepsilon) = \Phi\left(-\frac{\varepsilon s}{\Delta_2} + \frac{\Delta_2}{2s}\right) - e^{\varepsilon} \Phi\left(-\frac{\varepsilon s}{\Delta_2} - \frac{\Delta_2}{2s}\right) = \Phi\left(-\varepsilon\sigma + \frac{1}{2\sigma}\right) - e^{\varepsilon} \Phi\left(-\varepsilon\sigma - \frac{1}{2\sigma}\right), \tag{3}$$

where $\Phi$ denotes the standard normal cumulative distribution function (CDF) (see Theorem 2.7 and Corollary 2.13 from Dong et al. (2022)). Equation 3 illustrates the interplay between the parameters $\varepsilon$, $\delta$, and the noise scale $\sigma$, which together characterise the privacy guarantees of the Gaussian mechanism. In particular, since $\delta$ is a function of $\varepsilon$ and $\sigma$ (see right-hand side of Equation 3), the Gaussian mechanism satisfies $(\varepsilon, \delta)$-DP for infinitely many $(\varepsilon, \delta)$ pairs. In DP, the function $\delta_{\mathcal{M}}(\varepsilon)$ is referred to as the *privacy profile* (Balle et al., 2020) of the mechanism. However, in practice, DP mechanisms are typically calibrated to a single $(\varepsilon, \delta)$ pair, where $\delta$ is chosen to be much smaller than the inverse of number of records in the data set.

Naturally, when multiple outputs of the Gaussian mechanism are released over the same sensitive data set, the privacy guarantee deteriorates. In such cases, so-called *composition theorems* must be employed to compute the cumulative *privacy loss* and, consequently, the overall privacy guarantee. Such techniques play a crucial

role in computing the privacy guarantee when training models under DP constraints. However, a detailed discussion of composition theorems is beyond the scope of this work; we refer the reader to Dong et al. (2022) for an in-depth treatment of composition theorems specific to the Gaussian mechanism.

### 2.3 Bounding Data Reconstruction Attacks with DP

We now turn to several studies that are concerned with the construction of quantifiable guarantees against reconstruction attacks using DP.

Guo et al. (2022) employ DP to construct semantic guarantees against DRAs. In contrast to the standard DP threat model, they consider a slightly relaxed setting in which the adversary *does not* have knowledge of the specific target record they aim to reconstruct. To evaluate the success of such attacks, the authors use the expected mean squared error (MSE) between the reconstructed and the actual target record – where a lower expected MSE indicates a more successful reconstruction. Theoretically, Guo et al. (2022) show that any adversary attacking a mechanism satisfying a *specific* DP privacy guarantee – given in terms of $(\alpha, \epsilon)$ Rényi-DP (Mironov, 2017) for $\alpha = 2$ – cannot obtain a reconstruction with "low" expected MSE, thereby limiting the effectiveness of the DRA. However, Hayes et al. (2023) showed that the limitation to the specified privacy guarantee results in loose bounds concerning the adversary's reconstruction success.

Balle et al. (2022) are also concerned with formalising and bounding the success of DRAs using DP. They introduce the notion of $(\eta, \gamma)$-reconstruction robustness (ReRo) of a randomised mechanism:

**Definition 2.** [Definition 2 in Balle et al. (2022)] A randomised mechanism $\mathcal{M} : \mathcal{Z}^n \to \Theta$ is $(\eta, \gamma)$-reconstruction robust with respect to a prior $\pi$ over $\mathcal{Z}$ and a reconstruction error function $l : \mathcal{Z} \times \mathcal{Z} \to \mathbb{R}_{\geq 0}$ if for any dataset $D_- \in \mathcal{Z}^{n-1}$ and any reconstruction attack $R : \Theta \to \mathcal{Z}$:

$$\mathbb{P}_{Z \sim \pi, \Theta \sim \mathcal{M}(D_- \cup \{Z\})}[l(Z, R(\Theta)) \leq \eta] \leq \gamma.$$

In Definition 2, the reconstruction error function $l$ measures the similarity between the true data $Z$ and its reconstruction $R(\Theta)$, where $R(\Theta)$ is obtained using the randomised mechanism's output $\Theta$ and the adversary's prior knowledge $\pi$. The closer $Z$ and $R(\Theta)$ are, the lower the values of $l(Z, R(\Theta))$. Therefore, $\eta$ can be chosen in such a way that a reconstruction error above this threshold $\eta$ indicates a distorted, non-usable reconstruction, and an error below $\eta$ characterises informative, "successful" reconstructions. In turn, quantifying the probability of having an error lower than this threshold $\eta$ offers an insight into the semantic meaning of the protection provided by the DP mechanism $\mathcal{M}$. Concretely, if said probability is upper bounded by $\gamma$ for all reconstruction attacks $R$, then the probability of successfully reconstructing the input $Z$ is upper bounded by $\gamma$, and we call $\mathcal{M}$ $(\eta, \gamma)$-reconstruction robust with respect to $\pi$ and $l$. $(\eta, \gamma)$-ReRo enables practitioners to translate the randomness introduced by a DP mechanism into provable bounds on reconstruction successof an adversary.

Based on this work (Balle et al., 2022), Hayes et al. (2023) empirically analysed reconstruction robustness of a worst-case adversary, with access to all intermediate outputs (gradients), all training data except the target sample as well as an additional collection of data samples, including the target sample (prior set). Kaissis et al. (2023a) extended this work by providing closed-form ReRo-bounds for the Gaussian and the Laplacian mechanism. Under the above-mentioned worst-case assumptions, the reconstruction attack is equivalent to matching the privatised output of a query function (in the case of neural network this corresponds to the intermediate gradients) to the correct sample in the prior set. Hence, the adversary's success can be represented as a binary variable as they either succeed, i.e., achieve perfect reconstruction, or fail. The authors show that using DP formally bounds the success rate of such an adversary and fulfils $(0, \gamma)$-ReRo (Hayes et al., 2023; Kaissis et al., 2023a). Evaluating this worst-case scenario is appealing since an upper bound on the success of such a threat is also a bound on the success of any other reconstruction attack under less strict settings. Moreover, the constructed bound cannot be deteriorated by post-processing as the worst-case has control of all parameters (including the input and function) except for the introduced randomness by the DP mechanism. Therefore, no side knowledge could systematically increase the adversary's success. However, only considering the worst-case scenario comes at a cost: if the adversary is weaker than presumed, the constructed bounds become loose, overrating the adversary's success.

Kaissis et al. (2023b) challenged the idea that the worst-case scenario is realistic and instead examined privacy guarantees (including the bounds for the success of a reconstruction attack) under a slightly relaxed threat model. However, the proposed assumptions still cannot be considered to describe a typical real-world workflow either, as, for example, the adversary requires a perfect reconstruction algorithm and only checks whether it succeeded.

Cummings et al. (2024) recently proposed a taxonomy of threat models and alongside formulated bounds for a specific relaxed threat model. This analysed threat model does not impose the strict requirements to have knowledge about the concrete dataset. Furthermore, they assume that the distribution of the data is known and samples are drawn i.i.d. from this distribution. However, this implies that bounds are not over "most distinguishable" data samples. They can show that their bounds fulfil a strictly weaker notion of $(\eta, \gamma)$-ReRo, which they term distributional reconstruction robustness (DistReRo).

In opposition to that, Hayes et al. (2023) consider an adversary with dataset-level access to a chosen subset of the dataset, *including* the target sample. Thus, the goal of our work differs from Hayes et al. (2023) as they describe a posterior probability of correctly *matching* the target sample. In contrast, we describe the probabilities of reconstructing the input data to a certain quality.

## 3 Methods

### 3.1 Threat Model

As we aim to construct provable semantic guarantees against analytic gradient inversion attacks with DP, we choose our threat model to include all capabilities needed to perform SOTA real-world analytic gradient inversion attacks, e.g., Fowl et al. (2022); Boenisch et al. (2023); Feng & Tramèr (2024):

**Sufficient Threat Model for Analytic Gradient Inversion Attacks:** Consider an adversary who (1) can manipulate the architecture of a model, (2) control its hyperparameters, and (3) modify the loss function either before or during training. Assume that the adversary (4) can observe intermediate gradients during training or has access to the model after training. Further, assume that the adversary (5) has no access to or knowledge of the sensitive input dataset beyond its dimensionality.

The criteria outlined above can also be expressed using the terminology of the taxonomy proposed by Cummings et al. (2024): our threat model assumes an adversary with adaptive model access, population-level auxiliary information limited to data structure, and, in particular, no dataset-level auxiliary information.

We highlight that our subsequent analyses are not applicable when considering adversaries *with* access to prior information: such knowledge enables alternative reconstruction strategies that lie outside the scope of our current threat model. Moreover, in such cases, privacy violations must be assessed *relative* to the adversary's pre-existing knowledge – an aspect that we do not evaluate. However, in contrast to the works cited in Section 2.3, our study adopts the most relaxed threat model, which better captures the threats posed by the real-world attacks by Fowl et al. (2022); Boenisch et al. (2023); Feng & Tramèr (2024). We note that our work is *restricted* to adversaries whose goal is to perform an analytic gradient inversion attack against a ML model trained with DP-SGD (Song et al., 2013). Specifically, the parameters concerning DP are specified and remain under the control of the data-owning party.

### 3.2 Analytic Gradient Inversion Attacks

Under the threat model defined in Section 3.1, we first examine how analytic gradient inversion attacks operate against a model trained *without* DP. Consider a network whose first layer is a fully connected linear layer $f : \mathbb{R}^N \to \mathbb{R}^M$ with weight matrix $W \in \mathbb{R}^{M \times N}$ and bias term $b \in \mathbb{R}^M$. Formally, the operation of a linear layer on an input sample $X \in \mathbb{R}^N$ can be written as

$$f(X) = WX + b. \tag{4}$$

Typically, such a linear layer is succeeded by other neural network operations and a loss function, which we summarise in the term $g : \mathbb{R}^M \to \mathbb{R}$. Hence, $g$ describes the part of the network starting from the linear layer

and $g(f(X))$ denotes the network's loss function output. As customary, all network parameters are updated according to the loss function during each training step. For that purpose, the gradient of $g \circ f$ with respect to all network parameters is computed by a backward pass. We call this gradient the *global, concatenated gradient* and denote it by $G_X$. Naturally, the global gradient $G_X$ is dependent on the training step and on the input sample point $X$ of that specific training step, which is evident since model updates change from one training step to another. However, to ease notation, we do not additionally index $G_X$ with the iteration step. We recall that the adversary knows $G_X$ under the capabilities assumed by the threat model.

Therefore, for a fixed training step, besides other model updates concerning the parameters of the part of the network given by $g$, for all $j \in \{1, ..., M\}$, the adversary observes

$$\nabla_{W_j} g(f(X)) \in \mathbb{R}^N \qquad \text{and} \qquad \frac{\partial g(f(X))}{\partial b_j} \in \mathbb{R}, \tag{5}$$

namely, the gradient of $g \circ f$ with respect to $j$-th row of the matrix $W$ and the derivative of $g \circ f$ with respect to the $j$-th entry of the bias term $b$, respectively. Moreover, the adversary is aware that the gradient and the derivative in 5 are constructed by a backward pass in the following way:

$$\nabla_{W_j} g(f(X)) = \frac{\partial g(f(X))}{\partial f(X)_j} \nabla_{W_j} f(X)_j = \frac{\partial g(f(X))}{\partial f(X)_j} X, \tag{6}$$

$$\frac{\partial g(f(X))}{\partial b_j} = \frac{\partial g(f(X))}{\partial f(X)_j} \frac{\partial f(X)_j}{\partial b_j} = \frac{\partial g(f(X))}{\partial f(X)_j}, \tag{7}$$

for all $j \in \{1, ..., M\}$. Note that the gradient in Equation 6 is a scaled version of the input $X$ and that the multiplicative factor $\frac{\partial g(f(X))}{\partial f(X)_j}$ on the right-hand side of Equation 6 equals the observed update with respect to the $j$-th entry of the bias given in Equation 7. Thus, if there exists $j' \in \{1, ..., M\}$, such that the update $\frac{\partial g(f(X))}{\partial b_{j'}} \neq 0$, the adversary can reconstruct the input sample $X$ analytically by performing

$$\nabla_{W_{j'}} g(f(X)) \oslash \frac{\partial g(f(X))}{\partial b_{j'}} = X, \tag{8}$$

where $\oslash$ denotes the entry-wise division. Computing 8 is possible for all $j \in \{1, ..., M\}$ such that $\frac{\partial g(f(X))}{\partial b_j} \neq 0$.

Under our threat model, the adversary can modify the model to facilitate their analytic reconstruction attack. Specifically, if necessary, they can insert the fully connected linear layer $f$, as given in Equation 4, as a first layer into the network or even replace it entirely. From this point onward, we therefore consider neural networks of the form $g \circ f$. Similarly, we assume the adversary chooses the batch size $B$ to equal one to ensure that only one input sample $X$ is used per iteration step and avoid contaminating the model updates with the information from different samples. For a more detailed discussion concerning the batch size $B$, we refer to Section 2.1.

### 3.3 Analytic Gradient Inversion Attacks under DP

From now on, we consider training networks of the form $g \circ f$, as stated in Section 3.2, with DP to protect the training data against analytic gradient inversion attacks and evaluate the provided protection. In the context of neural networks, applying DP is usually achieved by training with DP-SGD (Song et al., 2013). Since in DP-SGD the privatised quantity is the global gradient, the noise must be calibrated to the sensitivity of the global gradient, which is not (necessarily) bounded and can be hard to compute. Hence, the DP-SGD algorithm is based on two main steps: (1) Clipping the $\ell_2$-norm of the global, concatenated gradient to a predefined bound $C$ in order to have an artificial bound on the sensitivity and (2) adding calibrated, zero-centered Gaussian noise to the gradient. The hyperparameter $C$ is usually called the maximum gradient norm.

For a network $g \circ f$, for a fixed iteration step, the global gradient $G_X$ has the following form:

$$G_X = \left[ \nabla_{W_1} g(f(X))^T, \frac{\partial g(f(X))}{\partial b_1}, ..., \nabla_{W_M} g(f(X))^T, \frac{\partial g(f(X))}{\partial b_M}, G_{X,P}^T \right]^T, \tag{9}$$

where $G_{X,P}$ denotes the concatenated gradient of $g \circ f$ with respect to the rest of the parameters of the network, where all vectors are unrolled to scalars. To induce a bound on the norm of the global gradient $G_X$, $G_X$ is multiplied by the clipping term:

$$\beta_C(X) := \frac{1}{\max\left(1, \frac{\|G_X\|_2}{C}\right)}. \tag{10}$$

We note that $\beta_C(X)$ is dependent on the iteration step by definition and it decreases with increasing norm of the global gradient $G_X$. Next, independent and identically distributed (i.i.d.) noise samples $\xi$ are drawn from a multivariate Gaussian distribution $\mathcal{N}(\mathbf{0}, C^2\sigma^2 I)$, where $I$ is the identity matrix of the dimension of the global gradient $G_X$ and $\mathbf{0}$ denotes zero vector also matching the dimension of $G_X$. Hence, when a network is trained with DP-SGD, the adversary observes *noisy* versions of the model updates in Equations 6 and 7 that can be used for the gradient inversion attack:

$$\widetilde{\nabla}_{W_j} := \beta_C(X)\nabla_{W_j}g(f(X)) + \xi_j = \beta_C(X)\frac{\partial g(f(X))}{\partial f(X)_j}X + \xi_j, \qquad \xi_j \sim \mathcal{N}(\mathbf{0}_N, C^2\sigma^2 I_N), \tag{11}$$

$$\widetilde{\nabla}_{b_j} := \beta_C(X)\frac{\partial g(f(X)))}{\partial b_j} + \xi_j' = \beta_C(X)\frac{\partial g(f(X))}{\partial f(X)_j} + \xi_j', \qquad \xi_j' \sim \mathcal{N}(0, C^2\sigma^2), \tag{12}$$

for all $j \in \{1, ..., M\}$. Under non-DP training, the gradient in Equation 6 stored a scaled version of the target input point $X$ (see Section 3.2). In contrast, when employing DP, the model updates have the form presented in 11 where the stored target $X$ has not only been rescaled but also perturbed by adding Gaussian noise. Therefore, for each $j \in \{1, ..., M\}$, Expression 11 denotes a *privatised* version of the input $X$. It is easy to see that the introduced randomness impedes performing a simple division to recover the input sample $X$, as was possible in Equation 8, where no DP was used.

Using the distribution of the sampled noise, the noisy model updates in Equations 11 and 12 can be expressed as samples from random variables in the following way

$$\widetilde{\nabla}_{W_j} \stackrel{d}{=} Y_j, \quad \text{for} \quad Y_j \sim \mathcal{N}\left(\beta_C(X)\frac{\partial g(f(X))}{\partial f(X)_j}X, C^2\sigma^2 I_N\right), \tag{13}$$

$$\text{and} \quad \widetilde{\nabla}_{b_j} \stackrel{d}{=} z_j, \quad \text{for} \quad z_j \sim \mathcal{N}\left(\beta_C(X)\frac{\partial g(f(X))}{\partial f(X)_j}, C^2\sigma^2\right), \tag{14}$$

for all $j \in \{1, ..., M\}$, where $\stackrel{d}{=}$ means equal in distribution. Even though the adversary cannot perform the division in 8 to reconstruct $X$, they can use the privatised global gradient, in particular, the observations $\widetilde{\nabla}_{W_1}, ..., \widetilde{\nabla}_{W_M}, \widetilde{\nabla}_{b_1}, ..., \widetilde{\nabla}_{b_M}$ and the knowledge about their distributions, as given in 13 and 14, to design an estimator for the target point $X$. Ultimately, this estimator serves as the reconstruction $Y_X$ of $X$. We highlight that $\widetilde{\nabla}_{W_1}, ..., \widetilde{\nabla}_{W_M}$ are $M$ observed, privatised versions of $X$, whereby $M$ can be modulated by the adversary since it denotes the number of rows of the matrix $W$ that specifies the linear layer $f$ (see 4). In Section 4.1, we address parameter $M$'s importance for the estimator and, thus, for the attack's success.

Without making assumptions on the part of the network given by $g$, it is impossible to determine whether, for all or for some iteration steps, the part of the gradient denoted by $G_{X,P}$ (see 9) and, thereby, its privatised version contain usable information to estimate the target $X$. Therefore, we first focus on the privatised model updates given by $\widetilde{\nabla}_{W_1}, ..., \widetilde{\nabla}_{W_M}, \widetilde{\nabla}_{b_1}, ..., \widetilde{\nabla}_{b_M}$ to construct an analytically tractable estimator for $X$ and ignore the remaining part of the privatised global gradient. Later, we address the influence $G_{X,P}$ has on formulating an estimator for $X$ and, in Corollary 1, we specify the choice of the part of the network given by $g$ that renders the analytic gradient inversion attack with the highest reconstruction success in terms of the MSE.

Reconstructing the target $X$ using *only* the observations $\widetilde{\nabla}_{W_1}, ..., \widetilde{\nabla}_{W_M}$ is "not far from" solving a classical mean estimation problem. However, the means of the distributions of the samples $\widetilde{\nabla}_{W_1}, ..., \widetilde{\nabla}_{W_M}$ are not $X$ but rescaled versions of $X$, namely $\beta_C(X)\frac{\partial g(f(X))}{\partial f(X)_1}X, ..., \beta_C(X)\frac{\partial g(f(X))}{\partial f(X)_M}X$, respectively (see 13). Removing the dependency on the scaling factors $\beta_C(X)\frac{\partial g(f(X))}{\partial f(X)_1}, ..., \beta_C(X)\frac{\partial g(f(X))}{\partial f(X)_M}$, is necessary for creating an unbiased

estimator for $X$ such as the sample mean (see Proposition B.1 in the appendix). Since the adversary also observes noisy versions of these scaling factors, namely $\widetilde{\nabla}_{b_1}, ..., \widetilde{\nabla}_{b_M}$ (see 14), we can differentiate between two cases: one in which the adversary directly employs $\widetilde{\nabla}_{b_1}, ..., \widetilde{\nabla}_{b_M}$ to reconstruct $X$, and another one in which $\widetilde{\nabla}_{b_1}, ..., \widetilde{\nabla}_{b_M}$ are *first* used to estimate the scaling factors, which are *then* employed to estimate $X$.

In the former case, the adversary can try to integrate the observed privatised gradients $\widetilde{\nabla}_{W_1}, ..., \widetilde{\nabla}_{W_M}, \widetilde{\nabla}_{b_1}, ..., \widetilde{\nabla}_{b_M}$ by dividing entry-wise $\widetilde{\nabla}_{W_j}$ by $\widetilde{\nabla}_{b_j}$ for all $j \in \{1, ..., M\}$ such that $\widetilde{\nabla}_{b_j} \neq 0$. Consequently, to determine the behaviour of $\widetilde{\nabla}_{W_j} \oslash \widetilde{\nabla}_{b_j}, j \in \{1, ..., M\}$, they combine 13 and 14 and consider

$$\widetilde{\nabla}_{W_j} \oslash \widetilde{\nabla}_{b_j} \overset{d}{=} V_j, \quad \text{for} \quad V_{j,i} \sim \frac{\mathcal{N}\left(\beta_C(X)\frac{\partial g(f(X))}{\partial f(X)_j}x_i, C^2\sigma^2\right)}{\mathcal{N}\left(\beta_C(X)\frac{\partial g(f(X))}{\partial f(X)_j}, C^2\sigma^2\right)}, \tag{15}$$

where $V_{j,i}$ denotes the $i$-th entry of $V_j$ for all $i \in \{1, ..., N\}$, and $V_{j,i}$ are pairwise independently distributed. The ratio of two Gaussian distributions, as the one in 15, follows the Cauchy distribution. Generally, this distribution has no defined statistical moments (Marsaglia, 2006), such as an expectation. Although, under certain conditions these moments can be approximated (Marsaglia, 2006; Díaz-Francés & Rubio, 2013), no general statements about the behaviour of the samples $\widetilde{\nabla}_{W_j} \oslash \widetilde{\nabla}_{b_j}, j \in \{1, ..., M\}$, or the asymptotic behaviour of estimators constructed using said samples, can be made if the statistical moments do not exist. This is particularly problematic considering that the distribution in 15 varies for all $j \in \{1, ..., M\}, X \in \mathcal{D}$ and each iteration step. Therefore, we assume that the adversary refrains from employing 15 to estimate $X$.

However, alternatively, the adversary can use $\widetilde{\nabla}_{b_1}, ..., \widetilde{\nabla}_{b_M}$ to estimate the scaling factors $\beta_C(X)\frac{\partial g(f(X))}{\partial f(X)_1}, ..., \beta_C(X)\frac{\partial g(f(X))}{\partial f(X)_M}$ first and then, separately, use these estimators to rescale the observed privatised gradients 11, reformulate their distributions 13 and solve the mean estimation problem. Due to the inherent randomness of estimators, it is evident that the accuracy of any subsequent estimation $Y_X$ of $X$ depends on the precision of the estimation of the scaling factors $\beta_C(X)\frac{\partial g(f(X))}{\partial f(X)_1}, ..., \beta_C(X)\frac{\partial g(f(X))}{\partial f(X)_M}$. Hence, to compute generalisable bounds on the success of the attack, we must account for all possible scenarios concerning the estimation of these scaling factors, within the constraints of the given attack category and threat model. It is particularly important to consider and bound the *best-case scenario* from the adversary's perspective, namely, when the scaling factors are estimated perfectly. Notably, scaling factors can also be approximated through other strategies independently of Equation 12. In the simplest case, this can be achieved by imposing constraints on the data; for example, in the case of images, by assuming pixel values range from 0 to 255. As a consequence, *from a privacy perspective*, upper bounding the success of a reconstruction $Y_X$ when the scaling factors are estimated is equivalent to upper bounding the success when the scaling factors are known. Therefore, in the following, we assume that these scaling factors are available to the attacker.

## 4 Theoretical results

This section presents all the theoretical results of our work and is divided into two parts. The first part, Section 4.1 is written from an adversarial perspective, with the aim of maximising the success of the reconstruction attack. Specifically, it outlines the optimal attack strategy under the MSE and derives the corresponding optimal estimator. The second part, Sections 4.2 and 4.3, adopt a privacy-preserving perspective, in which the success of the attack is explicitly bounded. Section 4.2 highlights the protection provided by the Gaussian mechanism against analytic gradient inversion attacks. Lastly, Section 4.3 presents a pessimistic perspective by adapting our bounds to the case in which the adversary executes the attack multiple times or, equivalently, observes intermediate gradients across all training iterations.

All proofs supporting our results are included in Appendix B.

### 4.1 Construction of the Optimal Analytic Gradient Inversion Attack under DP

Let $X = (x_1, ..., x_N)^T \in \mathcal{D} \subseteq \mathbb{R}^N$ be a fixed reconstruction target point, and also fix the training iteration step. Again, we emphasise that from now on, we assume the adversary has access to the scales

$\beta_C(X)\frac{\partial g(f(X))}{\partial f(X)_1}, ..., \beta_C(X)\frac{\partial g(f(X))}{\partial f(X)_M}$ which they use in the estimation problem of the target $X$, as explained in the last part of Section 3.3.

Having access to $\beta_C(X)\frac{\partial g(f(X))}{\partial f(X)_1}, ..., \beta_C(X)\frac{\partial g(f(X))}{\partial f(X)_M}$ implies that the adversary can rescale the observed privatised gradients $\widetilde{\nabla}_{W_1}, ..., \widetilde{\nabla}_{W_M}$ (see 11), construct their sample average $\hat{X}_M$, and use $\hat{X}_M$ as an unbiased estimator for $X$:

$$\hat{X}_M := \frac{1}{M}\sum_{j=1}^{M}\frac{1}{\beta_C(X)\frac{\partial g(f(X))}{\partial f(X)_j}}\widetilde{\nabla}_{W_j}. \tag{16}$$

Consequently, using the distribution of the observations $\widetilde{\nabla}_{W_j}$, $j \in \{1, ..., M\}$, given in 13, the distribution of the sample mean $\hat{X}_M$ is given by:

$$\hat{X}_M \sim \mathcal{N}\left(X, \frac{1}{M^2}\sum_{j=1}^{M}\frac{C^2\sigma^2}{\beta_C(X)^2\frac{\partial g(f(X))}{\partial f(X)_j}^2}I_N\right). \tag{17}$$

Under our threat model (see Section 3.1), the adversary is able to modify the model's architecture, hyper-parameters and loss function to facilitate their attack. Therefore, they are capable of adjusting the neural network $g \circ f$ to minimise the coordinate-wise variance of the sample mean $\hat{X}_M$ in 17, aiming to increase the probability that $\hat{X}_M$ takes a value "close enough" to its expectation, namely to the target $X$. One potential strategy could be to increase the number of rows $M$ of the matrix $W$ of the first (linear) layer $f$ (see 4) of the network. However, for increasing $M$, the norm of the global gradient $G_X$ (see 9) increases and the clipping term $\beta_C(X)$ (see 10) decreases requiring a careful consideration of the interaction between $M$ and $\beta_C(X)$. In the appendix (see Proposition B.2), we show that the coordinate-wise variance of the sample mean $\hat{X}_M$ is lower bounded by $\sigma^2\|X\|_2^2$ for $M \to \infty$. Intuitively, this implies that, from the adversary's perspective, the variability of the estimator $\hat{X}_M$ can only get "so good", but after a certain value for $M$ it will not further decrease and, in particular, it will never be zero such that $\hat{X}_M$ results in the undistorted target point.

The next proposition specifies the concrete modifications to the network that render the sample mean $\hat{X}_M$ with the lowest variance:

**Proposition 1.** *Let the part of neural network given by $g$ be replaced by the loss function $\mathcal{L} : \mathbb{R}^M \to \mathbb{R}$ with $\mathcal{L}(f(X)) = \mathbf{1}_M^T f(X)$, where $\mathbf{1}_M$ is the $M$-dimensional 1-vector, and*

$$M \geq \max\left(1, \left\lceil\frac{C}{\min_{X \in \mathcal{D}\setminus\{\mathbf{0}_N\}}\|X\|_2}\right\rceil\right), \tag{18}$$

*where $\lceil\cdot\rceil$ denotes the function that rounds up its argument to the nearest integer, then $\frac{1}{M^2}\sum_{j=1}^{M}\frac{C^2\sigma^2}{\beta_C(X)^2\frac{\partial g(f(X))}{\partial f(X)_j}^2}$ is minimal and takes the value $\sigma^2\|X\|_2^2$.*

Next, we identify several consequences of Proposition 1: Assume the adversary replaces the entire network by the linear function $f : \mathbb{R}^N \to \mathbb{R}^M$ given by $f(X) = WX$, $W \in \mathbb{R}^{M \times N}$ and the loss function $\mathcal{L} : \mathbb{R}^M \to \mathbb{R}$ given by $\mathcal{L}(f(X)) := \mathbf{1}_M^T f(X)$, then $\frac{\partial g(f(X))}{\partial f(X)_j} = \frac{\partial \mathcal{L}(f(X))}{\partial f(X)_j} = 1$ for all $j \in \{1, ..., M\}$. This implies that

$$\beta_C(X)\frac{\partial g(f(X))}{\partial f(X)_1} = ... = \beta_C(X)\frac{\partial g(f(X))}{\partial f(X)_M} = \beta_C(X), \tag{19}$$

reducing the approximation of $M$ scales, presented in the last part of Section 3.3, to the estimation of a single scale, namely $\beta_C(X)$.

Moreover, by Proposition 1, for all $M'$ satisfying 18, the sample mean $\hat{X}_{M'}$ is distributed in the following way:

$$\hat{X}_{M'} \stackrel{d}{=} \hat{X}, \quad \text{for} \quad \hat{X} \sim \mathcal{N}\left(X, \sigma^2\|X\|_2^2 I_N\right). \tag{20}$$

If $C/\min_{X \in \mathcal{D}\setminus\{\mathbf{0}_N\}}\|X\|_2 > 1$, Proposition 1 implies that setting $M$ below $\left\lceil\frac{C}{\min_{X \in \mathcal{D}\setminus\{\mathbf{0}_N\}}\|X\|_2}\right\rceil$, results in a sample mean with a higher variability in terms of its variance. Therefore, from the adversary's perspective,

engineering $M$ appropriately can be viewed informally as counteracting the addition of "overproportional" noise to the observed privatised model updates, since no clipping is triggered and thus the noise can be calibrated to a smaller value. However, increasing $M$ beyond the bound in 18 does not further minimise the variance of the estimator in Expression 20. In particular, the choices of the network presented in Proposition 1 suggest that no privatised version of the gradient $G_{X,P}$ (see 9) can improve the estimator's variance in 20 or improve the performance of the analytic reconstruction attack. A visual representation of Proposition 1 is provided in Figure 1.

Since the adversary cannot directly manipulate the randomisation introduced by the DP mechanism to their benefit, their strategy can only rely in increasing the number of observed privatised versions of the input $X$ and aggregating them, as presented in Proposition 1. As mentioned before, this approach makes the analytic reconstruction attack a classic mean estimation problem whose success depends on the variance of the estimator in 20. Due to the nature of the clipping term $\beta_C(X)$ (see 9), any non-usable observed information contained in $G_{X,P}$, increases the variance of this estimator (see proof of Proposition 1 in Appendix B). Moreover, usable observed information contained in $G_{X,P}$ can be optimally incorporated into the mean estimation problem if it is a privatised version of the input $X$ of the form in 11, making $G_{X,P}$ the gradient of a linear layer. Thus, the (insertion of the) linear layer and choosing $M$ as in Proposition 1 makes $G_{X,P}$ redundant at best. If $G_{X,P}$ is not the gradient of a linear layer but contains usable information about the target $X$, "sacrificing" the information in $G_{X,P}$ in favour of maximising the information extracted from the linear layer yields the most statistically efficient reconstruction in terms of estimator variance. In other words, $G_{X,P} \neq \mathbf{0}$ can only decrease the reconstruction quality by adding to the gradient norm and triggering the addition of "overproportional" noise.

If the adversary chooses $M$ according to Proposition 1, then the behaviour of the sample mean $\hat{X}_M$ coincides with $\hat{X}$'s one (see 20) independently of the specific value of $M$. Moreover, it is worth noting that $\hat{X}$ is independent of the clipping norm $C$ and its variance is exactly calibrated to the norm of the target $X$. Next, we examine some properties of $\hat{X}$:

**Proposition 2.** $\hat{X}$ *is the minimum variance unbiased estimator (MVUE) for $X$. Moreover, the expected mean squared error between the target $X$ and $\hat{X}$ is given by:*

$$\mathbb{E}_X[\mathrm{MSE}_X(X,\hat{X})] = \sigma^2 \|X\|_2^2. \tag{21}$$

A minimum variance unbiased estimator (MVUE) for the target $X$ is desirable when dealing with the statistical estimation problem because such an estimator achieves the lowest expected mean squared error (MSE) and has the lowest variability in terms of its variance compared to all other estimators for X constructed with the same observations $\widetilde{\nabla}_{W_1}, ..., \widetilde{\nabla}_{W_M}$. The following result further emphasises the relevance of the estimator $\hat{X}$ regarding reconstructing the target $X$:

**Theorem 1.** *Using the* MSE *as an optimality criterion, $\hat{X}$ is the best achievable estimator and, thus, reconstruction for the target point $X$.*

In this work, we utilise the MSE as an optimality criterion, because the mean squared error is equal to zero, i.e., $\mathrm{MSE}_X(X,\hat{X}) = 0$, if and only if $X = \hat{X}$, namely whenever the adversary perfectly reconstructs the target $X$. A perfect reconstruction denotes the best-case scenario for the adversary or conversely the worst-case scenario concerning privacy preservation. Therefore, by Theorem 1, analysing the error between $X$ and $\hat{X}$, specifically lower bounding this error, is sufficient to upper bound the reconstruction success of the adversary. Theorem 1 is one of the key results of this section and plays a central role in formulating the *optimal attack strategy* (see Corollary 1) for analytic gradient inversion attacks under the assumed threat model (see Section 3.1):

**Corollary 1.** *Assume the adversary replaces the original model by a single, fully connected linear layer $f(X) = WX$ of a single input $X \in \mathbb{R}^N$ by a learnable matrix $W \in \mathbb{R}^{M \times N}$, $M = \max\left(1, \left\lceil \frac{C}{\min_{X \in \mathcal{D} \setminus \{0_N\}} \|X\|_2} \right\rceil\right)$, and sets the loss function $\mathcal{L} : \mathbb{R}^M \to \mathbb{R}$ to be $\mathcal{L}(W, X) := \mathbf{1}_M^T W X$. Then, $\hat{X}$, as defined in Equation 20, is the MVUE with the lowest expected* MSE *and variability in terms of the variance compared to all other possible MVUE obtained under other choices regarding the model's architecture, hyperparameters and loss function.*

Original $\qquad M < \left(\frac{C}{\|X\|_2}\right)^2 \qquad M \geq \left(\frac{C}{\|X\|_2}\right)^2$

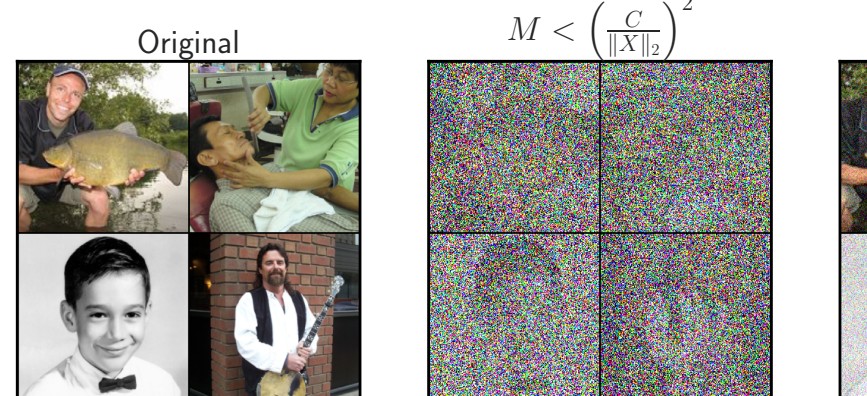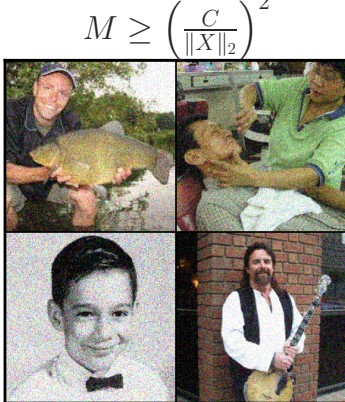

Figure 1: Demonstration of how an adversary can enforce obtaining the least amount of noise on the reconstruction for constant privacy parameters by adjusting the parameter $M$. If they increase $M$ such that $M \geq \left(\frac{C}{\|X\|_2}\right)$, they will get the least amount of additive noise (right column). If they do not exceed this threshold, the additive noise is stronger ("overproportional") for the same privacy parameters $C$ and $\sigma$ (middle column). Middle column: $M = 1$, right column: $M = 1000$. All other parameters remain constant for both reconstructions: $\sigma = 5 \cdot 10^{-4}$, $C = 5.0 \cdot 10^3$, $N = 150528$. All images are from the ImageNet dataset (Deng et al., 2009).

Corollary 1, which builds on Proposition 1 and Theorem 1, serves as a foundational result for our study, forming the basis for our subsequent theoretical developments. Its practical relevance lies in the following implication: upper bounding the reconstruction success of an adversary that performs the attack described in Corollary 1 automatically bounds the success of an adversary that performs any other analytical gradient inversion attack under DP. Therefore, we focus on the attack specified in Corollary 1 as well as on the optimal estimator $\hat{X}$ as in 20. From now, we set $\hat{X}$ to be the reconstruction of the adversary.

## 4.2 Determining the Reconstruction Success

In this section, we aim to determine the privacy effect DP has on the success of analytic reconstruction attacks performed against models privatised by the Gaussian Mechanism. To derive generalizable bounds, we consider the optimal attack strategy under MSE, as presented in Corollary 1. Accordingly, we evaluate how well the adversary's reconstruction $\hat{X}$, as defined in Equation 20, resembles its respective target input point $X$ via the mean squared error (MSE), the peak signal-to-noise ratio (PSNR) and the normalised crossed-correlation (NCC). For an introduction and motivation of these metrics, we refer the reader to Appendix A.1.

These metrics compare the input point $X$ to its reconstructed version $\hat{X}$ in different ways. However, we interpret it as high reconstruction success whenever they indicate high similarity between $X$ and $\hat{X}$. We associate the reconstruction success of the adversary with the privacy parameters $\sigma$ and $C$ that determine the privacy guarantee of the Gaussian mechanism. Additionally, we express the MSE and PSNR as random variables, examine their tail behaviour and provide guarantees in terms of the well-established notion of reconstruction robustness.

### 4.2.1 Reconstruction Success Measured by the MSE, PSNR and ReRo

First, we recover the value of the expected $\mathrm{MSE}_X(X, \hat{X})$ for all $X \neq \mathbf{0}_N$ stated in Equation 21, namely $\mathbb{E}_X[\mathrm{MSE}_X(X, \hat{X})] = \sigma^2 \|X\|_2^2$. Next, we consider the cases $C \leq \min_{X \in \mathcal{D} \setminus \{\mathbf{0}_N\}} \|X\|_2$ and $C > \min_{X \in \mathcal{D} \setminus \{\mathbf{0}_N\}} \|X\|_2$ separately, and examine how they impact the value of the expected MSE as well as possible lower bounds for it.

Assume $C \leq \min_{X \in \mathcal{D} \setminus \{\mathbf{0}_N\}} \|X\|_2$ holds. Then, the expected reconstruction error measured by the MSE between $X$ and and the reconstruction $\hat{X}$ can be lower bounded by the product of the squared maximum

gradient norm $C^2$ and the squared noise scale $\sigma^2$ for all $X \in \mathcal{D} \setminus \{\mathbf{0}_N\}$, implying that the MSE between $X$ and any of its unbiased estimators can be lower bounded by $C^2\sigma^2$. Hence, the higher the variance of the additive noise mechanism $\mathcal{M}$, the higher the expected error in terms of the MSE. It is, thus, easy to see that the privacy of the DP-mechanism and the expected MSE are positively correlated.

However, if we assume $C > \min_{X \in \mathcal{D} \setminus \{\mathbf{0}_N\}} \|X\|_2$ holds, then the expected reconstruction error measured by the MSE between $X$ and the reconstruction $\hat{X}$ equals $\sigma^2 \|X\|_2^2$ for all $X \in \mathcal{D} \setminus \{\mathbf{0}_N\}$ and cannot be lower bounded solely using the DP parameters $C$ and $\sigma$. In such a case, the value of the expected MSE (see Equation 21) illustrates the effect that increasing $M$, i.e., the number of reconstructions that are retrieved, has on the reconstruction success of the adversary, namely it cancels the addition of "overproportional" noise to all $X$ such that $C > \|X\|_2$. Specifically, in this case, the error measured by the MSE is exactly calibrated to the norm of each target $X$. Due to the optimality of the estimator $\hat{X}$, we conclude as before, that the expected MSE between $X$ and any of its unbiased estimators can be lower bounded by the product of the squared norm $\|X\|_2^2$ and $\sigma^2$ if $C > \min_{X \in \mathcal{D} \setminus \{\mathbf{0}_N\}} \|X\|_2$.

We refrain from making general claims regarding the occurrence of the cases $C \leq \min_{X \in \mathcal{X} \setminus \{\mathbf{0}_N\}} \|X\|_2$ and $C > \min_{X \in \mathcal{X} \setminus \{\mathbf{0}_N\}} \|X\|_2$ since, in practice, the maximum gradient norm $C$ is an additional hyperparameter not necessarily dependent on $\min_{X \in \mathcal{D} \setminus \mathbf{0}_N} \|X\|_2$. For instance, $C = 1$ is a very typical choice for classification tasks or whenever practitioners want to simplify all of their calculations (e.g., De et al. (2022)). We also refer to De et al. (2022) as a valuable source for understanding the interplay between all hyperparameters and parameters during training under DP constraints, which ultimately impact both privacy and performance.

To illustrate the behaviour of the $\mathrm{MSE}_X(X, \hat{X})$, we turn to the calculation of probabilistic bounds for this quantity. Concretely, fixing the target $X$, we can formulate the $\mathrm{MSE}_X(X, \hat{X})$ as a random variable determined by the randomness of the reconstruction $\hat{X}$ and explore its tail behaviour. We are particularly interested in computing and bounding the probability of the error measured by the MSE falling below a certain threshold $\eta$. The choice of the threshold $\eta$ indicates the interpretation of such a probabilistic bound since, for instance, $\eta$ can be chosen such that an error measured by the $\mathrm{MSE}_X(X, \hat{X})$ below $\eta$ characterises informative reconstructions.

**Theorem 2.**

$$\mathrm{MSE}_X(X, \hat{X}) \stackrel{d}{=} \frac{\sigma^2 \|X\|_2^2}{N} \cdot Y \quad with \quad Y \sim \chi_N^2, \tag{22}$$

where $\chi_N^2$ denotes the central chi-squared distribution with $N$ degrees of freedom. In particular, for $\eta$ given,

$$\mathbb{P}_{\hat{X}}(\mathrm{MSE}_X(X, \hat{X}) \leq \eta) = \Gamma_R \left( \frac{N}{2}, \frac{N\eta}{2\sigma^2 \|X\|_2^2} \right), \tag{23}$$

where $\Gamma_R$ is the regularised gamma function.

By Theorem 2, the $\mathrm{MSE}_X(X, \hat{X})$ between any target point and its reconstruction is a chi-squared distributed random variable with $N$ degrees of freedom multiplied by the product between $1/N$ and $\sigma^2 \|X\|_2^2$. Thus, it is no coincidence that the latter ratio equals the expected MSE given in Equation 21 because the expectation of a chi-squared distributed random variable with $N$ degrees of freedom equals $N$. Therefore, we can recover the interpretation of $\sigma^2 \|X\|_2^2$ stated in the first part of this section, specifically the behaviour of this quantity for the cases $C \leq \min_{X \in \mathcal{D} \setminus \{\mathbf{0}_N\}} \|X\|_2$ and $C > \min_{X \in \mathcal{D} \setminus \{\mathbf{0}_N\}} \|X\|_2$. Moreover, Theorem 2 demonstrates that $\sigma^2 \|X\|_2^2$ determines not only the expectation of the MSE but also its distribution, highlighting the influence that the DP parameters $C$ and $\sigma$ and the parameter $M$ – where $M$ is chosen by the adversary – have on the error measured by the MSE and consequently on the reconstruction success of the adversary measured by the MSE.

Informally, the chi-squared distribution with $N$ degrees of freedom converges to the normal distribution for $N \to \infty$. Therefore, for cases where the dimension of the training data is "very high", it is pertinent to additionally analyse the asymptotic behaviour of the $\mathrm{MSE}_X(X, \hat{X})$ for $N \to \infty$. In the appendix, in Proposition B.3, we formally present the convergence in distribution of the $\mathrm{MSE}_X(X, \hat{X})$ for $N \to \infty$. However, we leave out an in-depth discussion of the result in Proposition B.3.

Expressing the MSE as a random variable enables the formulation of its tail behaviour as given in 23. We note that the result in Equation 23 resembles the notion of reconstruction robustness as given in Definition 2.

In the following, we generalise this result to lose the dependency on the target point $X$ and translate the probabilistic bound in 23 into reconstruction robustness, leading to the first key takeaway of this section:

**Proposition 3.** *Let $\eta$ given. Then, for all $X \in \mathcal{D} \setminus \{\mathbf{0}_N\}$,*

$$\mathbb{P}_{\hat{X}}\left(\mathrm{MSE}_X\left(X, \hat{X}\right) \leq \eta\right) \leq \Gamma_R\left(\frac{N}{2}, \frac{N\eta}{2\sigma^2 \min_{X \in \mathcal{D}} \|X\|_2^2}\right), \tag{24}$$

*where $\Gamma_R$ is the regularised gamma function. Moreover, the DP-mechanism $\mathcal{M}$ is $(\eta, \gamma(\eta))$-reconstruction robust with respect to the MSE for any reconstruction and $\gamma(\eta) = \Gamma_R\left(\frac{N}{2}, \frac{N\eta}{2\sigma^2 \min_{X \in \mathcal{D}} \|X\|_2^2}\right)$. If $C \leq \min_{X \in \mathcal{X}} \|X\|_2$ holds, then $\mathcal{M}$ is $(\eta, \gamma'(\eta))$-reconstruction robust with respect to the MSE for any reconstruction and $\gamma'(\eta) = \Gamma_R\left(\frac{N}{2}, \frac{N\eta}{2\sigma^2 C^2}\right)$.*

Depending on the training data, other reconstruction error functions might be more informative when assessing the similarity between the target $X$ and the reconstruction $\hat{X}$. Concretely, we consider the peak signal-to-noise-ratio (PSNR) (see Definition 4) as a reconstruction quality function next. We note that the PSNR is defined via the MSE. In particular, as the MSE decreases, the PSNR increases, implying that if there exists one reconstruction that minimises the MSE and this minimum is non-zero, this reconstruction maximises the PSNR. Thus, by Theorem 1 and Corollary 1, the chosen reconstruction $\hat{X}$ is also optimal with respect to the PSNR.

Next, we provide probabilistic bounds for the PSNR between $X$ and $\hat{X}$ to determine its tail behaviour. To do so, we assume $\max_{X \in \mathcal{D}} \max(X)$ and $\min_{X \in \mathcal{D}} \min(X)$ are known quantities. This information is known whenever the adversary has access to the range of the data, such as when the training data are images. Unlike the MSE, the PSNR increases as the similarity between $X$ and $\hat{X}$ grows. Therefore, we formulate a bound for the probability of the event when the PSNR exceeds a certain threshold $\eta$. In contrast to the MSE, $\eta$ can be chosen such that a similarity measured by the PSNR above $\eta$ characterises informative reconstructions.

**Proposition 4.** *Assume $\max_{X \in \mathcal{D}} \max(X)$ and $\min_{X \in \mathcal{D}} \min(X)$ are known quantities. Then, for all $X \in \mathcal{D} \setminus \{\mathbf{0}_N\}$,*

$$\mathbb{P}_{\hat{X}}\left(\mathrm{PSNR}_X\left(X, \hat{X}\right) \geq \eta\right) \leq \Gamma_R\left(\frac{N}{2}, \frac{N\tilde{\eta}(\eta)}{2\sigma^2 \min_{X \in \mathcal{D} \setminus \{\mathbf{0}_N\}} \|X\|_2^2}\right), \tag{25}$$

*for $\tilde{\eta}(\eta) := 10^{-\frac{\eta}{10}}\left(\max_{X \in \mathcal{D}} \max(X) - \min_{X \in \mathcal{D}} \min(X)\right)^2$, and for $\Gamma_R$ being the regularised gamma function. If $C \leq \min_{X \in \mathcal{D}} \|X\|_2$ holds, then*

$$\mathbb{P}_{\hat{X}}\left(\mathrm{PSNR}_X\left(X, \hat{X}\right) \geq \eta\right) \leq \Gamma_R\left(\frac{N}{2}, \frac{N\tilde{\eta}(\eta)}{2\sigma^2 C^2}\right), \tag{26}$$

*for all $X \in \mathcal{D} \setminus \{\mathbf{0}_N\}$.*

To reformulate the result in Proposition 4 into reconstruction robustness terms, we utilise the negative PSNR ($-$PSNR) as a reconstruction error function. This leads us to the second key takeaway of this section:

**Proposition 5.** *Assume $\max_{X \in \mathcal{D}} \max(X)$ and $\min_{X \in \mathcal{D}} \min(X)$ are known quantities, and let $Y_X$ be any possible reconstruction. Then, for all $X \in \mathcal{D} \setminus \{\mathbf{0}_N\}$,*

$$\mathbb{P}_{Y_X}\left(\mathrm{PSNR}_X\left(X, Y_X\right) \geq \eta\right) \leq \Gamma_R\left(\frac{N}{2}, \frac{N\tilde{\eta}(\eta)}{2\sigma^2 \min_{X \in \mathcal{D} \setminus \{\mathbf{0}_N\}} \|X\|_2^2}\right), \tag{27}$$

*for $\Gamma_R$ being the regularised gamma function and $\tilde{\eta}(\eta)$ as defined in Proposition 4. In particular, the DP-mechanism $\mathcal{M}$ is $(-\eta, \tilde{\gamma}(\tilde{\eta}(\eta)))$-reconstruction robust with respect to the negative PSNR ($-$PSNR) for any analytic reconstruction and $\tilde{\gamma}(\tilde{\eta}(\eta)) = \Gamma_R\left(\frac{N}{2}, \frac{N\tilde{\eta}(\eta)}{2\sigma^2 \min_{X \in \mathcal{D} \setminus \{\mathbf{0}_N\}} \|X\|_2^2}\right)$. Moreover, if $C \leq \min_{X \in \mathcal{D}} \|X\|_2$ holds, then the DP-mechanism $\mathcal{M}$ is $(-\eta, \tilde{\gamma}'(\tilde{\eta}(\eta)))$-reconstruction robust with respect to $-$PSNR for any reconstruction, $\tilde{\gamma}'(\tilde{\eta}(\eta)) = \Gamma_R\left(\frac{N}{2}, \frac{N\tilde{\eta}(\eta)}{2\sigma^2 C^2}\right)$.*

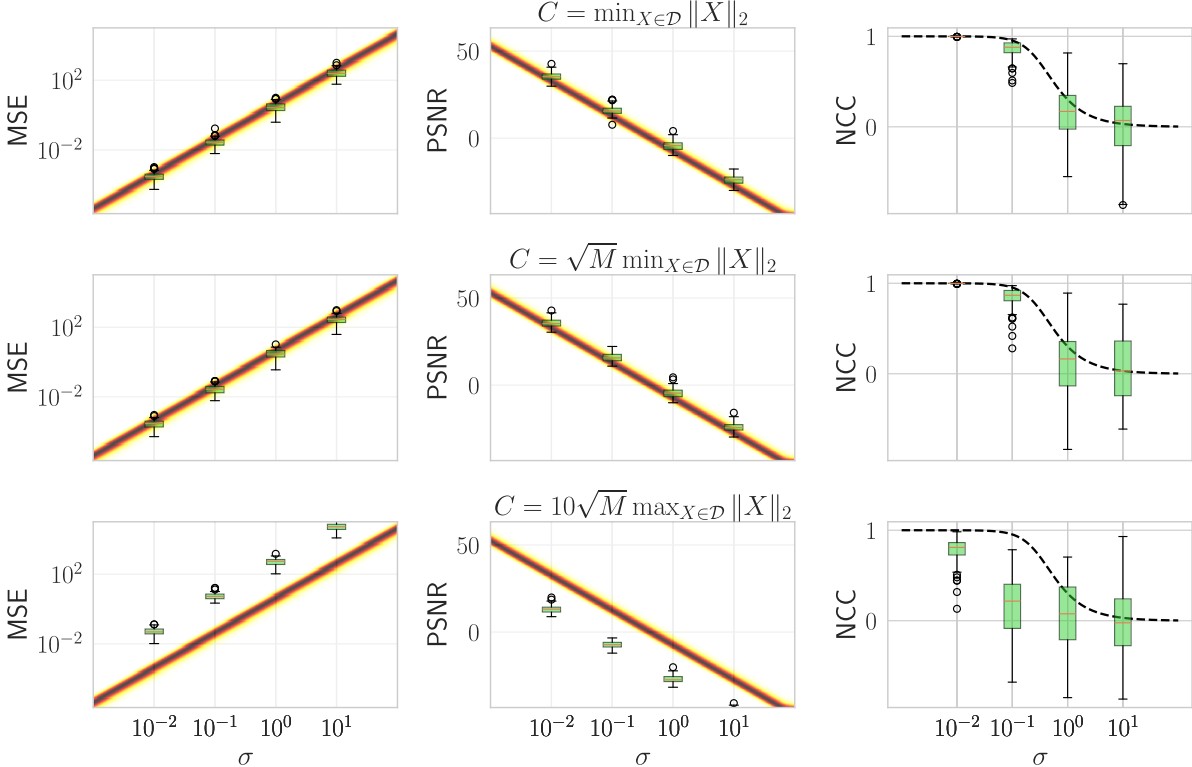

Figure 2: Empirical reconstruction results overlaid on our theoretical results. Each column shows one reconstruction metric: Left: MSE, Middle: PSNR, Right: NCC. For MSE and PSNR, the probability density function (PDF) of the theoretical distribution is encoded in colour, where white corresponds to low values and black to high values along each $\sigma$. For the NCC, the dashed line shows the theoretical bound. For all plots, we have performed empirical reconstructions of 100 four-dimensional data samples with $M = 100$, which are sampled from a uniform distribution $\mathcal{U}(0,1)$. The resulting reconstruction metrics are displayed as boxplots, where the orange line shows the mean reconstruction result, the green box displays the interquartile range (IQR) from the first quartile to the third quartile, and the whiskers extend to the data point, which is the last within 1.5 of the IQR. Further outliers are displayed as circles. *First row* shows the case where $C = \min_{X \in \mathcal{D}} \|X\|_2$. Hence, the sensitivity is exhausted, and the resulting reconstruction cannot be further improved for this scenario even if $M = 1$. *In the second row*, we have $C = \sqrt{M} \min_{X \in \mathcal{D}} \|X\|_2$. Here, several data samples only exceed the sensitivity threshold because multiple observations are combined. *In the last row*, no data sample is clipped as $C$ is set to be larger than any data norm and $M$ has not been chosen according to the optimal attack. Therefore, the sensitivity limit is not reached, and "overproportional" noise is applied. This decreases the reconstruction success.

The CDF of the $\mathrm{MSE}_X(X, \hat{X})$ and the survival funciton (SF) of the $\mathrm{PSNR}_X(X, \hat{X})$ for different values of $\sigma$, $N$ and $\min_{X \in \mathcal{D} \setminus \{\mathbf{0}_N\}} \|X\|_2$ are depicted in Figure 3. We remark that bounding the probability of a specific reconstruction loss allows for interpreting our results in terms of $(\eta, \gamma(\eta))$-ReRo (compare Figure 3 to Propositions 3 and 5).

### 4.2.2 Reconstruction Success Measured by the NCC

Let us consider the linear correlation given by the NCC between a target $X$ and its reconstruction $\hat{X}$. The NCC can be measured empirically using their entries $\{x_1, ..., x_N\}$ and $\{\hat{x}_1, ..., \hat{x}_N\}$ as sample sets, thereby computing the *sample* NCC. The calculation of the sample NCC and a brief comment regarding its usefulness can be found in the appendix in Proposition B.4 and Remark B.1, respectively. However, in this section, we concentrate on the (theoretical) correlation between $X$ and $\hat{X}$.

To compute the NCC between $X$ and $\hat{X}$, we apply the following strategy: First, let the target $X$ be fixed. We assume there exists a continuous, one-dimensional random variable $x$ distributed in such a way that $\{x_1, ..., x_N\}$ are probable samples from this distribution. In particular, we let $[\min_{i \in \{1,...,N\}} x_i, \max_{i \in \{1,...,N\}} x_i]$ be the support of $x$ and set $\mathrm{Var}(x) \leq \|X\|_2^2/N$. The latter assumption is motivated by the following fact: $X$ is an $N$-dimensional vector, hence, drawing a random element from its entries $\{x_1, ..., x_N\}$ can be represented by a discrete, uniformly distributed random variable with support $\{x_1, ..., x_N\}$ and variance bounded by $\|X\|_2^2/N$. However, for our analysis, $x$ needs to be a continuous random variable. Hence, we define $x$ to be continuous, but maintain the range of the support and the variance of its discrete counterpart. Analogously, we construct the continuous, one-dimensional random variable $\hat{x}$ such that $\hat{x} := x + \zeta$, for $\zeta \sim \mathcal{N}(0, \sigma^2\|X\|_2^2)$ independent of $x$. By definition, $\hat{x}_1, ..., \hat{x}_N$ are probable samples of the random variable $\hat{x}$. It is easy to see, that measuring the correlation between $x$ and $\hat{x}$ is equivalent to measuring the correlation between $X$ and $\hat{X}$.

**Proposition 6.** *Let $x$ and $\hat{x}$ be the two random variables as defined above. Then,*

$$\mathrm{NCC}(x, \hat{x}) = \sqrt{\frac{1}{1 + \sigma^2\|X\|_2^2/\mathrm{Var}(x)}} \leq \sqrt{\frac{1}{1 + \sigma^2 N}}. \tag{28}$$

By Proposition 6, the NCC $(x, \hat{x})$ is determined by the ratio between the entry-wise variance of the reconstruction $\hat{X}$ given by $\sigma^2\|X\|_2^2$ and variability within the target $X$ given by $\mathrm{Var}(x)$. Therefore, if this ratio is not "high enough", we conclude from Equation 28 that the DP mechanism's noise is insufficient to disrupt the linear association between $x$ and $\hat{x}$, or, conversely, between $X$ and its reconstruction $\hat{X}$. Moreover, due to the nature of the reconstruction $\hat{X}$ (see 20) a perfect (or "good enough") correlation between $x$ and $\hat{x}$ implies a perfect (or "good enough") reconstruction of the target $X$. In particular, Equation 28 equals one if and only if the noise scale $\sigma$ equals zero, i.e., when no noise is added to model updates before release and the observed gradient $G_X$ is not privatised during training. Since any non-affine transformation of the observed privatised gradient would distort the linear dependency between $X$ and the non-privatised gradient, it is easy to see that $\hat{X}$ is an optimal reconstruction with respect to the NCC as well.

If $\sigma \neq 0$, we observe that the NCC $(x, \hat{x})$ decreases with increasing noise scale $\sigma$. However, Proposition 6 demonstrates that the NCC between $X$ and $\hat{X}$ highly depends on the dimension $N$. In particular, the right-hand side of Expression 28 $\in \mathcal{O}(\sqrt{1/N})$. Thus, for increasing dimension $N$, the correlation between $X$ and $\hat{X}$ measured by the NCC decreases independently of the noise of the DP mechanisms as long as $\sigma \neq 0$. Therefore, in the context of our work, for high values of $N$, very low NCC values do not necessarily indicate low reconstruction success. In such a case, the NCC bound in 28 must be interpreted in a "comparative manner" relative to a case-specific threshold $\eta$ that distinguishes informative and non-informative reconstructions. For instance, if only reconstructions with NCC values exceeding $\eta$ are considered informative, and $\eta$ is greater than the right-hand side of Equation 28, then no reconstruction constructed by the adversary can be deemed useful. However, a key limitation of the NCC is that it cannot be used to compare reconstruction success across scenarios with differing input dimensionalities $N$ (see Section 6).

### 4.3 Information Accumulation Across Multiple Attacks

Our theoretical results (see Section 4.2) hold for a single iteration of the Gaussian mechanism in DP-SGD. We now turn to exploring how these results *could* change under *repeated* adversarial attacks. This scenario is equivalent to one in which the adversary observes intermediate gradients across all - or more than one - training iterations.

Based on the model updates from a single iteration, we concluded in Section 4.1 that the optimal estimator an adversary can construct – in terms of the MSE – is distributed as shown in Equation 20. In particular, we observed that the coordinates of the reconstruction for the target $X$ exhibit a variance of $\sigma^2\|X\|_2^2$.

When the model is trained using DP-SGD with a fixed batch size of one, the *subsampling* step of the algorithm randomly selects a single input for each iteration. As a result, the adversary *cannot* control whether the same target record $X$ is selected across multiple attacks – or, equivalently, training iterations. Specifically, if the algorithm selects one input point uniformly at random from a training dataset $D$, then the probability that a

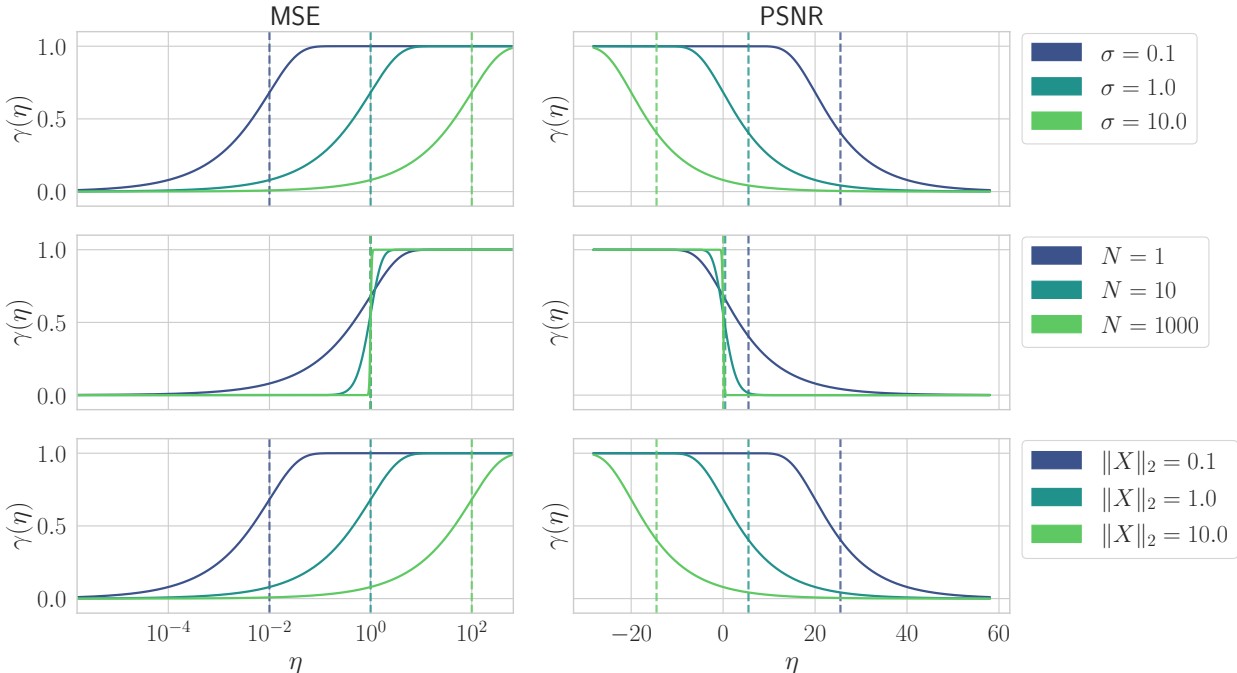

Figure 3: Cumulative Distribution Function of the MSE and Survival Function of the PSNR for varying parameters $\sigma$, $N$ and $\min_{X \in \mathcal{X}} \|X\|_2$. If not varied then parameters are set to be $\sigma = 1$, $N = 1$, $\max_{X \in \mathcal{D}} \|X\|_2 = 1$. Vertical dashed lines show the corresponding bounds on the expected values. For a given error threshold $\eta$, we have a risk probability lower than or equal to $\gamma(\eta)$. Note that lower values of the PSNR denote better reconstruction results. Hence, for $(\eta, \gamma(\eta))$-ReRo the negative PSNR needs to be considered (compare Proposition 5).

particular target record is selected exactly $k$ times is given by $\binom{\hat{K}}{k} \cdot \frac{1}{|D|^k}$ when the attack is performed $\hat{K}$ times. *If* this case arises, the reduction in variance of the reconstruction coordinates can be quantified as:

**Proposition 7.** *Assuming the adversary can match the $k$ reconstructions $\hat{X}_1, ..., \hat{X}_k$ to the same data sample $X$, they can average them. Let $\hat{X}_{\text{avg}} = \frac{1}{k} \sum_{j=1}^{k} \hat{X}_j$ denote the averaged reconstructed vector. Then the following holds for the ith entry of $\hat{X}_{\text{avg}}$:*

$$\mathbb{E}[\hat{x}_{i_{\text{avg}}}] = x_i \quad and \quad \text{Var}\left[\hat{x}_{i_{\text{avg}}}\right] = \frac{\sigma^2 \|X\|_2^2}{k}. \tag{29}$$

Therefore, *with probability $\binom{\hat{K}}{k} \cdot \frac{1}{|D|^k}$*, by Proposition 7, performing $\hat{K}$ attacks can reduce the variance per coordinate of the averaged reconstructed target. Note that the attack cannot be performed an unlimited number of times. Specifically, it can be executed at most $T$ times, i.e. $\hat{K} \leq T$, where $T$ denotes the number of iterations of DP-SGD. If we consider the case where the adversary observes the model updates across *all* training iterations $T$, we can adapt the results by modifying the probability that the algorithm selects $X$ as input $k$ times to be $\binom{T}{k} \cdot \frac{1}{|D|^k}$.

With probability unequal zero, the adversary *may* mitigate the privacy-preserving effects of the noise and thereby improve their reconstruction performance. However, the result above was derived under the *naive* assumption that the adversary is able to *distinguish* that their reconstructions $\hat{X}_1, ..., \hat{X}_k$ correspond to the same target point $X$. To evaluate the validity of such an assumption, it is necessary - amongst other considerations - to carefully devise an adversarial strategy for identifying when multiple reconstructions correspond to the same target record. We leave such an analysis, as well as the examination of our bounds – presented in Section 4.2 – under multiple attacks, for future work.

## 5 Empirical evaluation

In this section, we show the implications of our bounds and their correspondence to empirical results.

### 5.1 Visualisation and Interpretation

First, we illustrate the effect of the parameters $\sigma$, $C$, $M$, $N$ and $\|X\|_2$ on our results to allow for a more intuitive understanding of their behaviour. The code to reproduce our figures can be found on https://anonymous.4open.science/r/FromMeanToExtreme-46D1.

In Figure 1, we demonstrate the influence of $M$ on the reconstruction result, thereby illustrating Proposition 1. Only by selecting an $M$ large enough to exceed $\left(\frac{C}{\|X\|_2}\right)^2$ no overproportional noise is added and the reconstruction cannot be further improved. The same effect is present in Figure 2. Here, we display the behaviour of MSE, PSNR, and NCC under variation of the privacy parameters $C$ and $\sigma$. For the MSE and PSNR, we plot the PDF for each value of $\sigma$ as colour coding – according to the results in Theorem 2 and Proposition 4 – where white areas have low probability and dark show high probability areas. For the NCC, we plot the bound on the expected reconstruction success according to Proposition 6 . We create an artificial dataset consisting of 100 four-dimensional data points sampled from a uniform distribution $\mathcal{U}(0,1)$. An adversary attempts to reconstruct these data points using the attack specified in Corollary 1. The empirical reconstruction success measured by the respective metrics is displayed as boxplots. We observe, analogously to Figure 1, only if the clipping threshold is exceeded, either because the data norm is larger than C (first row) or because the adversary set $M$ to be large (second row), the empirical reconstruction results overlap with the worst case distributions we have derived earlier. However, if this threshold is not exceeded then the additive noise is larger, which in turn increases the MSE and respectively decreases the PSNR and NCC.

In Figure 3, we visualize the influence of $\sigma$, $N$, and $\min_{X \in \mathcal{D} \setminus \{\mathbf{0}_N\}} \|X\|_2$ on the CDF of the MSE and PSNR as presented in Propositions 3 and 5. If a parameter is not varied in a specific row this parameter is also set to be 1. Unsurprisingly, we observe in the first row that increasing the noise scale $\sigma$ leads to smaller probabilities for high-fidelity reconstructions. In the second row, we observe that the distributions become "steeper" for a changing data dimensionality, i.e., they are not spread out as much for high-dimensional data. This means that for high dimensional data it becomes increasingly unlikely to observe errors, which largely differ from the expected error. Of note, the expected error for the MSE remains constant, but changes for the PSNR. This is due to the conversion to an exponential scale, a transformation which the expected value is not robust against. In the last row, we see the impact of the $\ell_2$-norm of the data sample on the reconstruction error distribution. Here, we observe that if all other parameters remain unchanged, data samples with larger norms have higher errors. This is because the MSE is scaled linearly along with its input. More concretely, the same data sample and reconstruction have a different MSE if both are multiplied by the same constant, namely the multiplication of the constant and the previous MSE, although semantically the reconstruction contains the same information (compare Supplementary Figure 5). The PSNR is principally robust against this behaviour as it incorporates the data range. However, for the above experiment, the data range was kept constant.

Our work attempts to address a central question: How can practitioners choose the privacy parameters $\sigma$ and $C$ for their specific model to protect the training data? As explained in Section 4.1, the adversary can counteract the privatising effect of the maximum gradient norm $C$ under the given threat model. Therefore, $C$ does not (necessarily) influence the reconstruction success and can be set depending on the model application to retrieve the best-performing outcome. However, reconstruction success is determined by the selected noise scale $\sigma$ and depends on the norm of the data points and their dimension $N$. Given a fixed data set with samples of the same dimension N, practitioners can choose some data samples and observe the impact various choices of the noise scale $\sigma$ can have on their reconstruction success (see results in Sections 4.2.1 and 4.2.2). This is particularly important for assessing thresholds $\eta$ (see Definition 2) that describe informative reconstructions in terms of the MSE or PSNR. Understanding what are appropriate choices for such thresholds, practitioners can calculate the probability $\gamma$ of occurrence of such reconstructions and obtain the levels of protection determined by the reconstruction robustness. We highlight that no one general

threshold for the MSE, PSNR and NCC determines a successful reconstruction *for all* cases as one can see, for instance, in Supplementary Figure 6.

## 5.2 Understanding & Interpreting Privacy in Our Framework

While our theoretical bounds provide guarantees on reconstruction success under specific conditions, it is crucial to understand how these results translate into practical guidance for practitioners designing privacy-preserving systems. We emphasise that our work does not aim to prescribe optimal privacy parameters but rather to offer a rigorous framework for evaluating the security implications of chosen settings.

The $(\eta, \gamma)$-ReRo framework allows users to define acceptable levels of reconstruction risk based on their specific application and data sensitivity. We emphasise that this is a decision practitioners must make, not one we can lift from them, considering local laws, guidelines, scenarios, and datasets, but also the negative effect of overly stringent privacy protection on the utility of the resulting models. Here, it might come in handy to consider the combination of all three metrics we provide to gain a comprehensive understanding of the risk level and better understand its implications. Moreover, it is important to note that our bounds are designed for a specific threat model and will not hold for stronger threat models (where prior knowledge on the data is available) and may be loose for weaker threat models (with no adaptive model access). Specifically, if the threat model fits our considered setting and the acceptable risk level is chosen, our results can directly enable the choice of appropriate privacy parameters ($\sigma$ and $C$), and by that provide a mathematical guarantee that the above choice on the risk level cannot be violated. Thus, practitioners can use these bounds to determine whether a given set of parameters adequately enforces their desired level of protection against known reconstruction attacks.

It's important to recognise that quantities like $(\varepsilon, \delta)$, commonly used in differential privacy, represent different ways of quantifying the risk associated with an output, just as our $(\eta, \gamma)$-ReRo bounds do. All these metrics aim to capture the probability of a successful reconstruction attack. Crucially, all can be derived from the underlying privacy parameters ($\sigma$, $C$, and $T$). While $(\varepsilon, \delta)$ are often directly specified by practitioners, recent studies have shown that interpreting these values in terms of actual privacy loss can be challenging (Cummings et al., 2021; Franzen et al., 2022; Nanayakkara et al., 2023). In contrast, our bounds aim to provide a more directly interpretable measure of reconstruction risk.

However, it's also important to acknowledge the current state of composition analysis. For $(\varepsilon, \delta)$, there is extensive research on how repeated applications of differentially private mechanisms (composition) and data subsampling affect overall privacy loss, allowing for tight bounds. Currently, our framework relies on a simpler, worst-case bound for multiple time steps, which may be conservative – meaning it likely overestimates the actual risk. Further research is needed to develop more refined composition theorems specifically tailored to our $(\eta, \gamma)$-ReRo framework.

Importantly, these different metrics are complementary. If you have established values for $(\varepsilon, \delta)$, and therefore know the corresponding privacy parameters ($\sigma$, $C$, and $T$), you can always estimate the risk of successful reconstruction using our bounds – providing an alternative perspective on the overall privacy loss. This allows practitioners to leverage existing privacy analyses alongside our framework for a more comprehensive understanding of their system's security.

## 6 Discussion and Conclusion

In this work, we formalise bounds over three reconstruction metrics for any data protected by the Gaussian mechanism. This is motivated by several state-of-the-art attacks, which found ways to extract such privatised data in settings and are designed to be real-world applicable. Providing bounds for the best current reconstruction attacks allows practitioners to make informed decisions to defend against analytic gradient inversion attacks based on mathematical guarantees. Considering the problem of the so-called privacy-utility trade-off, it is important to choose privacy parameters adjusted to the specific threat model in order to achieve the lowest utility penalty while maintaining an acceptable level of data security.

The MSE, PSNR, and NCC show different but complementary notions of reconstruction success. The MSE is a standard, wide-spread error metric. However, a notable drawback in practice is that it is only comparable to other MSE-values if it is zero and thus, the data is perfectly reconstructed. Namely, general assessments regarding the quality of a reconstruction based on the MSE can only be made on a comparative basis with the exception of an MSE equal to zero which signifies that the data is perfectly reconstructed. For many applications such as optimisation tasks, this is sufficient. However, for a metric measuring reconstruction success, it is desirable to be comparable to any other value. Moreover, the MSE is not robust to scaling, implying that if the data and its reconstruction are multiplied by the same scalar value, the resulting MSE changes (see Figure 5) even though no real development in the difference between the data and its reconstruction occurred. Therefore, if the privacy parameters are decided with respect to a certain bound on the MSE, practitioners must be aware of the exact scale of their dataset. PSNR can correct for this effect by setting the value range appropriately. The NCC is, by design, robust against any (positive) multiplication of the original data sample or its reconstructed version. This is because it only measures the linear correlation between these entities. Most importantly, for the MSE and PSNR, we can, for any set of privacy parameters, calculate the risk of being above a certain value, which corresponds to the notion of $(\eta, \gamma)$-ReRo. This gives any practitioner the maximum flexibility of choosing a reconstruction error and its risk, which is acceptable for their workflow. We argue that for a holistic decision for a certain set of privacy parameters, practitioners should consider all three metrics for their specific dataset in order to decide on bounds that sufficiently protect against reconstruction.

Even though an adversary cannot manipulate the DP mechanism, we showed they can improve their reconstruction result by setting $M$ to be "large enough" so that even for large sensitivity thresholds $C$, the data is clipped and no overproportional noise is added. However, we demonstrate that the randomisation introduced by the DP mechanism still bounds the reconstruction success independently of the choice of $M$. Moreover, we note that increasing $M$ also increases the computational complexity in $\mathcal{O}(M)$, whereas the enforced increase in the gradient norm is only in $\mathcal{O}(\sqrt{M})$. Therefore, in real-world applications, an adversary needs to trade off the likelihood of an optimal reconstruction against the computational requirements. At the same time, the AI practitioner has an interest in setting the clipping threshold $C$ not too large, as this would again lead to overproportional additive noise and thus lead to stronger utility losses of a trained network architecture. Hence, for any defence evaluation, the optimal reconstruction under a specific set of privacy parameters should be considered as a realistic outcome.

We note that while our bounds are directly transferable to some of the currently best real-world reconstruction attacks, none of these attacks exploits data priors. Depending on the specific data, an adversary might have related data or prior knowledge that they can exploit to further improve their reconstruction success. This was, for example, empirically demonstrated by Schwethelm et al. (2025). In that case, our bounds also do not hold anymore, and other bounds, which consider prior knowledge, need to be applied, e.g., Hayes et al. (2023). However, years of research in generating artificial data are proof that modelling the distribution of a specific kind of data is not straightforward, especially not in providing a mathematical description of it. Hence, it is likely that only optimistic (our work) and pessimistic (Hayes et al., 2023) mathematical bounds can be provided.

One limitation of our work is that the NCC cannot be interpreted in a $(\eta, \gamma)$-ReRo notation. This is due to the fact that the NCC is a descriptive property of the interaction between two random variables or two data samples such as an image and its reconstructed version. The NCC is determined by the covariance between these quantities and their standard deviations, attributes that are deterministic in our work. Therefore, although we have established a bound indicating that the NCC does not exceed a certain value, individual samples drawn from the distribution are likely to deviate from this bound due to the probabilistic nature of the random variable (see Remark B.1). Furthermore, we note that the NCC varies drastically for different dimensions and impedes comparison of the reconstruction success between scenarios with unequal dimensionality. Especially for $N \to \infty$, only perfect reconstructions will lead to NCC $> 0$, rendering its practical use in very high-dimensional settings to be limited.

Other well-established metrics, such as the normalized mutual information (NMI), structural similarity (SSIM) or even perceptual losses, also empirically seem to directly correspond to the influences we observed (see Figure 4). Deriving mathematical bounds on these metrics is left for future work. Analogously, in this work,

we have considered the case for a single query of one data sample. However, a central field of research in the context of differentially private neural network training is accounting the privacy loss over repeated queries with subsampling amplification. Repeated queries would obviously allow for a better reconstruction as the result of these queries could be averaged, which does not affect the underlying signal but, in expectation, cancels out the additive noise. Subsampling, on the other hand, would impede this strategy, as the adversary needs to correctly match queries which are based on the same sample. It remains for future work to analyse how these counteracting effects affect the results of our work.

The overarching question of providing bounds for adversarial attacks remains: how can we optimally choose the *least* amount of privacy in order to not introduce utility losses but still provide reasonable protection? We argue that solely investigating worst-case bounds introduces stronger privacy-utility trade-offs than necessary. Our work provides the theoretical bounds on the risks for analytic gradient inversion attacks against the Gaussian mechanism and allows for a more precise evaluation of the potential privacy leakage in these settings. Specifically, we allow a practitioner to tune their privacy parameters from *mean* attack success to *extreme* attack success given by the expected reconstruction success and the tail behaviour of the reconstruction, respectively. However, we note that our bounds should be seen in context and as an augmentation to the worst-case bounds, as adversaries exploiting data priors outperform the reconstruction success our bounds suggest. We see our work as a first step towards a broad suite of threat model analyses as part of a full system model card, which provide a tailored risk report for practitioners for their specific settings in addition to a contextualisation of how changes in the capabilities of the adversary can influence the attack success probabilities.

### Broader Impact Statement

This paper presents work that aims to enhance the understanding and applicability of formal privacy-preserving methods in machine learning workflows. It has several important implications. First, we see our work as a first step towards better understanding and formalising the risks data providers and AI practitioners face under various threat models. Second, by enabling more tailored choices of privacy budgets, we hope to mitigate privacy-utility trade-offs and address the ethical dilemma of balancing data privacy with model accuracy.

However, it is important to acknowledge potential negative consequences. Our bounds, while providing a quantifiable measure of reconstruction risk, could be misinterpreted as offering absolute protection against all attacks. Specifically, our analysis focuses on analytic gradient inversion attacks under specific assumptions (no prior knowledge), and adversaries exploiting data priors or employing more sophisticated techniques may still succeed in reconstructing sensitive information. Furthermore, an overreliance on our bounds without considering the broader context of a system's security could lead to a false sense of security. Therefore, we emphasize that our work should be viewed as one component of a comprehensive privacy strategy, rather than a standalone solution. We believe open discussion and responsible dissemination of these findings are crucial for fostering informed decision-making in the field of privacy-preserving machine learning.

### Author Contributions

### Acknowledgments

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

## A  Background

### A.1  Metrics

In this section, we introduce and motivate three metrics used in our analysis: the mean squared error (MSE), the peak signal-to-noise ratio (PSNR) and the normalised cross-correlation (NCC). These metrics allow us to assess the similarity and relation between an input point and its reconstruction and, consequently, determine the quality of said reconstruction.

In the following, let $X \in \mathbb{R}^N$, $N \in \mathbb{N}$, represent a fixed target input point and $Y_X \in \mathbb{R}^N$, be its estimator that serves as its reconstructed version. Moreover, we let $x_i$ and $y_{i,X}$, for $i \in \{1, ..., N\}$, denote the $i$-th element of $X$ and $Y_X$, respectively. First, we turn to the mean squared error (MSE), defined as average of the squared distances between the entries of the input vector and the entries of its reconstruction:

**Definition 3.** The mean squared error (MSE) between a fixed input $X$ and its estimator $Y_X$ is given by

$$\mathrm{MSE}_X(X, Y_X) = \frac{1}{N}||X - Y_X||_2^2,$$

where $|| \cdot ||_2$ denotes the euclidean norm.

The MSE is one of the most common measures to quantify the reconstruction error. A decreasing MSE denotes a high similarity between $X$ and $Y_X$ or, conversely, a low reconstruction error. In particular, we note that $X = Y_X$ if and only if the $\mathrm{MSE}_X(X, Y_X)$ equals zero. Nevertheless, while the MSE is widely used as an error function, it has unfavourable properties when assessing the reconstruction quality. Most importantly, it is not robust to scaling, hence, two reconstructions with the same $\mathrm{MSE} \neq 0$ could have a very different reconstruction quality. This drawback is well-known and to overcome this, the PSNR, which is based on the MSE, has been established as a standard metric in signal processing applications:

**Definition 4.** The peak signal-to-noise-ratio (PSNR) between a fixed input point $X$ and its estimator $Y_X$ is given as

$$\mathrm{PSNR}_X(X, Y_X) = 10 \cdot \log_{10}\left(\frac{(\max(X) - \min(X))^2}{\mathrm{MSE}(X, Y_X)}\right),$$

with

$$\max(X) = \max_{i \in \{1,...,N\}} x_i, \quad \text{and} \quad \min(X) = \min_{i \in \{1,...,N\}} x_i.$$

The PSNR contrasts the maximal range of the values of the entries of $X$ to the MSE between $X$ and its estimator $Y_X$. Thus, the PSNR puts the error measured in terms of the MSE into the context of the input value range of $X$. By that, the PSNR is robust to a linear scaling of the data and and can be better compared across different scenarios. In this regard, the PSNR is a superior indicator of the reconstruction quality, and, although it is based on the same measurement, has advantages over the MSE (Wang & Bovik, 2009). Notably, unlike MSE, the PSNR is no error but a reconstruction quality function where larger values correspond to higher fidelity (and lower error) between $X$ and $Y_X$.

Furthermore, we note that fixing the target input point $X$ enables formulating the MSE and the PSNR in terms of the randomness of the estimator $Y_X$. In particular, if we view $Y_X$ as the sample of a random variable, the MSE and the PSNR can be formulated as random variables and we can examine their tail behaviour (see Section 4.2).

Next, we introduce the NCC:

**Definition 5.** The normalised cross-correlation (NCC) (Rodgers & Nicewander, 1988) between a fixed input point $X$ and its estimator $Y_X$ is defined as

$$\mathrm{NCC}(X, Y_X) = \frac{\mathrm{Cov}(X, Y_X)}{\sigma_X \sigma_{Y_X}},$$

where

$$\text{Cov}(X, Y_X) = \frac{1}{N} \sum_{i=1}^{N} (x_i - \mathbb{E}_x[x])(y_{i,X} - \mathbb{E}_{y_X}[y_X]),$$

for $\mathbb{E}_x[x], \mathbb{E}_{y_X}[y_X]$ denoting the numerically obtained expected values within a sample. Moreover, $\sigma_X = \sqrt{\text{Var}(X)}$ and $\sigma_{Y_X} = \sqrt{\text{Var}(Y_X)}$, for $\text{Var}(X)$ and $\text{Var}(Y_X)$ being the sample variances of $X$ and $Y_X$, respectively.

We note that the NCC is equivalent to the Pearson's Correlation Coefficient, which is common in statistical testing. As both names suggest, this metric measures the linear correlation between a target point $X$ and its estimated counterpart $Y_X$ instead of measuring a difference. Hence, as opposed to metrics such as the MSE or PSNR, it is robust to linearly transformed inputs. High values of the NCC denote a high linear correlation between $X$ and $Y_X$, which, depending on the context, can imply a high similarity between $X$ and $Y_X$. For more information on interpreting the NCC results in the context of this work, we refer to Section 4.2.

## B Proofs

In the following, we give the proofs for our theoretical results.

**Proposition B.1.** *If the scaling factors $\beta_C(X)\frac{\partial g(f(X))}{\partial f(X)_1}, ..., \beta_C(X)\frac{\partial g(f(X))}{\partial f(X)_M}$ are unknown, then there is no "realisable" unbiased estimator of the target $X$ that can be constructed solely using the observed privatised gradients $\widetilde{\nabla}_{W_1}, ..., \widetilde{\nabla}_{W_M}$.*

*Proof.* First, we show *there is no deterministic transformation $T_X : \mathbb{R}^N \to \mathbb{R}^N$, such that $T_X\left(\widetilde{\nabla}_{W_j}\right)$, $j \in \{1, ..., M\}$, is a "realisable" unbiased estimator of $X$.*

Let $X$ be a fixed reconstruction target point and without loss of generality (w.l.o.g.) assume $X \neq \mathbf{0}_N$, for $\mathbf{0}_N$ the $N$-dimensional zero vector. Moreover, for a fixed $j \in \{1, ..., M\}$, let $T_X\left(\widetilde{\nabla}_{W_j}\right)$ be a realisable, unbiased estimator of $X$. Since

$$\widetilde{\nabla}_{W_j} = \beta_C(X)\frac{\partial g(f(X))}{\partial f(X)_j}X + \xi_j, \qquad \xi_j \sim \mathcal{N}(\mathbf{0}_N, C^2\sigma^2 I_N),$$

$T_X\left(\widetilde{\nabla}_{W_j}\right)$ must be an affine function, because that is the only transformation that can invert a multiplication and a sum. Hence, $T_X\left(\widetilde{\nabla}_{W_j}\right)$ has the following form:

$$T_X\left(\widetilde{\nabla}_{W_j}\right) = A\widetilde{\nabla}_{W_j} + b, \tag{30}$$

for $A \in \mathbb{R}^{N \times N}$ invertible and constant, and $b \in \mathbb{R}^{N \times N}$ constant. In particular, $A$ and $b$ are not functions of $X$, since $T_X\left(\widetilde{\nabla}_{W_j}\right)$ is a realisable estimator. Next, we compute the expectation of $T_X\left(\widetilde{\nabla}_{W_j}\right)$ for $X$ fixed:

$$\begin{aligned}
\mathbb{E}_X\left[T_X\left(\widetilde{\nabla}_{W_j}\right)\right] &= \mathbb{E}_X\left[T_X\left(\beta_C(X)\frac{\partial g(f(X))}{\partial f(X)_j}X + \xi_j\right)\right] \\
&= \mathbb{E}_X\left[A\beta_C(X)\frac{\partial g(f(X))}{\partial f(X)_j}X + A\xi_j + +b\right] \\
&= A\beta_C(X)\frac{\partial g(f(X))}{\partial f(X)_j}X + b.
\end{aligned}$$

$T_X\left(\widetilde{\nabla}_{W_j}\right)$ is an unbiased estimator of $X$ if and only if $\mathbb{E}_X\left[T_X\left(\widetilde{\nabla}_{W_j}\right)\right] = X$, which is equivalent to the following:

$$A\beta_C(X)\frac{\partial g(f(X))}{\partial f(X)_j}X + b = X \iff \left(\beta_C(X)\frac{\partial g(f(X))}{\partial f(X)_j}A - I_N\right)X + b = \mathbf{0}_N. \tag{31}$$

Since $b$ cannot depend on $X$ and $\left(\beta_C(X)\frac{\partial g(f(X))}{\partial f(X)_j}A - I_N\right)X$ is a linear transformation of $X$, we cannot set $\left(\beta_C(X)\frac{\partial g(f(X))}{\partial f(X)_j}A - I_N\right)X = -b$ and solve the equation. Therefore, the equations in 31 are satisfied if and only if

$$\left(\beta_C(X)\frac{\partial g(f(X))}{\partial f(X)_j}A - I_N\right)X = \mathbf{0}_N \quad \text{and} \quad b = \mathbf{0}_N. \tag{32}$$

Since $X \neq \mathbf{0}$, it is easy to see that $A\beta_C(X)\frac{\partial g(f(X))}{\partial f(X)_j}X + b = X$ holds if and only if

$$A = \frac{1}{\beta_C(X)\frac{\partial g(f(X))}{\partial f(X)_j}}I_N \quad \text{and} \quad b = \mathbf{0}_N,$$

implying that $A$ depends on $\beta_C(X)\frac{\partial g(f(X))}{\partial f(X)_j}$, a quantity that can only be computed knowing $X$ and contradicting the assumption that $T_X\left(\widetilde{\nabla}_{W_j}\right)$ is a realisable estimator. Therefore, for any $j \in \{1,...,M\}$, there is no realisable unbiased estimator of $X$ that can be constructed using the observed privatised gradient $\widetilde{\nabla}_{W_j}$.

It is easy to see that, for this setting, if there was an unbiased estimator of $X$ that can be constructed with $\widetilde{\nabla}_{W_1},...,\widetilde{\nabla}_{W_M}$, then there would exists at least one $j \in \{1,...,M\}$ such that an unbiased estimator of $X$ can be constructed with $\widetilde{\nabla}_{W_j}$. However, since there is no $j \in \{1,...,M\}$ such that an unbiased estimator of $X$ can be constructed with $\widetilde{\nabla}_{W_j}$, it follows by contraposition, that there is no unbiased estimator of $X$ than can be constructed with $\widetilde{\nabla}_{W_1},...,\widetilde{\nabla}_{W_M}$. $\qquad\square$

**Proposition B.2.** *The coordinte-wise variance of sample average $\hat{X}_M$ stated in 16 is lower bounded by $\sigma^2\|X\|_2^2$ for $M \to \infty$ and for all $X \in \mathcal{D} \setminus \{\boldsymbol{0}_N\}$.*

*Proof.* Let the iteration step be fixed and observe the sample mean $\hat{X}_M$ given in 16 with distribution described in 17. W.l.o.g., we assume $X \neq \boldsymbol{0}_N$ and that $\frac{\partial g(f(X))}{\partial f(X)_j} > 0$ for all $j \in \{1,...,M\}$. Then, consider the $i$th entry of $\hat{X}_M$, $i \in \{1,...,N\}$, particularly, its variance given by

$$\text{Var}(\hat{X}_{M,i}) = \frac{1}{M^2}\sum_{j=1}^{M}\frac{C^2\sigma^2}{\beta_C(X)^2\frac{\partial g(f(X))}{\partial f(X)_j}^2}. \tag{33}$$

It is easy to see that 33 decreases for decreasing $\sum_{j=1}^{M}\frac{1}{\beta_C(X)^2\frac{\partial g(f(X))}{\partial f(X)_j}^2}$. Therefore, minimising

$$\sum_{j=1}^{M}\frac{1}{\beta_C(X)^2\frac{\partial g(f(X))}{\partial f(X)_j}^2} \tag{34}$$

with respect to $\frac{\partial g(f(X))}{\partial f(X)_1},...,\frac{\partial g(f(X))}{\partial f(X)_M}$ minimises the variance in 33. Note that doing so does not affect the multiplicative term $C^2\sigma^2/M$. By definition of the global concatenated gradient $G_X$ (see 9), the squared norm of $G_X$ is given by

$$\|G_X\|_2^2 = \sum_{j=1}^{M}\frac{\partial g(f(X))}{\partial f(X)_j}^2\|X\|_2^2 + \sum_{j=1}^{M}\frac{\partial g(f(X))}{\partial b_j}^2 + \|G_{X,P}\|_2^2. \tag{35}$$

Set $\|\text{Rest}\|_2^2 := \sum_{j=1}^{M}\frac{\partial g(f(X))}{\partial b_j}^2 + \|G_{X,P}\|_2^2$. Then, we can reformulate 35 and obtain the following constraint regarding $\frac{\partial g(f(X))}{\partial f(X)_1},...,\frac{\partial g(f(X))}{\partial f(X)_M}$:

$$\sum_{j=1}^{M}\frac{\partial g(f(X))}{\partial f(X)_j}^2 = \frac{\|G_X\|_2^2 - \|\text{Rest}\|_2^2}{\|X\|_2^2}. \tag{36}$$

Minimising the variance in 33 with respect to $\frac{\partial g(f(X))}{\partial f(X)_1},...,\frac{\partial g(f(X))}{\partial f(X)_M}$ under the constraint given in 36, does not affect the norm of the global gradient $G_X$ and, thus, it does not affect the value of $\beta_C(X)$. Therefore, minimising 34 with respect to $\frac{\partial g(f(X))}{\partial f(X)_1},...,\frac{\partial g(f(X))}{\partial f(X)_M}$ under the constraint given in 36 is equivalent to minimising $\sum_{j=1}^{M}\frac{1}{\frac{\partial g(f(X))}{\partial f(X)_j}^2}$ with respect to $\frac{\partial g(f(X))}{\partial f(X)_1},...,\frac{\partial g(f(X))}{\partial f(X)_M}$ under 36. Hence, setting $y_j = \frac{\partial g(f(X))}{\partial f(X)_j}^2$, for $j \in \{1,...,M\}$, we have an optimisation problem of the following form:

$$\text{Minimise } \sum_{j=1}^{M}\frac{1}{y_j} \text{ for } \sum_{j=1}^{M}y_j = \frac{\|G_X\|_2^2 - \|\text{Rest}\|_2^2}{\|X\|_2^2} \text{ and } y_1,...,y_M > 0, \tag{37}$$

37 is a well-known minimisation problem with solution given by $y_j = \frac{\|G_X\|_2^2 - \|\text{Rest}\|_2^2}{M\|X\|_2^2}$ for all $j \in \{1, ..., M\}$. However, if needed, a proof of the statement can be obtained using the gradient of the function in 37 to construct the direction of the steepest descent and combining this with the given constraints in 37. Hence, setting

$$\frac{\partial g(f(X))}{\partial f(X)_j}^2 = \frac{\|G_X\|_2^2 - \|\text{Rest}\|_2^2}{M\|X\|_2^2} \tag{38}$$

for all $j \in \{1, ..., M\}$, minimises the variance given in 33 with respect to $\frac{\partial g(f(X))}{\partial f(X)_1}, ..., \frac{\partial g(f(X))}{\partial f(X)_M}$.

We insert the choice 38 into the variance 39 and obtain

$$\text{Var}(\hat{X}_{M,i}) \geq \frac{1}{M^2} \sum_{j=1}^{M} \frac{C^2 \sigma^2 M \|X\|_2^2}{\beta_C(X)^2 (\|G_X\|_2^2 - \|\text{Rest}\|_2^2)} = \frac{C^2 \sigma^2 \|X\|_2^2}{\beta_C(X)^2 (\|G_X\|_2^2 - \|\text{Rest}\|_2^2)}, \tag{39}$$

for $i \in \{1, ..., N\}$.

Recall the definition of the clipping term $\beta_C(X)$ given in 10. Using 38, we can see that the norm of the global gradient $\|G_X\|_2$ is linearly increasing in $M$. Thus, there exist $\hat{M}$, such that for all $M \geq \hat{M}$, $\|G_X\|_2 \geq C$ and $\beta_C(X) = \frac{C}{\|G_X\|_2}$. Hence, by 39

$$\lim_{M \to \infty} \text{Var}(\hat{X}_{M,i}) \geq C^2 \sigma^2 \|X\|_2^2 \cdot \lim_{M \to \infty} \frac{1}{\beta_C(X)^2 (\|G_X\|_2^2 - \|\text{Rest}\|_2^2)}$$

$$= C^2 \sigma^2 \|X\|_2^2 \cdot \lim_{M \to \infty} \frac{1}{C^2 - \frac{\|\text{Rest}\|_2^2}{\|G_X\|_2^2}}$$

$$= \sigma^2 \|X\|_2^2. \tag{40}$$

Equality 40 holds because $\|\text{Rest}\|_2$ is independent of $M$. Lastly, if $X \neq \mathbf{0}_N$, then $\sigma^2 \|X\|_2^2 > 0$. □

**Proposition 1.** *Let the part of neural network given by $g$ be replaced by the loss function $\mathcal{L} : \mathbb{R}^M \to \mathbb{R}$ with $\mathcal{L}(f(X)) = \mathbf{1}_M^T f(X)$, where $\mathbf{1}_M$ is the M-dimensional 1-vector, and*

$$M \geq \max\left(1, \left\lceil \frac{C}{\min_{X \in \mathcal{D} \setminus \{\mathbf{0}_N\}} \|X\|_2} \right\rceil\right), \tag{18}$$

*where $\lceil \cdot \rceil$ denotes the function that rounds up its argument to the nearest integer, then $\frac{1}{M^2} \sum_{j=1}^{M} \frac{C^2 \sigma^2}{\beta_C(X)^2 \frac{\partial g(f(X))}{\partial f(X)_j}^2}$ is minimal and takes the value $\sigma^2 \|X\|_2^2$.*

*Proof.* Let the iteration step be fixed and observe the sample mean $\hat{X}_M$ given in 16 with distribution described in 17. Then, consider the $i$th entry of $\hat{X}_M$, particularly, its variance given by

$$\text{Var}(\hat{X}_{M,i}) = \frac{1}{M^2} \sum_{j=1}^{M} \frac{C^2 \sigma^2}{\beta_C(X)^2 \frac{\partial g(f(X))}{\partial f(X)_j}^2}, \tag{41}$$

for $i \in \{1, ..., N\}$. Without loss of generality, let $X \neq \mathbf{0}_N$. We have shown in the proof of Proposition B.1 that the choice

$$\frac{\partial g(f(X))}{\partial f(X)_j}^2 = \frac{\|G_X\|_2^2 - \|\text{Rest}\|_2^2}{M\|X\|_2^2} \tag{42}$$

for all $j \in \{1, ..., M\}$, minimises the variance given in 41 with respect to $\frac{\partial g(f(X))}{\partial f(X)_1}, ..., \frac{\partial g(f(X))}{\partial f(X)_M}$. Let us set

$$\frac{\partial g(f(X))}{\partial f(X)_1}^2 = \frac{\|G_X\|_2^2 - \|\text{Rest}\|_2^2}{M\|X\|_2^2}, \tag{43}$$

and insert this choice into the variance 41:

$$\text{Var}(\hat{X}_{M,i}) \geq \frac{C^2\sigma^2}{M} \frac{1}{\beta_C(X)^2 \frac{\partial g(f(X))}{\partial f(X)_1}^2}, \tag{44}$$

for $i \in \{1, ..., N\}$. If $\|G_X\|_2$ is fixed, it follows from 42 that $\frac{\partial g(f(X))}{\partial f(X)_1}^2$ increases with decreasing norm $\|\text{Rest}\|_2$. However, $\|\text{Rest}\|_2$ cannot be bounded or quantified for any iteration step without specific knowledge of the neural network. Thus, the adversary cannot minimise $\|\text{Rest}\|_2$ without manipulating some layers of the network. If they manipulate these layers, we see 44 is minimal whenever $\frac{\partial g(f(X))}{\partial f(X)_1}$ is maximal, i.e., whenever $\|\text{Rest}\|_2 = 0$. $\|\text{Rest}\|_2 = 0$ occurs for all $X \in \mathcal{D}$ and all iteration steps when the adversary replaces the entire network by the linear layer $f$ (see Section 3.2, specifically Equation 4) and sets the bias term $b$ to be equal to $\mathbf{0}_M$. In such a case the neural network is given by the linear layer $f(X) = WX$ and a loss function which we denote by $\mathcal{L} : \mathbb{R}^M \to \mathbb{R}$. As a consequence, $\frac{\partial g(f(X))}{\partial f(X)_1} = \frac{\partial \mathcal{L}(f(X))}{\partial f(X)_1}$. In particular, 42 implies

$$\frac{\partial g(f(X))}{\partial f(X)_1}^2 = \frac{\partial \mathcal{L}(f(X))}{\partial f(X)_1}^2 = \frac{\|G_X\|_2^2}{M\|X\|_2^2}. \tag{45}$$

Inserting 45 into the right hand side of 44 further bounds the variance $\text{Var}(\hat{X}_{M,i})$:

$$\text{Var}(\hat{X}_{M,i}) \geq \frac{C^2\sigma^2\|X\|_2^2}{\beta_C(X)^2\|G_X\|_2^2}, \tag{46}$$

for all $i \in \{1, ..., N\}$.

Now, we observe the lower bound in 46. Naturally, the right hand side of 46 is lowest when the denominator in 46 is highest. By definition of the clipping term $\beta_C(X)$ (see 10), the product $\beta_C(X)^2\|G_X\|_2^2$ is upper bounded by $C^2$, delivering

$$\text{Var}(\hat{X}_{M,i}) \geq \sigma^2\|X\|_2^2, \tag{47}$$

for $i \in \{1, ..., N\}$. In particular, no change in the parameters or architecture of the network can increase the product $\beta_C(X)^2\|G_X\|_2^2$ beyond $C^2$ to further decrease the lower bound given in 47. Therefore, we assume, the adversary chooses $M$ and $\frac{\partial \mathcal{L}(f(X))}{\partial f(X)_1}$ such that that $\|G_X\|_2^2 \geq C^2$ for as many data points $X$ as possible. Using 45, $\|G_X\|_2^2 \geq C^2$ implies

$$M\frac{\partial \mathcal{L}(f(X))}{\partial f(X)_1}^2 \geq \frac{C^2}{\|X\|_2^2}. \tag{48}$$

Next, we consider two cases, when $\min_{X\in\mathcal{D}}\|X\|_2 > 0$ and $\min_{X\in\mathcal{D}}\|X\|_2 = 0$. If $\min_{X\in\mathcal{D}}\|X\|_2 > 0$, then

$$\|G_X\|_2^2 \geq C^2 \quad \forall X \in \mathcal{D} \iff M\frac{\partial \mathcal{L}(f(X))}{\partial f(X)_1}^2 \geq \frac{C^2}{\min_{X\in\mathcal{D}}\|X\|_2^2} \quad \forall X \in \mathcal{D}. \tag{49}$$

$M$, $\min_{X\in\mathcal{D}}\|X\|_2$ and $C$ fixed during training and do not changed from iteration to iteration. Thus, 49 holds for all $X \in \mathcal{D}$ if $\frac{\partial \mathcal{L}(f(X))}{\partial f(X)_1}$ is constant for all $X$ and all iteration steps, implying $\mathcal{L}$ is an affine function of $f(X)$. If $C > \min_{X\in\mathcal{D}}\|X\|_2$, then choosing $M \geq \left\lceil \frac{C}{\min_{X\in\mathcal{D}}\|X\|_2} \right\rceil$ and the loss function $\mathcal{L} : \mathbb{R}^M \to \mathbb{R}$ to be $\mathcal{L}(f(X)) = \mathbf{1}_M^T f(X)$, where $\mathbf{1}_M$ is the $M$-dimensional 1-vector, delivers the sample average $\hat{X}_M$ with the lowest variance per entry given by

$$\text{Var}(\hat{X}_{M,i}) = \sigma^2\|X\|_2^2, \tag{50}$$

where $\lceil \cdot \rceil$ denotes the function that rounds up its argument to the nearest integer. If $C \leq \min_{X\in\mathcal{D}}\|X\|_2$, then choosing $M \geq 1$ and the loss function to be $\mathcal{L}(f(X)) = f(X)$ delivers the sample average $\hat{X}_M$ with the lowest variance per entry given by

$$\text{Var}(\hat{X}_{M,i}) = \sigma^2\|X\|_2^2 \geq \sigma^2 C^2. \tag{51}$$

All in all, we conclude that if $\min_{X \in \mathcal{D}} \|X\|_2 > 0$, replacing the subpart of the neural network given by $g$ by the loss function $\mathcal{L}(f(X)) = \mathbf{1}_M^T f(X)$ and setting $M \geq \max\left(1, \left\lceil \frac{C}{\min_{X \in \mathcal{D}} \|X\|_2} \right\rceil \right)$ minimises the variance $\mathrm{Var}(\hat{X}_{M,i})$ for all $i \in \{1, ..., N\}$.

If $\min_{X \in \mathcal{D}} \|X\|_2 = 0$, then there is no choice for $\frac{\partial \mathcal{L}(f((X))}{\partial f(X)_1}$ or $M$ such that 48 holds for all $X \in \mathcal{D}$. However, in such a case, without loss of generality, we assume that the adversary sets the loss function to be $\mathcal{L}(f(X)) = f(X)$ and chooses $M$ to ensure that 48 holds for all $X \in \mathcal{D}$ with $X \neq \mathbf{0}_N$. In such a case, the adversary sets $M = \left\lceil \frac{C}{\min_{X \in \mathcal{D} \setminus \{\mathbf{0}_N\}} \|X\|_2} \right\rceil$, analogously as the argumentation above. □

The following lemma serves as an auxiliary result to obtain Proposition 2:

**Lemma B.1.** *For all $j \in \{1, ..., N\}$, the $j$th entry $\hat{x}_j$ of the estimator $\hat{X}$ is a (fully) efficient estimator for the $j$th entry $x_j$ of the target $X$.*

*Proof.* The estimator $\hat{X}$ given in 20 is normally distributed with mean $X$ and covariance matrix given by $\sigma^2 \|X\|_2^2 I_N$. Let $\hat{x}_i$, $i \in \{1, ..., N\}$, denote the $i$th entry of $\hat{X}$. Then, by distribution of $\hat{X}$, $\hat{x}_1, ..., \hat{x}_N$ are independent, normally distributed with mean $x_1, ..., x_N$, respectively, and same variance given by $\sigma^2 \|X\|_2^2$. Thus, for all $i \in \{1, ..., N\}$, $\hat{x}_i$ is an unbiased estimator of the $i$th entry of the target $X$.

Moreover, applying the Cramér-Rao bound for scalar unbiased estimators, we compute a lower bound for the variance of any the estimator of $\hat{x}_i$, $i \in \{1, ..., N\}$:

$$\mathrm{Var}_X(\hat{x}_i) \geq I(x_i)^{-1} = \sigma^2 \|X\|_2^2, \tag{52}$$

where $I(x_i)$ denotes the Fisher information matrix that measures the amount of information the rescaled, observable normally distributed random variables $\hat{x}_i$ carries about its unknown mean $x_i$. Since this matrix is well-known in literature, we do not provide a proof for the right hand side of the equality in 52.

Since for all $i \in \{1, ..., N\}$, $\hat{x}_i$ is an unbiased estimator of $x_i$ that achieves the Cramér-Rao bound, it is a *(fully) efficient* estimator of $x_i$ achieving the smallest variability in terms of the variance. □

**Proposition 2.** *$\hat{X}$ is the minimum variance unbiased estimator (MVUE) for $X$. Moreover, the expected mean squared error between the target $X$ and $\hat{X}$ is given by:*

$$\mathbb{E}_X[\mathrm{MSE}_X(X, \hat{X})] = \sigma^2 \|X\|_2^2. \tag{21}$$

*Proof.* Let $\hat{Y} = (\hat{y}_1, ..., \hat{y}_N)^T$ denote any estimator of $X$. Then,

$$\mathbb{E}_X[\mathrm{MSE}_X(X, \hat{Y})] = \mathbb{E}_X \left[ \frac{\|X - \hat{Y}\|_2^2}{N} \right]$$

$$= \frac{1}{N} \sum_{i=1}^N \mathbb{E}_X \left[ (x_i - \hat{y}_i)^2 \right]$$

$$= \frac{1}{N} \sum_{i=1}^N \mathbb{E}_{x_i} \left[ (x_i - \hat{y}_i)^2 \right]$$

$$= \frac{1}{N} \sum_{i=1}^N \left( \mathbb{E}_{x_i}[x_i - \hat{y}_i]^2 + \mathrm{Var}_{x_i}(x_i - \hat{y}_i) \right)$$

$$= \frac{1}{N} \sum_{i=1}^N \left( \mathrm{Bias}_{x_i}(x_i, \hat{y}_i)^2 + \mathrm{Var}_{x_i}(\hat{y}_i) \right). \tag{53}$$

For all unbiased estimators, the expected MSE, as given in 53, is solely determined by the sum of the variances of each entry $\hat{y}_i$. Therefore, by Lemma B.1, $\hat{X}$ is the unbiased estimator that minimises 53. In other words,

$\hat{X}$ is the unbiased estimator that achieves the lowest expected MSE. Such estimators are called *minimum variance unbiased estimators* in the literature and achieve the smallest variability in terms of the variance.

Lastly, we compute the expected MSE between $X$ and $\hat{X}$:

$$\mathbb{E}_X[\text{MSE}_X(X, \hat{X})] = \frac{1}{N} \sum_{i=1}^{N} \text{Var}_{x_i}(\hat{x}_i) = \sigma^2 \|X\|_2^2.$$

$\square$

**Theorem 1.** *Using the* MSE *as an optimality criterion, $\hat{X}$ is the best achievable estimator and, thus, reconstruction for the target point $X$.*

*Proof.* Since the adversary only observes one privatised version of the gradient $\tilde{\nabla}_W$ they can use to construct an estimator for $X$, it is easy to see that $\hat{X}$ is a sufficient statistic for estimating $X$. Moreover, by Proposition 2, $\hat{X}$ is the unbiased estimator, which uses the sufficient statistic $\tilde{\nabla}_W$ as input, that achieves the lowest expected $\text{MSE}_X(X, \hat{X})$. Such unbiased estimators achieve the lowest possible MSE and have the smallest variability in terms of their variance. Thus, using the MSE as an optimality criterion, $\hat{X}$ is the optimal estimator for $X$. Since lower values of $\text{MSE}_X(X, \hat{X})$ denote high similarity between $X$ and $\hat{X}$, we conclude that $\hat{X}$ is the best achievable reconstruction for $X$. $\square$

**Theorem 2.**

$$\text{MSE}_X(X, \hat{X}) \stackrel{d}{=} \frac{\sigma^2 \|X\|_2^2}{N} \cdot Y \quad with \quad Y \sim \chi_N^2, \tag{22}$$

*where $\chi_N^2$ denotes the central chi-squared distribution with $N$ degrees of freedom. In particular, for $\eta$ given,*

$$\mathbb{P}_{\hat{X}}(\text{MSE}_X(X, \hat{X}) \le \eta) = \Gamma_R \left( \frac{N}{2}, \frac{N\eta}{2\sigma^2 \|X\|_2^2} \right), \tag{23}$$

*where $\Gamma_R$ is the regularised gamma function.*

*Proof.* We can compute the MSE between $X$ and its reconstruction $\hat{X}$ as the mean error over their components:

$$\begin{aligned}
\text{MSE}_X(X, \hat{X}) &= \frac{1}{N} \sum_{i=1}^{N} (x_i - \hat{x}_i)^2 \\
&= \frac{1}{N} \sum_{i=1}^{N} \tilde{\xi}_i^{\,2}, \qquad \tilde{\xi}_i \sim \mathcal{N}(0, \sigma^2 \|X\|_2^2) \\
&= \frac{1}{N} \sum_{i=1}^{N} \sigma^2 \|X\|_2^2 \left( \frac{1}{\sigma \|X\|_2} \tilde{\xi}_i \right)^2 \\
&= \frac{\sigma^2 \|X\|_2^2}{N} \sum_{i=1}^{N} \rho_i^2,
\end{aligned} \tag{54}$$

where $\rho_i := \frac{1}{\sigma \|X\|_2} \xi_i$, for $i \in \{1, ..., N\}$. If $X$ is fixed, $\rho_1, ..., \rho_N$ are pairwise independent random variables with $\rho_i \sim \mathcal{N}(0, 1)$ for all $i \in \{1, ..., N\}$. Hence,

$$\sum_{i=1}^{N} \rho_i^2 \sim \chi_N^2, \tag{55}$$

where $\chi_N^2$ denotes the central chi-squared distribution with $N$ degrees of freedom. Thus,

$$\text{MSE}_X(X, \hat{X}) \stackrel{d}{=} \frac{\sigma^2 \|X\|_2^2}{N} \cdot Y \quad with \quad Y \sim \chi_N^2.$$

Then, for $\eta$ given

$$\mathbb{P}_{\hat{X}}\left(\mathrm{MSE}_X(X,\hat{X}) \leq \eta\right) = \gamma \iff \mathbb{P}_Y\left(\frac{\sigma^2\|X\|_2^2}{N} \cdot Y \leq \eta\right) = \gamma.$$

Since $Y$ is a centered chi-squared distributed random variable with $N$ degrees of freedom, its cumulative distribution function can be computed via the regularized gamma function $\Gamma_R$. Hence,

$$\mathbb{P}_Y\left(\frac{\sigma^2\|X\|_2^2}{N} \cdot Y \leq \eta\right) = P_Y\left(Y \leq \frac{N\eta}{\sigma^2\|X\|_2^2}\right) = \Gamma_R\left(\frac{N}{2}, \frac{N\eta}{2\sigma^2\|X\|_2^2}\right)$$

implies

$$\mathbb{P}_{\hat{X}}\left(\mathrm{MSE}_X(X,\hat{X}) \leq \eta\right) = \Gamma_R\left(\frac{N}{2}, \frac{N\eta}{2\sigma^2\|X\|_2^2}\right). \tag{56}$$

$\square$

**Proposition B.3.** *For $N \to \infty$, it holds*

$$\sqrt{N}\left(\mathrm{MSE}_X(X,\hat{X}) - \sigma^2\|X\|_2^2\right) \xrightarrow{d} \mathcal{N}\left(0, 2\sigma^4\|X\|_2^4\right). \tag{57}$$

*Proof.* By Theorem 2, it holds

$$\mathrm{MSE}_X(X,\hat{X}) = \sigma^2\|X\|_2^2 \frac{1}{N}\sum_{i=1}^N \rho_i^2 = \frac{1}{N}\sum_{i=1}^N \sigma^2\|X\|_2^2\rho_i^2, \tag{58}$$

where $\rho_i$ are i.i.d. standard normally distributed random variables, for all $i \in \{1,...,N\}$. Consider the behaviour of $\sigma^2\|X\|_2^2\rho_i^2$ for $i \in \{1,...,N\}$:

$$\mathbb{E}\left[\sigma^2\|X\|_2^2\rho_i^2\right] = \sigma^2\|X\|_2^2\mathbb{E}\left[\rho_i^2\right] = \sigma^2\|X\|_2^2\mathrm{Var}(\rho_i) = \sigma^2\|X\|_2^2,$$

and

$$\mathrm{Var}\left(\sigma^2\|X\|_2^2\rho_i^2\right) = \sigma^4\|X\|_2^4\mathrm{Var}\left(\rho_i^2\right) = \sigma^4\|X\|_2^4\left(\mathbb{E}\left[\rho_i^4\right] - \mathbb{E}\left[\rho_i^2\right]^2\right) = \sigma^4\|X\|_2^4(3-1) = 2\sigma^4\|X\|_2^4,$$

since $\mathbb{E}\left[\rho_i^4\right]$ corresponds to the 4-th moment of the standard normal distribution and can be, thus, computed easily. Therefore, using the central limit theorem, for $N \to \infty$, it follows

$$\sqrt{N}\left(\mathrm{MSE}_X(X,\hat{X}) - \sigma^2\|X\|_2^2\right) = \sqrt{N}\left(\frac{1}{N}\sum_{i=1}^N \sigma^2\|X\|_2^2\rho_i^2 - \sigma^2\|X\|_2^2\right) \xrightarrow{d} \mathcal{N}\left(0, 2\sigma^4\|X\|_2^4\right).$$

$\square$

**Proposition 3.** *Let $\eta$ given. Then, for all $X \in \mathcal{D} \setminus \{\mathbf{0}_N\}$,*

$$\mathbb{P}_{\hat{X}}\left(\mathrm{MSE}_X\left(X,\hat{X}\right) \leq \eta\right) \leq \Gamma_R\left(\frac{N}{2}, \frac{N\eta}{2\sigma^2\min_{X \in \mathcal{D}}\|X\|_2^2}\right), \tag{24}$$

*where $\Gamma_R$ is the regularised gamma function. Moreover, the DP-mechanism $\mathcal{M}$ is $(\eta, \gamma(\eta))$-reconstruction robust with respect to the MSE for any reconstruction and $\gamma(\eta) = \Gamma_R\left(\frac{N}{2}, \frac{N\eta}{2\sigma^2\min_{X \in \mathcal{D}}\|X\|_2^2}\right)$. If $C \leq \min_{X \in \mathcal{X}}\|X\|_2$ holds, then $\mathcal{M}$ is $(\eta, \gamma'(\eta))$-reconstruction robust with respect to the MSE for any reconstruction and $\gamma'(\eta) = \Gamma_R\left(\frac{N}{2}, \frac{N\eta}{2\sigma^2C^2}\right)$.*

*Proof.* Consider the CDF of the $\mathrm{MSE}_X(X, \hat{X})$ for $X \in \mathcal{D} \setminus \{\mathbf{0}_N\}$ given Equation 23 in Theorem 2. Since the regularised gamma function is increasing in its second argument, using Theorem 2, it follows that

$$\mathbb{P}_{\hat{X}}\left(\mathrm{MSE}_X(X, \hat{X}) \leq \eta\right) = \Gamma_R\left(\frac{N}{2}, \frac{N\eta}{2\sigma^2 \|X\|_2^2}\right) \leq \Gamma_R\left(\frac{N}{2}, \frac{N\eta}{2\sigma^2 \min_{X \in \mathcal{X}} \|X\|_2^2}\right), \tag{59}$$

for all reconstruction target points $X \in \mathcal{D} \setminus \{\mathbf{0}_N\}$. Moreover, due to the optimality of the estimator $\hat{X}$, it holds

$$\mathbb{P}_{Y_X'}\left(\mathrm{MSE}_X(X, Y_X') \leq \eta\right) \leq \mathbb{P}_{\hat{X}}\left(\mathrm{MSE}_X(X, \hat{X}) \leq \eta\right) \tag{60}$$

for all possible reconstructions $Y_X'$. Therefore, combining 59 and 60, we conclude that $\mathcal{M}$ is $(\eta, \gamma(\eta))$-reconstruction robust with respect to the MSE for any reconstruction and $\gamma(\eta) = \Gamma_R\left(\frac{N}{2}, \frac{N\eta}{2\sigma^2 \min_{X \in \mathcal{X}} \|X\|_2^2}\right)$.

Moreover, if $C \leq \min_{X \in \mathcal{X}} \|X\|_2$, then

$$\mathbb{P}_{\hat{X}}\left(\mathrm{MSE}_X(X, \hat{X}) \leq \eta\right) \leq \Gamma_R\left(\frac{N}{2}, \frac{N\eta}{2\sigma^2 C^2}\right), \tag{61}$$

for all $X \in \mathcal{D} \setminus \{\mathbf{0}_N\}$. In particular, in such a case, $\mathcal{M}$ is $(\eta, \gamma'(\eta))$-reconstruction robust with respect to the MSE for any reconstruction and $\gamma'(\eta) = \Gamma_R\left(\frac{N}{2}, \frac{N\eta}{2\sigma^2 C^2}\right)$. $\square$

**Proposition 4.** *Assume* $\max_{X \in \mathcal{D}} \max(X)$ *and* $\min_{X \in \mathcal{D}} \min(X)$ *are known quantities. Then, for all* $X \in \mathcal{D} \setminus \{\mathbf{0}_N\}$,

$$\mathbb{P}_{\hat{X}}\left(\mathrm{PSNR}_X\left(X, \hat{X}\right) \geq \eta\right) \leq \Gamma_R\left(\frac{N}{2}, \frac{N\tilde{\eta}(\eta)}{2\sigma^2 \min_{X \in \mathcal{D} \setminus \{\mathbf{0}_N\}} \|X\|_2^2}\right), \tag{25}$$

*for* $\tilde{\eta}(\eta) := 10^{-\frac{\eta}{10}}\left(\max_{X \in \mathcal{D}} \max(X) - \min_{X \in \mathcal{D}} \min(X)\right)^2$, *and for* $\Gamma_R$ *being the regularised gamma function. If* $C \leq \min_{X \in \mathcal{D}} \|X\|_2$ *holds, then*

$$\mathbb{P}_{\hat{X}}\left(\mathrm{PSNR}_X\left(X, \hat{X}\right) \geq \eta\right) \leq \Gamma_R\left(\frac{N}{2}, \frac{N\tilde{\eta}(\eta)}{2\sigma^2 C^2}\right), \tag{26}$$

*for all* $X \in \mathcal{D} \setminus \{\mathbf{0}_N\}$.

*Proof.* The cumulative distribution function (CDF) of the PSNR can be calculated using the CDF of the MSE. In particular, this implies that we can also compute probabilistic bounds for the PSNR using Theorem 2. Let $\eta$ be given. Then,

$$\mathrm{PSNR}_X(X, \hat{X}) \geq \eta$$

$$\iff \qquad 10 \log_{10}\left(\frac{(\max(X) - \min(X))^2}{\mathrm{MSE}(X, \hat{X})}\right) \geq \eta$$

$$\iff \log_{10}((\max(X) - \min(X))^2) - \log_{10}(\mathrm{MSE}(X, \hat{X})) \geq \frac{\eta}{10}$$

$$\iff \qquad 2\log_{10}[(\max(X) - \min(X))] - \frac{\eta}{10} \geq \log_{10}\left(\mathrm{MSE}(X, \hat{X})\right)$$

$$\iff \qquad (\max(X) - \min(X))^2 10^{-\frac{\eta}{10}} \geq \mathrm{MSE}(X, \hat{X}).$$

Thus, setting

$$\hat{\eta}(\eta) = 10^{-\frac{\eta}{10}}(\max(X) - \min(X))^2, \tag{62}$$

it follows from Theorem 2 that

$$\mathbb{P}_{\hat{X}}(\mathrm{PSNR}_X(X, \hat{X}) \geq \eta)$$

$$= \mathbb{P}_{\hat{X}}(\mathrm{MSE}_X(X, \hat{X}) \leq \hat{\eta}(\eta)) \leq \Gamma_R\left(\frac{N}{2}, \frac{N\hat{\eta}(\eta)}{2} \frac{1}{\sigma^2 \min_{X \in \mathcal{D} \setminus \{\mathbf{0}_N\}} \|X\|_2^2}\right). \tag{63}$$

We note that the right hand side of the previous result is still dependent on the target value $X$ due to $\hat{\eta}(\eta)$. To remove this dependency, we find an upper bound for $\hat{\eta}(\eta)$:

$$\hat{\eta}(\eta) = 10^{-\frac{\eta}{10}} (\max(X) - \min(X))^2 \leq 10^{-\frac{\eta}{10}} \left( \max_{X \in \mathcal{D}} \max(X) - \min_{X \in \mathcal{D}} \min(X) \right)^2 := \tilde{\eta}(\eta).$$

Since the regularised gamma function $\Gamma_R$ is increasing with respect to the second argument, it follows:

$$\mathbb{P}_{\hat{X}}(\mathrm{PSNR}_X(X, \hat{X}) \geq \eta) \leq \Gamma_R \left( \frac{N}{2}, \frac{N\tilde{\eta}(\eta)}{2\sigma^2 \min_{X \in \mathcal{D} \setminus \{\mathbf{0}_N\}} \|X\|_2^2} \right), \tag{64}$$

for all $X \in \mathcal{D} \setminus \{\mathbf{0}_N\}$. Using the same argument, if $C \leq \min_{X \in \mathcal{D} \setminus \{\mathbf{0}_N\}} \|X\|_2$, then

$$\mathbb{P}_{\hat{X}}(\mathrm{PSNR}_X(X, \hat{X}) \geq \eta) \leq \Gamma_R \left( \frac{N}{2}, \frac{N\tilde{\eta}(\eta)}{2\sigma^2 C^2} \right). \tag{65}$$

$\square$

**Proposition 5.** *Assume $\max_{X \in \mathcal{D}} \max(X)$ and $\min_{X \in \mathcal{D}} \min(X)$ are known quantities, and let $Y_X$ be any possible reconstruction. Then, for all $X \in \mathcal{D} \setminus \{\mathbf{0}_N\}$,*

$$\mathbb{P}_{Y_X} \left( \mathrm{PSNR}_X (X, Y_X) \geq \eta \right) \leq \Gamma_R \left( \frac{N}{2}, \frac{N\tilde{\eta}(\eta)}{2\sigma^2 \min_{X \in \mathcal{D} \setminus \{\mathbf{0}_N\}} \|X\|_2^2} \right), \tag{27}$$

*for $\Gamma_R$ being the regularised gamma function and $\tilde{\eta}(\eta)$ as defined in Proposition 4. In particular, the DP-mechanism $\mathcal{M}$ is $(-\eta, \tilde{\gamma}(\tilde{\eta}(\eta)))$-reconstruction robust with respect to the negative PSNR ($-$PSNR) for any analytic reconstruction and $\tilde{\gamma}(\tilde{\eta}(\eta)) = \Gamma_R \left( \frac{N}{2}, \frac{N\tilde{\eta}(\eta)}{2\sigma^2 \min_{X \in \mathcal{D} \setminus \{\mathbf{0}_N\}} \|X\|_2^2} \right)$. Moreover, if $C \leq \min_{X \in \mathcal{D}} \|X\|_2$ holds, then the DP-mechanism $\mathcal{M}$ is $(-\eta, \tilde{\gamma}'(\tilde{\eta}(\eta)))$-reconstruction robust with respect to $-$PSNR for any reconstruction, $\tilde{\gamma}'(\tilde{\eta}(\eta)) = \Gamma_R \left( \frac{N}{2}, \frac{N\tilde{\eta}(\eta)}{2\sigma^2 C^2} \right)$.*

*Proof.* On the one hand, by the optimality of the reconstruction $\hat{X}$ with respect to the PSNR, it holds that

$$\mathbb{P}_{Y_X}(\mathrm{PSNR}_X(X, Y_X') \geq \eta) \leq \mathbb{P}_{\hat{X}}(\mathrm{PSNR}_X(X, \hat{X}) \geq \eta),$$

for any analytic reconstruction $Y_X'$. On the other hand, by Proposition 4,

$$\mathbb{P}_{\hat{X}}(\mathrm{PSNR}_X(X, \hat{X}) \geq \eta) = \mathbb{P}_{\hat{X}}(-\mathrm{PSNR}_X(X, \hat{X}) \leq -\eta) \leq \Gamma_R \left( \frac{N}{2}, \frac{N\tilde{\eta}(\eta)}{2\sigma^2 \min_{X \in \mathcal{D} \setminus \{\mathbf{0}_N\}} \|X\|_2^2} \right),$$

for $\tilde{\eta}(\eta)$ as defined in Proposition 4 and $\Gamma_R$ being the regularised gamma function. Therefore, using Definition 2, we conclude that the DP mechanism $\mathcal{M}$ is $(-\eta, \tilde{\gamma}(\tilde{\eta}(\eta)))$-reconstruction robust for $\tilde{\gamma}(\tilde{\eta}(\eta)) = \Gamma_R \left( \frac{N}{2}, \frac{N\tilde{\eta}(\eta)}{2\sigma^2 \min_{X \in \mathcal{D} \setminus \{\mathbf{0}_N\}} \|X\|_2^2} \right)$ with respect to the negative PSNR, i.e., $-$PSNR.

Lastly, since the regularised gamma function $\Gamma_R$ is increasing with respect to the second argument, if $C \leq \min_{X \in \mathcal{D}} \|X\|_2$, then the DP mechanism $\mathcal{M}$ is $(-\eta, \tilde{\gamma}'(\tilde{\eta}(\eta)))$-reconstruction robust for $\tilde{\gamma}'(\tilde{\eta}(\eta)) = \Gamma_R \left( \frac{N}{2}, \frac{N\tilde{\eta}(\eta)}{2\sigma^2 C^2} \right)$ with respect to the negative PSNR, i.e., $-$PSNR. $\square$

**Proposition B.4.** *Let $\sigma_X$ and $\sigma_{\hat{X}}$ denote the sample standard deviations of $\{x_1, ..., x_N\}$ and $\{\hat{x}_1, ..., \hat{x}_N\}$, respectively. Then, the sample NCC between $X$ and $\hat{X}$ is given by*

$$\mathrm{NCC}(X, \beta_C(X)^{-1}\hat{X}) = \frac{\sigma_X}{\sigma_{\hat{X}}} + \frac{1}{\sigma_X \sigma_{\hat{X}}} \left( \frac{1}{N} \sum_{i=1}^{N} x_i \zeta_i - \bar{x}\bar{\zeta} \right), \tag{66}$$

*for $\zeta \sim \mathcal{N}(\mathbf{0}_N, \sigma^2 \|X\|_2^2 I_N)$.*

*Proof.* We recall that the reconstruction $\hat{X}$ of $X$ is obtained by the adversary using the observed privatised gradients given in 11. Due to the from $X$ independent random noise added to these gradients, they can be viewed as samples from random variables (see 13). Consequently, $\hat{X}$ can also be viewed as a sample from a random variable (see 20):

$$\hat{X} \stackrel{d}{=} \hat{Y}_X, \quad \text{for} \quad \hat{Y}_X \sim \mathcal{N}\left(X, \sigma^2 \|X\|_2^2 I_N\right).$$

For the target point $X$ being fixed and, thus, $\|X\|_2$ being also fixed, we let $\zeta \sim \mathcal{N}\left(\mathbf{0}_N, \sigma^2 \|X\|_2^2 I_N\right)$ be a random variable drawn independently of $X$. It is easy to see that

$$\hat{X} \stackrel{d}{=} X + \zeta.$$

Let

$$\bar{x} := \frac{1}{N}\sum_{i=1}^N x_i, \qquad \text{and} \qquad \bar{\hat{x}} := \frac{1}{N}\sum_{i=1}^N \hat{x}_i = \frac{1}{N}\sum_{i=1}^N (x_i + \zeta_i) = \bar{x} + \bar{\zeta}, \qquad \text{for} \qquad \bar{\zeta} := \frac{1}{N}\sum_{i=1}^N \zeta_i, \qquad (67)$$

denote the sample means of $\{x_1, ..., x_N\}$ and $\{\hat{x}_1, ..., \hat{x}_N\}$, respectively. Moreover, let

$$\sigma_X = \sqrt{\frac{1}{N}\sum_{i=1}^N (x_i - \bar{x})^2}, \qquad \text{and} \qquad \sigma_{\hat{X}} = \sqrt{\frac{1}{N}\sum_{i=1}^N (\hat{x}_i - \bar{\hat{x}})^2}, \qquad (68)$$

denote the sample standard deviations of $\{x_1, ..., x_N\}$ and $\{\hat{x}_1, ..., \hat{x}_N\}$, respectively. Then, by definition of the sample NCC (see Definition 5), it follows

$$
\begin{aligned}
\text{NCC}(X, \hat{X}) &= \frac{1}{N}\sum_{i=1}^N \frac{(x_i - \bar{x})}{\sigma_X} \frac{(\hat{x}_i - \bar{\hat{x}})}{\sigma_{\hat{X}}} \\
&= \frac{1}{N}\sum_{i=1}^N \frac{(x_i - \bar{x})}{\sigma_X} \frac{(x_i - \bar{x} + \zeta_i - \bar{\zeta})}{\sigma_{\hat{X}}} \\
&= \frac{1}{N}\sum_{i=1}^N \frac{(x_i - \bar{x})^2 + (x_i - \bar{x})(\zeta_i - \bar{\zeta})}{\sigma_X \sigma_{\hat{X}}} \\
&= \frac{\sigma_X}{\sigma_{\hat{X}}} + \frac{1}{\sigma_X \sigma_{\hat{X}}}\left(\frac{1}{N}\sum_{i=1}^N (x_i - \bar{x})(\zeta_i - \bar{\zeta})\right).
\end{aligned}
\qquad (69)
$$

The multiplicative term $\frac{1}{N}\sum_{i=1}^N (x_i - \bar{x})(\zeta_i - \bar{\zeta})$ of 69 corresponds to the sample covariance between the entries of target $X$ and the random noise, namely between the sets $\{x_1, ..., x_N\}$ and $\{\zeta_1, ..., \zeta_N\}$. Next, we simplify the sample covariance:

$$
\begin{aligned}
\frac{1}{N}\sum_{i=1}^N (x_i - \bar{x})(\zeta_i - \bar{\zeta}) &= \frac{1}{N}\sum_{i=1}^N (x_i \zeta_i - \bar{x}\zeta_i - x_i\bar{\zeta} + \bar{x}\bar{\zeta}) \\
&= \frac{1}{N}\sum_{i=1}^N x_i\zeta_i - \bar{x}\frac{1}{N}\sum_{i=1}^N \xi_i - \bar{\zeta}\frac{1}{N}\sum_{i=1}^N x_i + \bar{x}\bar{\zeta} \\
&= \frac{1}{N}\sum_{i=1}^N x_i\zeta_i - \bar{x}\bar{\zeta}.
\end{aligned}
\qquad (70)
$$

Using 70, we conclude:

$$\text{NCC}(X, \beta_C(X)^{-1}\hat{X}) = \frac{\sigma_X}{\sigma_{\hat{X}}} + \frac{1}{\sigma_X \sigma_{\hat{X}}}\left(\frac{1}{N}\sum_{i=1}^N x_i\zeta_i - \bar{x}\bar{\zeta}\right).$$

$\square$

*Remark* B.1. Since the entries of the target $\{x_1, ..., x_N\}$ and the noise $\{\zeta_1, ..., \zeta_N\}$ are independent, the sample covariance, i.e., multiplicative term $\frac{1}{N} \sum_{i=1}^{N} x_i \zeta_i - \bar{x}\bar{\zeta}$ on the right-hand side of 66, converges to its theoretical value, i.e., zero, for $N \to \infty$. Otherwise, for $N < \infty$, the sample covariance can be positive or negative, increasing or decreasing the right-hand side of 66 arbitrarily, hindering the computation of meaningful upper and lower bounds for the sample NCC.

**Proposition 6.** *Let $x$ and $\hat{x}$ be the two random variables as defined above. Then,*

$$\text{NCC}(x, \hat{x}) = \sqrt{\frac{1}{1 + \sigma^2 \|X\|_2^2 / \text{Var}(x)}} \leq \sqrt{\frac{1}{1 + \sigma^2 N}}. \tag{28}$$

*Proof.* Using the definition of the NCC (see Definition 5) it follows:

$$\begin{aligned}
\text{NCC}(x, \hat{x}) &= \frac{\text{Cov}(x, \hat{x})}{\sqrt{\text{Var}(x)}\sqrt{\text{Var}(\hat{x})}} \\
&= \frac{\text{Cov}(x, x) + \text{Cov}(x, \zeta)}{\sqrt{\text{Var}(x)}\sqrt{\text{Var}(x) + \text{Var}(\zeta)}} \\
&= \frac{\text{Var}(x)}{\sqrt{\text{Var}(x)}\sqrt{\text{Var}(x) + \text{Var}(\xi)}} \\
&= \sqrt{\frac{\text{Var}(x)}{\text{Var}(x) + \text{Var}(\zeta)}} \\
&= \sqrt{\frac{1}{1 + \text{Var}(\zeta)/\text{Var}(x)}}. \tag{71}
\end{aligned}$$

We recall that $\text{Var}(\zeta) = \sigma^2 \|X\|_2^2$ by definition. Thus, using the assumption that $\text{Var}(x) \leq \|X\|_2^2/N$, it follows from 71:

$$\text{NCC}(x, \hat{x}) = \sqrt{\frac{1}{1 + \sigma^2 \|X\|_2^2 / \text{Var}(x)}} \leq \sqrt{\frac{1}{1 + \sigma^2 N}}. \tag{72}$$

$\square$

**Proposition 7.** *Assuming the adversary can match the $k$ reconstructions $\hat{X}_1, ..., \hat{X}_k$ to the same data sample $X$, they can average them. Let $\hat{X}_{\text{avg}} = \frac{1}{k} \sum_{j=1}^{k} \hat{X}_j$ denote the averaged reconstructed vector. Then the following holds for the $i$th entry of $\hat{X}_{\text{avg}}$:*

$$\mathbb{E}[\hat{x}_{i\,\text{avg}}] = x_i \quad and \quad \text{Var}[\hat{x}_{i\,\text{avg}}] = \frac{\sigma^2 \|X\|_2^2}{k}. \tag{29}$$

*Proof.* Let $\hat{X}_j$, for $j \in \{1, ..., k\}$, denote reconstructions of the same data sample $X$ obtained separately and independently by performing multiple attacks. The distribution of each reconstruction $\hat{X}_j$ is given in Equation 20 (Section 4.1). Let $\hat{x}_{j,i}$, for $j \in \{1, ..., k\}$ denote the $i$th coordinate of $\hat{X}_j$. Then, the expectation and variance of the $i$th component of $\hat{X}_{avg}$, i.e., $\hat{x}_{i\,avg}$, can be computed in the following way:

$$\mathbb{E}[\hat{x}_{i\,avg}] = \mathbb{E}\left[\frac{1}{k} \sum_{j=1}^{k} \hat{x}_{j,i}\right] = \frac{1}{k} \sum_{j=1}^{k} \mathbb{E}[\hat{x}_{j,i}] = x_i,$$

and

$$\text{Var}(\hat{x}_{i\,avg}) = \text{Var}\left(\frac{1}{k} \sum_{j=1}^{k} \hat{x}_{j,i}\right) = \frac{1}{k^2} \sum_{j=1}^{k} \text{Var}(\hat{x}_{j,i}) = \frac{1}{k^2} k \sigma^2 \|X\|_2^2.$$

$\square$

## C   Additional Figures

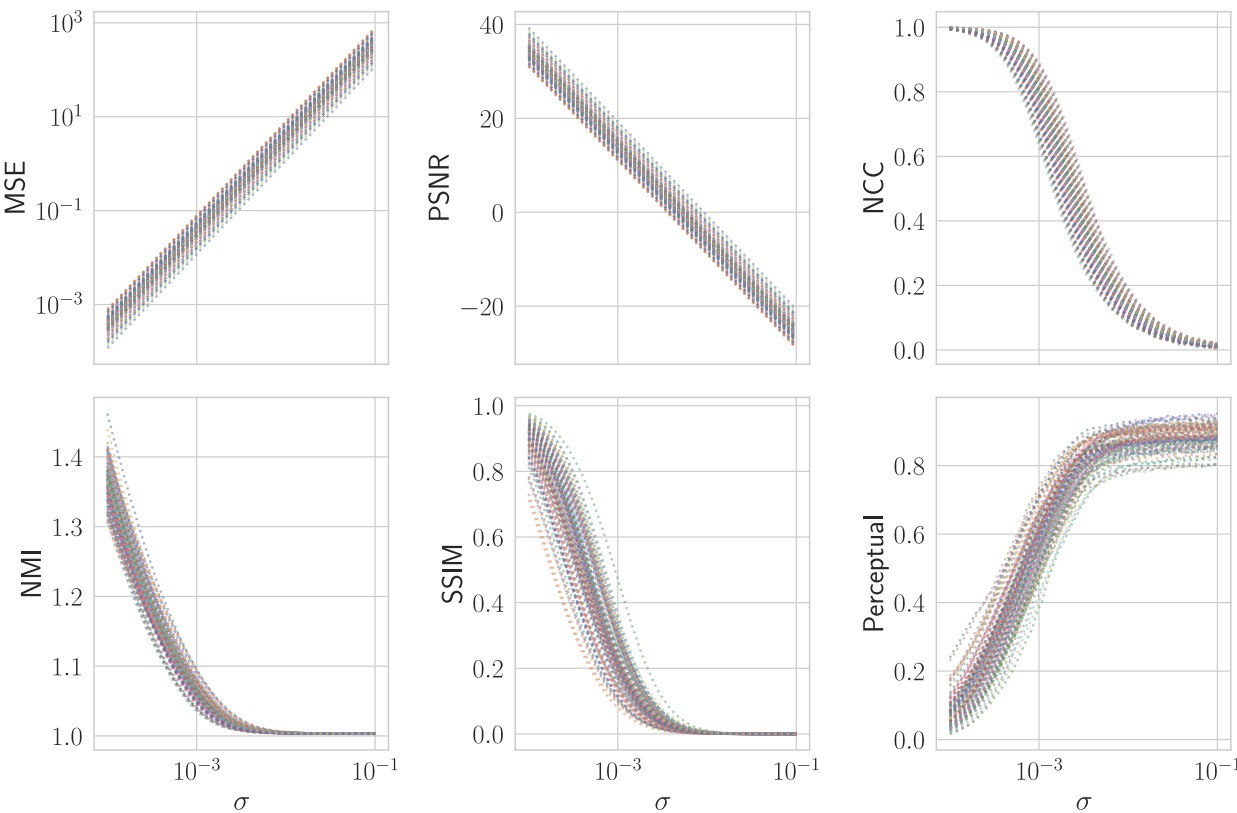

Figure 4:   Reconstruction metrics for 100 reconstructed data samples from the ImageNet dataset under varying $\sigma$. Additionally to the MSE, PSNR, and NCC we also include the Normalized Mutual Information (NMI), Structural Similarity (SSIM) and a perceptual loss. We see that the NMI, SSIM and the perceptual loss show a very similar result as our bounded metrics, especially the NCC. Of note, the perceptual loss inherently incorporates an image prior due to the network being trained on image data. Although this violates our assumption for the adversary, the metric still aligns with all other reconstruction metrics.

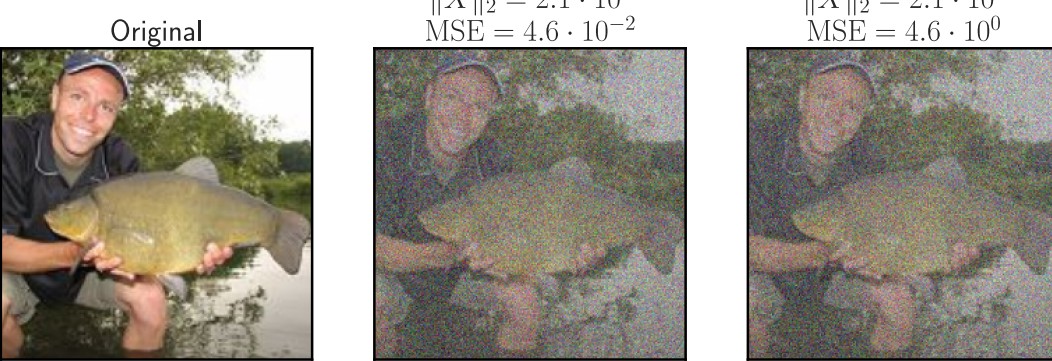

Figure 5: Demonstration how the $\ell_2$-norm of the data sample influences the MSE. The same image, which is rescaled by a constant factor $a = 10$, is being reconstructed. We keep $\sigma = 1.0 \cdot 10^{-3}$ constant for both and set $C = \|X\|_2$. The resulting MSE is scaled by the factor $a^2$.

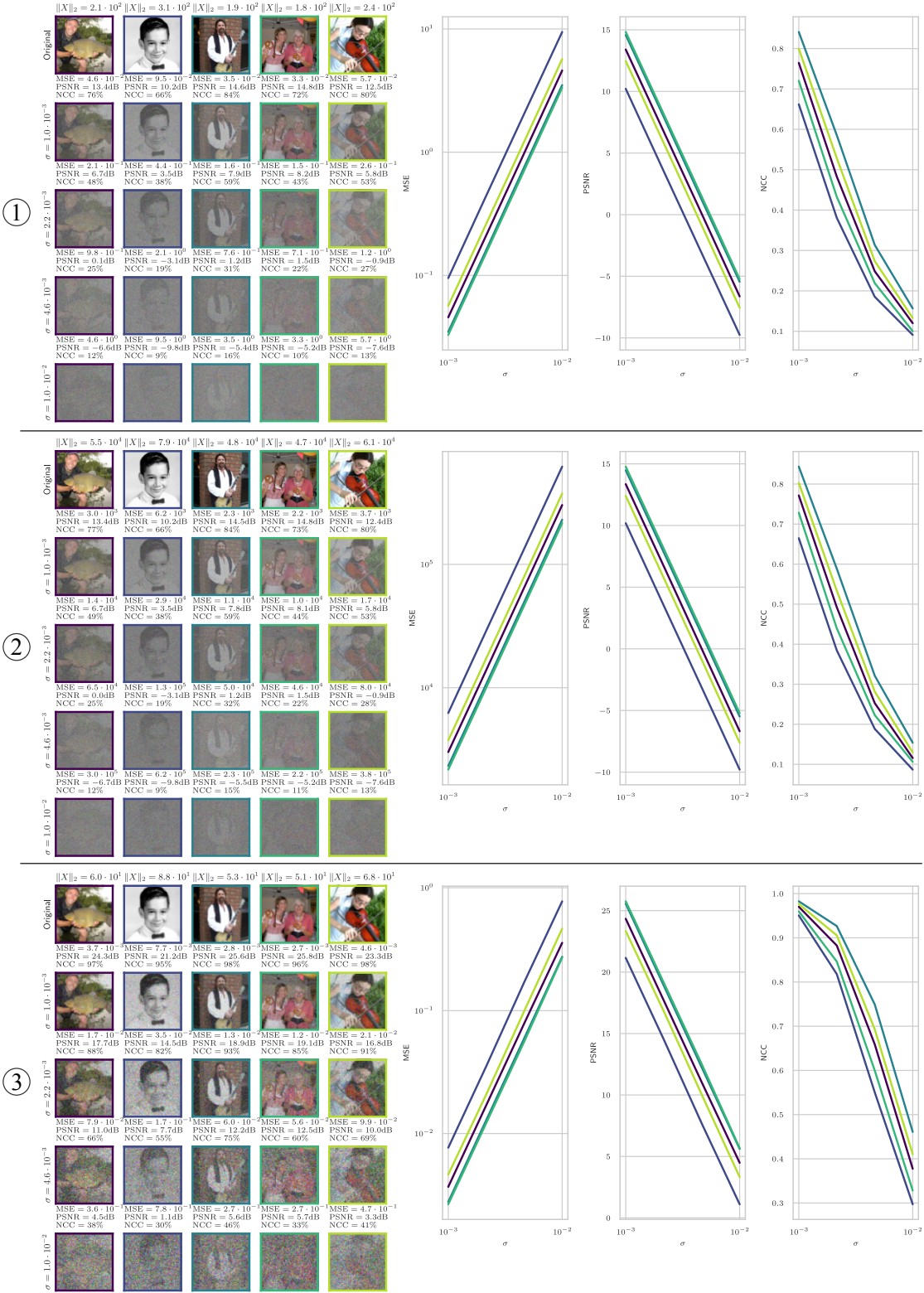

Figure 6: Exemplary reconstructions on the ImageNet dataset (Deng et al., 2009) under varying $\sigma$. Scenario (1): $N = 224 \times 224 \times 3$, data range in $(0,1)$. Scenario (2): $N = 224 \times 224 \times 3$, data range in $(0,255)$. Scenario (3): $N = 64 \times 64 \times 3$, data range in $(0,1)$. All scenarios: $C = 1$, $M = 1$. The colours of the image frames match the corresponding metric curves.

