# OpenReview forum: "From Mean to Extreme: Formal Differential Privacy Bounds on the Success of Real-World Data Reconstruction Attacks"
_TMLR — Rejected by TMLR_

### Review · Reviewer_DZ1Q · 2025-03-01

**Summary Of Contributions:**

The authors consider reconstruction attacks against a neural network with two constraints: (a) the first layer is a fully-connected linear layer (of the form $WX+b$) (b) the network is the outcome of DP-SGD training: gradients are scaled so that their L2 norms are bounded by $C$ before being noised by Gaussian samples. Prior work showed "gradient inversion" is possible with networks that just have property (a): it suffices to perform an entry-wise division between the gradient with respect to a row of $W$ and partial derivatives. Attacks against networks that have both properties (a) and (b) are inherently more challenging due to distortions in both numerator and denominator.

Under the assumption that the attacker has noiseless access to scaling factors (denominators of the division), the authors derive a variant of the earlier attack that yields an estimator with minimum variance and zero bias [Sec. 3.1]. Then they give characterizations of reconstruction success via MSE, Peak Signal-to-Noise Ratio (PSNR), and Reconstruction Robustness (ReRo) [Sec. 3.2.1] followed by Normalized Cross-Correlation (NCC) [Sec. 3.2.2]

**Audience:**

Yes

**Claims And Evidence:**

No

**Requested Changes:**

See weaknesses --- clarity of the model and justification thereof are very much needed.

Less significant points:
- The authors write that "we believe that only practitioners who can accurately estimate the risks for their specific threat model can make an informed decision about an appropriate choice of privacy parameter." This statement seems like a *non-sequitur*: how does the debate around privacy parameter relate to the question of threat model? More precisely, the question of threat model is qualitative (what do we believe the adversary can do?) while the choice of privacy parameter is quantitative (how much risk do we tolerate?).
- Add pointers to the proofs. When I first saw Proposition 1, I was under the impression that the text that followed was the proof / would preface a proof. Then I wondered if there was any proof at all.
- the bias term looks like a vector but is referred to as "scalar term" (see the text above eq (1))

**Strengths And Weaknesses:**

Attacks against DP learning and analysis thereof are always welcome. I found the exposition in Sec. 2.1 generally easy to read.

The contributions are made under a threat model that is never made fully precise. The authors write that "real-world attacks operate under a relaxed threat model, i.e., the considered adversaries have fewer capabilities and/or knowledge and are thus weaker [than worst-case]"  which justifies their work on "semantic privacy guarantees of Gaussian mechanisms." *But this model is not made concrete* even with the plethora of citations: the reader is obliquely told similarities and differences with prior work but not told up front what the model exactly is.

The moments that the authors provide hints to the model are either confusing or lack motivation. For example,
- In what natural scenarios can an adversary dictate the "the model’s architecture, hyperparameters and loss function" but still be forced to deal with clipping and DP noise?
- why is it natural to assume the adversary has access to the scaling factors? I understand we sometimes make simplifying assumptions to get our proverbial foot in the door, but there should be a follow-up section or comment that removes the assumption. Alternatively, the authors could re-phrase the theoretical work as a derivation of an attack under the optimistic condition where the noise on the scaling factors is sufficiently small.

In Section 1.3, it is odd that there is no condition on $\pi$; using ReRo as written will consider the "smoking causes cancer" scenario to be privacy violating. Specifically, if the prior is such that the adversry already knows a person smokes with 99% probability, recovering correlations with smoking should not count as a leakage by the algorithm in question but it seems that Secttion 1.3 would consider it a problem. I know that this is partially addressed in the Introduction ("an adversary is rather interested in recovering the data sample that is not already in their possession.") but this condition needs to be made more clear.

---

> ### Author Response · Authors · 2025-05-28
> **Clarifications on the threat model**
>
> We sincerely thank you for your constructive feedback, which has contributed to improving the clarity and quality of our work. Please find below our detailed responses to the weaknesses and concerns you raised.
>
>
> **Clarification of the Threat Model**
>
> In response to your feedback, we have introduced a new subsection (Subsection 3.1: Threat Model) in which we explicitly define the threat model employed in our study.
> Furthermore, to help readers better contextualise our contributions earlier on, we have revised both the abstract and the introduction to clearly state the specific reconstruction attack considered. We also highlight key characteristics of the threat model, such as the ability to manipulate the model either before or during training, and the absence of any need for knowledge of the input data beyond its dimensionality (see paragraph 2 of the abstract and paragraph 5 of the introduction).
>
> **Qualitative Threat Model vs Quantitative Privacy Parameters**
>
> Thank you for raising this important point – it highlights a key aspect of our work that we can clarify further, namely the distinction between threat model (qualitative) and privacy parameter selection (quantitative).
>
> You are correct that defining the threat model involves understanding what an adversary is capable of, while choosing a privacy parameter like $\gamma$ determines how much risk we’re willing to tolerate. However, these aren't independent. The level of acceptable risk – and therefore the appropriate value for $\sigma$– directly depends on the capabilities outlined in the threat model.
>
> To illustrate: if we assume a very powerful adversary (a strong threat model), we need to choose a higher noise multiplier $\sigma$ (assuming constant parameters otherwise) to provide sufficient protection (i.e., the same risk level $\gamma$) against their attacks. Conversely, with a weaker adversary, a lower noise multiplier suffices to guarantee the same risk level. Essentially, the quantitative choice of the privacy parameters is informed by the qualitative assessment of the threat model.
>
> We view privacy parameters not as an arbitrary setting, but as a function of the adversarial capabilities. This connection is central to our analysis and allows us to rigorously assess the privacy-utility trade-off under realistic attack scenarios.
>
> **Motivation and Justification of the Threat Model**
>
> We propose a threat model that, to the best of our knowledge, has not been previously analysed in the context of semantic guarantees and captures the essential capabilities of an adversary performing analytic gradient inversion attacks.
> Prior works (Boenisch et al., Fowl et al., Feng & Tramér) have shown that the adversary specified by the chosen threat model can successfully execute an analytic gradient inversion attack and, thus, reconstruct sensitive input data.
>
> Therefore, we motivate our threat model by underscoring the relevance of analytic gradient inversion attacks. These attacks are noteworthy both theoretically and practically: they are difficult to detect in real-world settings, enable data reconstruction, and are based on computationally low-cost techniques. Most importantly, they are directly rooted in the prior works by Boenisch et al. & Fowl et al., which we complement by deriving theoretical bounds. While these motivations were previously included in the appendix, we have now incorporated them into the main body of the manuscript to better support the rationale behind our work (see paragraph 6 of the introduction).

---

> ### Author Response · Authors · 2025-05-28
> **On whether assumptions are "natural"**
>
> **"In what natural scenarios can an adversary dictate the ”the model’s architecture, hyperparameters and loss function” but still be forced to deal with clipping and DP noise?":**
>
> Thank you for raising this important point. Analytic gradient inversion attacks are particularly feasible in federated learning (FL) settings involving a malicious central server, where gradient updates are exchanged during training (Fowl et al., Boenisch et al.). Additionally, adversaries with access to the model both before and after training can embed data traps that encode intermediate gradients into model weights, enabling the reconstruction of sensitive data even post-training (Feng & Tramér). Both variants of this attack are highly relevant in practice. In response to your question, we have added a new paragraph to the introduction (paragraph 5) to make this clearer to the reader.
> In the FL setting, participating nodes can train with DP, thereby sharing privatised model updates (i.e., privatised gradients) with the central server. Similarly, a pretrained model can be fine-tuned using DP such that the released model satisfies a DP guarantee.
> In both scenarios, under the assumed threat model, while an adversary may have the ability to manipulate the model’s architecture, hyperparameters, and loss function, they cannot control the DP parameters, which are set independently by the data-owning parties.
> To clarify this point in the manuscript, we have added these examples to the second-to-last paragraph of Section 2.1 and additionally clarified this in the threat model (Section 3.1).
>
> **"why is it natural to assume the adversary has access to the scaling factors?**
>
> We appreciate the opportunity to clarify this point. As stated at the end of Section 3.3 ("Analytic Gradient Inversion Attacks under DP"), the observed noisy version of the scaling factors cannot be used directly. However, the adversary can leverage these observations to construct approximations of the scaling factors first and then use these approximations in the estimation problem for X. In this context, the best-case scenario for the adversary—corresponding to the worst-case scenario from a privacy perspective—is one in which the scaling factors are perfectly reconstructed.
> Since our goal is to establish a generalisable theoretical privacy bound, it is necessary to consider this most successful attack scenario.
> Moreover, this assumption is not purely hypothetical; it is realistic in certain settings. For example, when the maximum and minimum values of the data are known—as is often the case with image data where pixel values range from 0 to 255—the adversary may be able to infer the scaling factors.
> To improve clarity, we have rewritten this explanatory text at the end of Section 3.3, explicitly framing it in terms of the best-case scenario for the adversary, which is the appropriate case to consider when constructing theoretical bounds.

---

> ### Author Response · Authors · 2025-05-28
> **ReRo & Misc**
>
> **“In Section 1.3, it is odd that there is no condition on $\pi$; using ReRo as written will consider the "smoking causes cancer" scenario to be privacy violating. …I know that this is partially addressed in the Introduction ("an adversary is rather interested in recovering the data sample that is not already in their possession.") but this condition needs to be made more clear.”**
>
> We acknowledge the reviewer's concern about scenarios where the adversary possesses prior knowledge. Our work explicitly focuses on analysing reconstruction risk under the assumption of no prior knowledge about the data beyond its dimensionality. As highlighted in our manuscript (and citing Schwethelm et al., 2024, and Hayes et al., 2023), adversaries with access to prior information can indeed surpass our established bounds. This is because leveraging such knowledge allows for more effective reconstruction strategies that are not captured by our current threat model.
> We view this as a limitation of our work – one we explicitly acknowledge and decided to include – rather than a flaw in the ReRo metric itself. Our goal was to provide rigorous guarantees under a specific, well-defined threat model, and, as outlined in the discussion, future research will be needed to develop bounds applicable to scenarios with varying degrees of prior knowledge.
> To clarify this point, we have added a comment on this limitation of our work in the final paragraph of Section 3.1.
>
> **Less significant points:**
>
> Thank you for pointing out that we had missed referencing the proofs. We have now added a note at the beginning of Section 4 to clarify that all proofs can be found in the appendix.
> Additionally, we have corrected the text above Equation (4) (previously Equation (1)); it now correctly refers to the "bias term."
>
>
> We hope that our responses and the revised manuscript adequately address your concerns.
> We would be happy to clarify any further questions you may have.

---

### Review · Reviewer_KWnq · 2025-03-05

**Summary Of Contributions:**

- The paper presents an analysis of data reconstruction attacks on ML models protected by differential privacy (DP)
- Prior studies focused on worst-case scenarios, which led to overly cautious estimates and unnecessarily strong privacy settings. Instead, this paper focuses on probabilistic bounds that reflect realistic threat scenarios.
- The paper anchors on gradient inversion attacks as the underlying procedure for all reconstruction attacks. It then analyzes gradient inversion attacks with theoretical derivations of probabilistic bounds on attack success, measured using mean squared error (MSE), peak signal-to-noise ratio (PSNR), and normalized cross-correlation (NCC). The paper shows that different metrics may reflect different reconstruction success.

**Audience:**

Yes

**Broader Impact Concerns:**

A broader impact statement is not included. I recommend that the authors discuss the risks associated with misinterpretation or incorrect application of the proposed theoretical results / bounds, since a misunderstanding or misapplication by practitioners could lead to unintended privacy vulnerabilities.

**Claims And Evidence:**

No

**Requested Changes:**

Writing:
- the main issue is that the paper is packed with dense sentences and a lack of intuitive explanations or clear narrative transitions between sections. The paper would significantly benefit from:
  - simpler, more concise explanations of concepts and results
  - clearer, step-by-step motivation for why certain attacks/metrics are studied and for each theoretical development, explicitly stating why each step matters practically
  - intuitive examples earlier in the paper to illustrate key results before introducing technical derivations
  - a better flow of writing: improved transitions to guide the reader smoothly through the logic and relationships between sections and ideas

Misc:
- provide concrete, practical examples or guidelines that practitioners can use to select DP parameters based on realistic trade-offs
- add explicit comparisons to previous studies (instead of just citing them) that address similar threat models, clearly differentiating the contributions and practical utility of the present work
- expand on the empirical validation to include more diverse datasets or realistic case studies to demonstrate the practical applicability of derived bounds

**Strengths And Weaknesses:**

Strengths:
- the paper studies a relevant problem of analyzing data reconstruction success under realistic (as opposed to worst-case) adversaries

Weaknesses: The main and fatal weakness is that the paper, in my opinion, is poorly written.
- The prose is a bit long and redundant, and has a lack of focus. Much of the technical writing depends on the reader going through prior papers (e.g. Balle et al., 2022), which severely limits readability. There are also heavy use of mathematical notations which somewhat obscures the delivery of the paper.
- Example writing issues in introduction:
  - "Gradient inversion attacks" are first mentioned in Sec 1.1 (contributions) and are put front and center of the paper, but it was not mentioned at all in the introduction. It may need better motivation and context as to why this is relevant to study.
  - Similarly, the choice of metrics (MSE, PSNR, NCC) are not mentioned in the intro, and yet directly presented as a contribution.
- There is very limited discussions on what the takeaways are for the theoretical derivations (sec 3); and even the discussions in sec 4 (visualization and interpretations, including recommendations for practitioners) does not seem to refer back to the derived theorems/propositions.
- Unclear practical value of NCC bounds (even pointed out by the authors: "in the context of our work, for high values of N , the NCC is not necessarily a good metric to assess the reconstruction success of the adversary"), yet it was phrased as a core contribution
- If the paper aims to focus on practicality of the attacker, then important aspects such as the implications of repeated queries and larger batch sizes should be more adequately explored.
- The paper lacks concrete guidelines or practical examples to clearly demonstrate how these theoretical results should inform practical parameter settings in real-world deployments.

---

> ### Author Response · Authors · 2025-05-28
> **Thank you for the detailed feedback concerning clarity & narrative**
>
> We sincerely thank the reviewer for their detailed and constructive feedback. We have undertaken significant revisions to address all concerns raised and believe these changes substantially strengthen the paper. While one reviewer noted the neatness of our writing, we acknowledge areas where clarity and structure could be improved and have focused on enhancing these aspects throughout the manuscript.
>
> 1. Abstract & Introduction Rewrite: We have completely revised the abstract and introduction to provide a more focused narrative. The revised introduction now clearly articulates the problem of data reconstruction attacks, motivates the study of analytic gradient inversion attacks, and explicitly states our contributions – namely, providing formal bounds on reconstruction success under specific conditions.
>
> 2. Related Work Revision: The related work section has been thoroughly revised to provide a more nuanced comparison with existing studies. We explicitly highlight how our approach differs and builds upon prior works, detailing the limitations of previous bounds and emphasising the novelty of our analysis in terms of threat model, metrics considered, and theoretical rigour.
>
> 3. New Background Section: We have added a dedicated background section to provide readers with essential context on:
> *   Differential Privacy (DP): Including a clear explanation of $(\varepsilon, \delta)$-DP and its relevance to privacy-preserving ML.
> *   Reconstruction Robustness (ReRo): Defining the $(\eta, \gamma)$-ReRo framework and explaining its utility for quantifying reconstruction risk.
> *   Analytic Gradient Inversion Attacks: Describing the attack methodology and its practical implications.
>
> 4. Enhanced Theoretical Derivations: We have specified the key theoretical results that serve as the main takeaways, highlighting their practical significance (for example of Corollary 1). Moreover, we have associated these takeaways with figures that visually illustrate the corresponding results. This also includes referencing the mathematical results to the interpretations presented in the (now) Section 5.1, titled 'Visualisation and Interpretation'.
>
> 5. Practical Implications Paragraph: We included a new paragraph explicitly outlining how our results can inform practitioners’ decisions regarding privacy parameter selection, emphasising that while our bounds provide a quantifiable measure of reconstruction risk, we cannot lift the decision for concrete risk levels from the user. These need to be decided upon considering local laws, guidelines, scenarios and datasets.
>
> 6. Improved Transitions & Flow: Throughout the manuscript, we focused on improving transitions between paragraphs and sections to create a smoother and more coherent narrative. We used clear signposting language to guide readers through the logic of our arguments.
>
> Regarding Specific Concerns:
>
> Additional Datasets: We are happy to include additional datasets if the reviewer insists. To facilitate this, we would appreciate clarification on which specific datasets they believe would be most beneficial for evaluating our bounds. We also remark that we focused on imaging datasets as they can be easily visually interpreted. In case you want us to add a different modality, we kindly ask for guidance on the display of the outcome. However, we do not currently see a strong justification for adding further empirical validation, as the generalizability of our theoretical results is independent of the data distribution.
>
> While undertaking an analysis of privacy-utility trade-offs for a specific dataset could help demonstrate the potential of our bounds, we believe that generalising these results to other tasks and datasets would be unreliable due to task-specific variations in model architecture, training procedures, and data characteristics.
>
> Unclear practical value of the NCC bounds: Thank you for your comment. We have revised the manuscript to clarify the practical value of the NCC bounds. Specifically, we now emphasise that the NCC bound can be interpreted in a comparative manner, relative to a case-specific threshold that distinguishes between informative and non-informative reconstructions (see the end of Section 4.2.2).

---

### Review · Reviewer_PmVC · 2025-05-17

**Summary Of Contributions:**

This papers considers the setting where a clipped and noised (individual) sample gradient from a neural network (whose first layer is a fully connected layer with bias) is accessible to an external attacker, with the aim of reconstructing the sample.

Reconstruction is quantified via different metrics, and the authors quantify the resulting quantity for the attack discussed, for example, in Boenish&al.2023 (https://arxiv.org/pdf/2112.02918), which I think was previously considered in Geiping&al.2020 (https://arxiv.org/abs/2003.14053). The authors comment and analyze the dependency of the attack success on the private noise scaling, the clipping constant, and the width of the first layer.

**Audience:**

Yes

**Claims And Evidence:**

No

**Requested Changes:**

I invite the authors in reconsidering the narrative of the paper. In particular, I would avoid any reference in how practitioners should choose the hyper-parameters for a given privacy budget to minimize the success of this recovery attack. I am suggesting this because I do believe that at the current stage, this paper provides misleading take-aways.

One example, mentioned above, is to narrow the contribution to the considered setting: fully connected first layer, attack from Geiping&al.2020 (https://arxiv.org/abs/2003.14053), only one gradient is accessible taken with respect to an individual sample, scalings with respect to $\sigma$ and $C$. This setup is studied well enough in the current version.

**Strengths And Weaknesses:**

This paper is neatly written, mathematical steps are clear and seem correct, and considers a problem of interest. However, I have few problems with its narrative and presentation, which motivate my choice regarding the "claims and evidence" section.

- - - -

A first weakness of this paper is what I believe to be a lack of clarity in the introduction. I believe this paper does what I wrote in the "summary of contributions" section. It looks at one specific attack, for one type of architecture, as a function of 3 variables, which are the clipping constant $C$, the noise multiplier $\sigma$, and the width of the first hidden layer $N$. This paper does not "provide formal bounds on the protection of DP" and arguably does "equip practitioners with a larger wealth of information to decide upon a certain set of privacy parameters". I will elaborate below:

DP protection is parameterized by some values, e.g. $(\varepsilon, \delta)$ in the context of $(\varepsilon, \delta)$-DP. If we set these value sufficiently large, no effective protection is granted (e.g. $\delta = 1$). Then, if the goal is to track the effects of DP in terms of protection, I do think it is imporant to track "the effects of these numbers" to bee able to claim any quantitative statement. Importantly, for a fixed optimization algorithm (let's say DP-SGD with batch size equal to 1, which is probably the closest setup to what this work looks at), the number of training iterations (let's set them to $T$), $C$, and $\sigma$ determine the resulting privacy parameter, which I emphasize not being something "abstract". In practice, more privacy brings less performances, and I agree with the authors that "practitioners" should be interested to see the effects of the privacy paramenters with respect to the resulting attack (let's say, $\varepsilon = 2$ vs. $\varepsilon = 8$...). However, this work is not doing this, as the dependence with privacy is implicit with $C$ and $\sigma$ for one gradient step.

I will give an example on how this might be misleading. Equation (14) seems to suggest that setting $\sigma$ very large would protect gradient more from external attacks. However, from the perspective of the "practitioner" that sets the value of $\sigma$ before training, setting this value large will lead (for a fixed $C$ and a fixed privacy budget) simply to train for more iterations (see, for example, https://arxiv.org/abs/1607.00133), which in turn could mean that the number of gradients that are released is higher, giving the option to the attacker of producing better estimates of the original sample. Very similar story applies for $C$, which I need to stress is not "an arbitrary hyper-parameter, chosen on experience"... I invite the authors to review the effects of the privacy hyper-parameters in deep learning training, which are very clearly discussed, for example, in https://arxiv.org/abs/2204.13650 , which is also cited in this work.

- - - -

Another major problem or question I have is why the authors assume the attacker has the freedom to change the architecture (the value of $M$ to be precise). This is something I am missing in the analysis. The attacker should have access to the gradient with respect to the sample $X$, which is taken with respect to the original architecture with width $N$. Why the attacker has the freedome to "replace the entire network by the linear function [...]"? I am missing the point of defining two architectures here, can the author further explain the setting?

- - - -

Another problem I have with the current manuscript is the a bit vague formulation of the Assumptions. As mentioned above, I do think correctly describing the model / attack (first fully connected layer + bias, gradient leakage of an individual sample, gradient clipped with $C$ and noise $\sigma$) in the abstract is important for a fair description of the contributions (at the moment this is all implicit in the rather uninformative string "against reconstruction attacks"). Beyond this, there are also minor assumptions and logical leaps in the discussion.

- assumption on that the gradients with respect to the first layer are sufficient to estimate $X$. This is not really an assumption, but rather a restriction on the attacks that are being studied, which is a reasonable assumption if stated clearly.

- assumption on the fact that attacker accesses the not noisy gradient on the biases. This, as discussed, gives more information to the attacker and makes the analysis easier. Maybe I would rephrase "the most precise reconstruction an adversary..." (right before Section 3) to "we provide an upper bound to...", explaining when this upper bound will not be tight.

- assumption on $C$ to be larger than $\| X\|_2$. This is used to frame the result without this parameter. This is done in a bit obscure way, and I remark that there is empirical evidence to favour small clipping constants during training (see again https://arxiv.org/abs/2204.13650 ).

- in proposition B.2 in the Appendix the author make a not restrictive assumption on $X \neq 0$. I would change the phrasing "without loss of generality" to "with probability 1" or specifying this assumption earlier. Also, in this proposition the authors are looking at the limit of $M \to \infty$ which I find misleading. Changing the architecture, leaving the clipping constant fixed, moves the optimal hyper-parameters to achieve the same utility given a fixed privacy budget (Tables 13 and 14 in https://arxiv.org/abs/2204.13650), so taking this limit leaving everything fixed has the effect of either considering models with different privacy guarantees, or considering models with very different utility. I think this limit is useful as it gets rid of the term $\|rest\|_2^2$, but this also stresses how neglecting this term gives less and less tight estimates if bigger architectures are considered (leaving fixed $M$ and increasing the relative size of $\|rest\|_2^2$ with respect to the norm of the first layer gradients).

---

> ### Author Response · Authors · 2025-05-28
> **DP protection is parameterized by some values [...]**
>
> Thank you for raising this important point regarding the connection between our reconstruction bounds and standard differential privacy parameters (ε, δ).
> Our primary focus has been on providing formal bounds on the protection offered by the Gaussian mechanism within a single iteration of DP-SGD. The overall privacy guarantee for a model trained with T iterations can be computed using established composition theorems (e.g., as described in Dong et al., 2022).
> However, we recognise the importance of understanding how composition impacts our guarantees. To address this, we have added an analysis demonstrating how multiple gradient steps impact our bounds using a simple composition theorem (Section 4.3). While this provides a preliminary estimate, we acknowledge that more sophisticated composition techniques – particularly those accounting for subsampling – are crucial for real-world DP training and represent an important and valuable direction for future work. Specifically, our baseline composition theorem is only tight if the adversary can successfully match reconstructions to the same target record across all T iterations. However, in more realistic settings where the dataset has more than one sample and therefore the probability of resampling the same data point is p < 1, our bounds become pessimistic – overestimating reconstruction risk. This is because the assumption of repeated reconstructions of the same target record becomes less likely.
>
> Therefore, while we provide bounds for a single iteration, extending these to account for composition and subsampling requires further investigation. We can offer an upper bound based on worst-case assumptions (perfect matching across iterations), acknowledging that this will be loose in many practical scenarios. A more refined analysis would necessitate developing strategies for modelling the probability of repeated reconstructions and is left as future work. We also remark that this is not trivial; for classical DP accounting, years of research were necessary to provide tight, subsampling composition theorems.
>
> Importantly, our work isn’t aimed at guiding practitioners in choosing optimal (ε, δ) values. Rather, it provides a framework for understanding how DP protects against specific reconstruction attacks given a fixed set of privacy parameters (σ and C). Even under these worst-case assumptions, our results demonstrate that DP offers quantifiable protection.

---

> ### Author Response · Authors · 2025-05-28
> **“...C, which I need to stress is not "an arbitrary hyper-parameter, chosen on experience…"**
>
> Thank you for addressing the importance of the privacy hyperparameter C. We agree that the choice of all parameters and hyperparameters can influence the privacy-utility trade-off, as clearly discussed in the work by De et al. (2022). Accordingly, we have revised the referenced sentence in our manuscript to avoid suggesting that C is simply “arbitrarily chosen based on experience.” Additionally, we have added a comment acknowledging that this work is a good source for understanding the interplay between all hyperparameters and parameters that ultimately impact both privacy and performance.
>
> We have retained the original comment that C = 1 is a typical choice made by practitioners. For instance, as noted in Section 3.1 (second paragraph) of De et al. (2022), the authors set C = 1 for all their experiments. Moreover, setting C to 1 can simplify computations and reduce the cost of hyperparameter tuning—an approach also supported in Appendix B of the cited work.
>
> However, it’s important to note that while C controls the clipping threshold, an adversary can still circumvent its effect by increasing M. This suggests that for a fixed ratio of $\mu=\frac{C}{\sigma}$ (i.e. $\mu$ in GDP) reconstruction success is not solely determined by the specific value of C. We believe this nuance is important to highlight as it demonstrates the interplay between different privacy-preserving mechanisms. This is also addressed in the last paragraph of Section 5.1.

---

> ### Author Response · Authors · 2025-05-28
> **Why do the authors assume the attacker has the freedom to change the architecture (the value of M to be precise)?**
>
> We acknowledge that our assumption regarding the attacker's ability to modify the network architecture (specifically, the value of M) requires careful justification. This choice is motivated by prior work demonstrating that attackers capable of influencing model parameters represent a realistic threat (e.g., Boenisch et al., Fowl et al.). We frame this as a worst-case scenario: an attacker seeking to maximise reconstruction success may strategically alter the architecture to optimise their attack.
>
> Introducing two architectures is essential for mathematically demonstrating which network enables a more effective attack. The key insight is that by replacing the entire network with a single linear layer, the adversary ensures that all gradient information contributes directly to reconstructing the input data. This avoids any penalty on the gradient norm from parts of the network not used for reconstruction – effectively concentrating all available signal into the gradient of the linear layer and minimising its variance.
>
> We recognise that a more subtle attacker might choose to insert a linear layer without fully replacing the network, potentially for a stealthier operation. However, our bounds remain valid even under this constrained scenario because they are derived based on the worst-case optimisation of maximising usable gradient information.
>
> It’s important to note that this threat model is stronger than those considered in works like Geiping et al., which focus on honest-but-curious adversaries with limited access. Because our analysis provides bounds for a more powerful attacker, it implicitly also holds for scenarios where the adversary does not modify the architecture.

---

> ### Author Response · Authors · 2025-05-28
> **Vague formulation of assumptions**
>
> In response to your feedback, we have introduced a new subsection (Subsection 3.1: Threat Model) in which we explicitly define the threat model employed in our study.
> Furthermore, to help readers better contextualise our contributions earlier on, we have revised both the abstract and the introduction to clearly state the specific reconstruction attack considered. We also highlight key characteristics of the threat model, such as the requirement to manipulate the model either before or during training, and the absence of any need for knowledge of the input data beyond its dimensionality (see paragraph 2 of the abstract and paragraph 5 of the introduction).

---

> ### Author Response · Authors · 2025-05-28
> **Assumption on that the gradients with respect to the first layer are sufficient to estimate X.**
>
> We value the opportunity to address our approach in more detail.
> As noted in Section 3.3 (“Analytic Gradient Inversion Attack under DP”), we initially focus solely on the updates with respect to the linear layer. This simplification enables the construction of an estimator that is analytically tractable, but as outlined below does not limit the generalisability of our bounds.
> In Section 4.1 (“Construction of the Optimal Analytic Gradient Inversion Attack under DP”), we show that replacing the entire network with a linear layer yields an estimator with minimal variance - meaning it is as accurate as possible under the constraints of differential privacy. Although gradients from other layers may contain additional information to estimate X, our analysis demonstrates that including them does not improve reconstruction accuracy and, in fact, increases uncertainty.
> Our approach, therefore, prioritises statistical efficiency and represents the best-case scenario for the adversary in terms of reconstruction success. Since our goal is to establish generalisable bounds, this is the appropriate case to consider.
> We have revised the explanatory text in Section 3.3 (second paragraph following Equations 13 and 14) to better reflect this reasoning and enhance clarity.

---

> ### Author Response · Authors · 2025-05-28
> **Assumption on the fact that attacker accesses the not noisy gradient on the biases.**
>
> We appreciate the opportunity to clarify this point. As stated at the end of Section 3.3 ("Analytic Gradient Inversion Attacks under DP"), the observed noisy version of the scaling factors cannot be used directly. However, the adversary can leverage these observations to construct approximations of the scaling factors first and then use these approximations in the estimation problem for X. In this context, the best-case scenario for the adversary—corresponding to the worst-case scenario from a privacy perspective—is one in which the scaling factors are perfectly reconstructed.
> Since our goal is to establish a generalisable theoretical privacy bound, it is necessary to consider this most successful attack scenario.
> Moreover, this assumption is not purely hypothetical; it is realistic in certain settings. For example, when the maximum and minimum values of the data are known—as is often the case with image data where pixel values range from 0 to 255—the adversary may be able to infer the scaling factors.
> To improve clarity, we have rewritten this explanatory text at the end of Section 3.3, explicitly framing it in terms of the best-case scenario for the adversary, which is the appropriate case to consider when constructing theoretical bounds.

---

> ### Author Response · Authors · 2025-05-28
> **Assumption on C to be larger than |X|_2**
>
> Could you kindly clarify where this assumption appears to be made in an obscure manner?
> In our analyses, we have included both cases where C is smaller and larger than the norm of X, to ensure completeness in our results and to avoid restricting them to a specific choice of C that may depend on the characteristics of the data. We hope these clarifications help improve the clarity and rigour of our manuscript.

---

> ### Author Response · Authors · 2025-05-28
> **Concerns regarding Proposition B.2**
>
> We appreciate the opportunity to clarify the purpose and interpretation of the result presented in Proposition B.2. This proposition is introduced in a context where the goal is to analyse how specific manipulations by the adversary to the model architecture can influence the success of the attack. As you correctly noted, modifying the model architecture – such as increasing M – can indeed affect the model's utility.
> At this stage of the analysis, our objective is to determine whether there exists a sufficiently large M such that the variance of the estimator can be “averaged out.” This question is non-trivial, as M interacts with the clipping term and can amplify its privacy-preserving effect.
> As elaborated in Section 4.1 and formally proven in Proposition B.2, even in the limit as M→∞, the variance of the sample average cannot be reduced below the threshold σ^2⋅∥X∥ ^2. This result is central to our argument: it demonstrates that—even if the adversary is willing to entirely sacrifice model utility—they still cannot directly recover the target input point by simply computing the sample average of their observations.
> Finally, we emphasise that Proposition B.2 serves a conceptual purpose and is not used to derive any other result in the paper.
> The changes concerning 'without loss of generality' have been incorporated into the proof of Proposition B.2.

---

> ### Author Response · Authors · 2025-05-28
> **Concern regarding claims**
>
> We introduce a new Section 5.2, where we discuss how practitioners can interpret our results in the context of $(\varepsilon, \delta)$-differential privacy guarantees. However, since there is no natural translation (equivalence) between our bounds and $(\varepsilon, \delta)$-DP guarantees, we have revised our claims to the following: 'By providing bounds for the success of data reconstruction attacks in real-world scenarios, we provide practitioners with a richer foundation for understanding specific reconstruction risks,' and 'We envision that such analyses could lead to comprehensive system cards—analogous to model cards (Mitchell et al.) —in which privacy risks across various contexts (i.e., threat models) are systematically described, thereby equipping practitioners with deeper insights into specific reconstruction risks.'
> These revised statements can be found at the end of the abstract and the introduction, respectively.
> Analogously, we have adapted our conclusion: “Our work provides the theoretical bounds on the risks for analytic gradient inversion attacks against the Gaussian mechanism and allows for a more precise evaluation of the potential privacy leakage in these settings.”

---

> ### Comment · Reviewer_PmVC · 2025-06-20
>
> I thank the authors for their follow up and revision of the manuscript. In particular, I would like to apologize for the (very poorly written) comment on the abscurity of the assumption on $C$: this was itself a *very* abscure statement... More concretely, I believe what is still confusing me is the threat model. It seems the attacker can change the architecture, but there is also an honest developer that sets the values of $C$ and $\sigma$. On the other hand, these hyper-parameters are generally optimized with respect to a fixed architecture (by the honest developer), and the attacker has no room of play with, for example, $M$. Implicitely, playing with $M$ is connected with playing with $C$, as these quantities are quite tied together (this is what I meant in the point later, and I would simply disregard the previous comment, and I apologize again for that).
>
> A think I would want to remark is that the authors state that they look at a single Gaussian mechanism only starting from the introduction section. This restrictive setup is not what one would think it is considered from, for example, the abstract. I acknowledge that solving the general problem with big batches and many training iterations is hard, but I would be more cautios in having statements connected to "how can practitioners choose the privacy parameters...". While I do believe this paper is correct and studies something legitimate, I do not believe the threat model is what practitioners have in mind when optimizing hyper-parameters. In this I agree with Reviewer DZ1Q. As final (minor) note, I agree with Reviewer KWnq on the perhaps redundant writing style of the paper, which could definitely be revised in a more compact way.

---

> ### Author Response · Authors · 2025-06-23
> **Follow up: On the threat model and clarifying our claims**
>
> Thank you very much for your follow-up and for clarifying your earlier comment regarding the assumption on $C$. We appreciate your engagement and the opportunity to further clarify our contribution.
>
> **On the Threat Model and the Interplay Between $C$ and $M$:**
>
> Perhaps we can help you resolve this confusion by making the threat model very concrete: Let’s assume we have a federated learning setup in a hub-and-spoke topology with a central server. This central server is controlled by a malicious entity, e.g., a hacker who took control of it. The central server has the authority to set the entire learning setup, including hyperparameters and model architecture. This is because the central server is an AI practitioner who is commissioned to train a model on the data of all participating sites.
> The clients trust the server to know best how to train the model. However, they have made precautions to limit the information leakage from their respective training data. Specifically, they perform the setup as suggested by the central server, but apply the Gaussian Mechanism with parameters that they choose.
> In other words, the setup is in full control of the central server, but the privacy guarantee is controlled by each client.
>
> As you rightly pointed out, manipulating $M$ is inherently linked to manipulating $C$. This connection is one of the central insights of our work. We *mathematically demonstrate* that an attacker can improve their reconstruction success by altering the model architecture—such as by increasing $M$—which affects the sensitivity of the gradients and, in turn, interacts with the clipping norm $C$. More importantly, we *quantify* this effect through the relationship between $M$ and $C$, as detailed in Proposition 1 and Corollary 1.
> Even when $C$ and $\sigma$ are fixed by an honest party, an attacker can still exploit this interaction to undermine semantic privacy guarantees. This is precisely why we focus on this scenario: to develop guarantees that remain *robust* even under such adversarial influence.
> One could instead assume a restricted adversary. However, assuming an attacker can insert a linear layer into the model without being able to choose $M$ to their benefit is contradictory.
>
> Naturally, there are complementary studies that explore how an honest party could optimise privacy parameters for a fixed architecture. However, our objective is different—we aim to understand the limits of semantic privacy when the architecture itself may be adversarially chosen. Analysing how semantic guarantees behave under specific tuning strategies falls outside the scope of this work, as such strategies are often case-specific and do not easily generalise.
>
> **On the Scope and Clarifying the Claims:**
>
> Thank you for pointing out that we could make it clearer earlier in the manuscript that we are considering a single execution of the Gaussian mechanism. To address this, we propose the following changes to the second paragraph of the abstract:
>
> **First sentence:**
> “Our theoretical contributions include (1) formulating an optimal attack strategy under the mean squared error for the specified threat model *against the Gaussian mechanism*,...”
>
> **Second-to-last sentence:**
> “Theoretically and empirically, our work underscores the protection *the Gaussian mechanism* provides against analytic gradient inversion attacks across varying privacy guarantees and model choices.”
>
> Additionally, we suggest rewriting the final two sentences of the **last paragraph of the introduction** to:
> “Our work provides practitioners with deeper insights into specific reconstruction risks *against the Gaussian mechanism*. We envision that such analyses can lead to full system cards (analogously to model cards (Mitchell et al., 2019)), where the privacy risks in different contexts (i.e., threat models) are described.”
>
> Please let us know if these changes help clarify your concerns—we would be happy to further revise the manuscript based on your suggestions.

---

### Author Response · Authors · 2025-05-28
**Overall response: Thanks for the constructive feedback**

Dear Reviewers,

We would like to express our sincere gratitude for your thorough and constructive feedback on our manuscript. We are pleased that you have all agreed that our paper studies a relevant problem and our methods are sound. We have carefully considered all comments and suggestions and believe that addressing them has significantly improved the clarity, rigour, and impact of our work.

Throughout the revision process, we focused on several key areas in response to your collective concerns. Importantly, we understand that your feedback primarily centred around the communication of our methods and results, the specific scenario/threat model we assume, and their practical usefulness.
We have uploaded a revised version of our manuscript in which all areas which have changed are marked up in green.

Specifically, we have:

**Enhanced Clarity & Organisation**: We have rewritten the Abstract and Introduction to provide a more focused narrative and clearly articulate the contributions of our work. While the key aspects remained the same, we have focused on making them more accessible.

**Expanded Background Material**: We moved backgrounds which were previously in the appendix and further expanded the Related Works section to provide essential context on Differential Privacy, Reconstruction Robustness (ReRo), and Analytic Gradient Inversion Attacks. This aims to improve accessibility.

**Improved Theoretical Explanations**: We have adapted the explanations surrounding our theoretical derivations, highlighting practical relevance and highlighted the main takeaways.

**Strengthened Practical Implications**: We have added a section on understanding the interpretation of our results to clarify how they can be used and offer benefits for practitioners (and what they cannot do).

**Refined Related Work**: The Related Work section has been thoroughly revised to better contextualise our research within the existing literature and highlight its novelty.

**Added Simple Composition Theorem**: We now also provide a preliminary worst-case estimate of reconstruction risk over multiple training steps. While currently based on naive assumptions, this represents an important direction for future work and offers initial insights into the cumulative impact of repeated attacks.

We are confident that these revisions address your concerns and significantly strengthen our manuscript. We look forward to hearing from you again soon. We are open to any questions and look forward to a constructive discussion.

Sincerely,

the Authors

---

### Decision · Action_Editor_gzyK · 2025-07-04

**Recommendation:** Reject

**Audience:**

No

**Audience Explanation:**

There is definitely a TMLR audience interested in gradient inversion attacks against FL. However, I think the scope of *this* paper is very specific, which limits its significance and potential audience. The derived bounds best apply to analytic gradient inversion under DP-SGD training where a malicious server can modify the model architecture, which is a very restrictive setting that is not applicable to any application that I am aware of. This is also a common criticism among reviewers and is cited as the reason for recommending rejection. For these reasons, I believe the audience for this paper is too limited for TMLR.

**Claims And Evidence:**

Yes

**Claims Explanation:**

The paper derives theoretical bounds for reconstruction risk and validates them with empirical attacks. These reconstruction risk bounds are well-defined and are grounded in prior work by Guo et al. (2022) and Balle et al. (2022).

Reviewers initially raised some concerns about the paper's message around practical implications and its formulation of assumptions. The authors clarified these points and rewrote a significant portion of the draft to incorporate reviewer suggestions. I believe these changes sufficiently addressed this concern.